# Robust Stochastic Gradient Posterior Sampling with Lattice Based Discretisation

Zier Mensch [1 2]  Lars Holdijk [3 4]  Samuel Duffield [4]
Maxwell Aifer [4]  Patrick J Coles [4]  Max Welling [5 6]  Miranda Cheng [1 7 8]

## Abstract

Stochastic-gradient MCMC methods enable scalable Bayesian posterior sampling but often suffer from sensitivity to minibatch size and gradient noise. To address this, we propose Stochastic Gradient Lattice Random Walk (SGLRW), an extension of the Lattice Random Walk discretisation. Unlike conventional Stochastic Gradient Langevin Dynamics (SGLD), SGLRW introduces stochastic noise only through the off-diagonal elements of the update covariance; this yields greater robustness to minibatch size while retaining asymptotic correctness. Furthermore, as a comparison we analyse a natural analogue of SGLD utilising gradient clipping. Experimental validation on Bayesian regression and classification demonstrates that SGLRW remains stable in regimes where SGLD fails, including in the presence of heavy-tailed gradient noise, and matches or improves predictive performance.

## 1. Introduction

Bayesian methods provide a principled framework for learning probabilistic models from data and natively capturing uncertainty by replacing the parameter point estimates in frequentist methods with a posterior distribution over parameters. By marginalising over parameters, Bayesian methods act as a form of regularisation and enable uncertainty quantification and robust model selection (Neal, 2012). In doing so, Bayesian models can potentially mitigate overfitting and miscalibration, which are prevalent in modern large-scale,

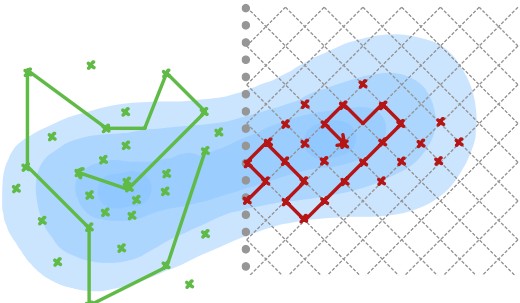

Figure 1. Comparing SGLD (**left**) and SGLRW (**right**) discretisations of Langevin dynamics, we can observe that the lattice-based discretisation suppresses large parameter jumps that occur due to minibatch noise, resulting in more stable sampling.

overparameterised neural networks (Guo et al., 2017; Yang et al., 2024). Realising these benefits in the modern hyper-scaling era, however, requires posterior inference algorithms that scale to both dataset size and model complexity.

Within Bayesian methods, Markov chain Monte Carlo (MCMC) (Neal, 1993; Robert et al., 1999) remains the gold standard for posterior sampling, but it is also among the methods most affected by scalability and computational cost (Gelman et al., 1997). Alternative approaches, including variational inference (Blei et al., 2017), Laplace approximations (Tierney & Kadane, 1986), and single-pass methods (Gal & Ghahramani, 2016), are often less computationally demanding, but still introduce substantial overhead in training and inference (Blei et al., 2017; Lakshminarayanan et al., 2017; Wilson & Izmailov, 2020). As a result, these methods have, in some settings, fallen out of favour relative to modern approaches for assessing model trustworthiness, such as explainable and interpretable models (Li et al., 2023a).

One core issue of MCMC methods for Bayesian posterior inference is the theoretical requirement to evaluate the gradient of the posterior over the entire dataset at each iteration (Welling & Teh, 2011; Ma et al., 2015). With growing model complexity and dataset size, this is often prohibitively expensive. Stochastic-gradient variants of MCMC methods, such as Stochastic Gradient Langevin Dynamics (SGLD) (Welling & Teh, 2011), alleviate this concern to some extent by allowing the gradient to be evaluated on a small minibatch of data at each iteration. However, these methods

---

[1]Institute of Physics, University of Amsterdam, Netherlands
[2]Department of Physics, National Taiwan University, Taiwan
[3]Department of Computer Science, University of Oxford, United Kingdom [4]Normal Computing Corporation, New York, New York, USA [5]Amsterdam Machine Learning Lab, University of Amsterdam, Netherlands [6]CuspAI, Amsterdam, Netherlands [7]Institute for Mathematics, Academia Sinica, Taiwan [8]Korteweg-de Vries Institute for Mathematics, University of Amsterdam, Netherlands. Correspondence to: Lars Holdijk <larsholdijk@gmail.com>.

*Proceedings of the 43rd International Conference on Machine Learning*, Seoul, South Korea. PMLR 306, 2026. Copyright 2026 by the author(s).

are still known to be sensitive to the minibatch size (Brosse et al., 2018). As a result, they do not scale to regimes where only a small minibatch is available at each evaluation step, or where only a small number of samples from the dataset can be stored in memory, as is becoming increasingly common.

In this work, we propose Stochastic Gradient Lattice Random Walk (SGLRW), a stochastic-gradient extension of the recently introduced Lattice Random Walk (LRW) (Duffield et al., 2025a) discretisation of overdamped Langevin dynamics. LRW replaces the Gaussian increments of Langevin dynamics with bounded binary or ternary updates on a lattice. As we show, unlike SGLD, the stochastic gradient noise in SGLRW enters only through the off-diagonal elements of the covariance matrix of the update and therefore remains robust to the minibatch size. This allows SGLRW to sample from the posterior distribution with the same asymptotic correctness as SGLD, but with improved stability for small minibatches, as shown in Figure 1.

In short, our contributions are as follows:

- We propose Stochastic Gradient Lattice Random Walk (SGLRW), a lattice-based stochastic-gradient discretisation of overdamped Langevin dynamics (Section 4.2).

- Extending the analysis of Chen et al. (2015), we provide a mean-squared-error analysis that justifies its improved stability for small minibatches (Section 4.3).

- We validate our theoretical findings on a mix of analytically understood problems and real-world tasks, including sentiment classification using an LLM (Section 5).

- We discuss a clipped version of SGLD as a strong baseline that is analogous to gradient clipping in stochastic gradient descent (Section 5.1).

## 2. Background

As stated in the introduction, we consider the problem of minibatch-induced instability in stochastic gradient MCMC methods for Bayesian posterior sampling. Here, we recap the necessary background on Bayesian machine learning, posterior inference, and stochastic gradient methods.

**Bayesian Machine Learning.** We consider the supervised learning setting where we have observed data $\mathcal{D} = \{(x_i, y_i)\}_{i=1}^N$ and aim to infer a posterior distribution $p(\theta \mid \mathcal{D})$ over the parameter vector $\theta \in \mathbb{R}^d$. In contrast to frequentist approaches, which seek a single point estimate $\theta^*$ that maximises the likelihood $p(\mathcal{D} \mid \theta)$, Bayesian machine learning maintains a full distribution over parameters, $p(\theta \mid \mathcal{D})$. This *posterior distribution* captures the uncertainty in our parameter estimates given the observed data.

The posterior distribution is given by Bayes' theorem as

$$p(\theta \mid \mathcal{D}) = \frac{p(\mathcal{D} \mid \theta)p(\theta)}{p(\mathcal{D})} \propto p(\theta) \prod_{i=1}^N p(y_i \mid x_i, \theta), \quad (1)$$

where $p(\mathcal{D} \mid \theta)$ is again the *likelihood* and $p(\theta)$ is the *prior*. Notably, we will often write the posterior distribution as $p(\theta \mid \mathcal{D}) \propto \exp[-U(\theta)]$ where

$$U(\theta) = -\log p(\theta) - \sum_{i=1}^N \log p(y_i \mid x_i, \theta), \quad (2)$$

is the negative log-posterior.

Given the posterior distribution, predictions for a new input $x^*$ are then obtained via the posterior predictive distribution,

$$p(y^* \mid x^*, \mathcal{D}) = \int p(y^* \mid x^*, \theta)\, p(\theta \mid \mathcal{D})\, d\theta, \quad (3)$$

which marginalises over the parameters $\theta$ and can thus provide calibrated predictive distributions. Crucially, however, evaluating (3) is generally infeasible, as it involves a high-dimensional integral over $\theta$.

### 2.1. Bayesian Posterior Sampling

A common approach to Bayesian inference is to replace the integral in (3) with a Monte Carlo average. Most commonly, this is achieved by drawing parameter samples $\theta \sim p(\theta \mid \mathcal{D})$ from the posterior using a Markov chain Monte Carlo (MCMC) approach.

In this work, we specifically focus on MCMC samplers that are expressed as discretisations of stochastic differential equations (SDEs) whose stationary distribution coincides with the target posterior (Ma et al., 2015). Among these, the most common choice is the overdamped Langevin diffusion,

$$d\theta_t = f(\theta_t)\, dt + \sqrt{2D(\theta_t)}\, dW_t, \quad (4)$$

where $D(\theta_t)$ is a symmetric positive semidefinite diffusion matrix and $f(\theta_t)$ is the drift. In the context of Bayesian posterior sampling, the drift is given by $f(\theta_t) = -\nabla U(\theta_t)$ and the diffusion matrix is typically set to $D(\theta_t) = I$, resulting in the following SDE:

$$d\theta_t = -\nabla U(\theta_t)\, dt + \sqrt{2}\, dW_t \quad (5)$$

$$= (\nabla \log p(\theta_t) + \sum_{i=1}^N \nabla \log p(y_i \mid x_i, \theta_t))dt + \sqrt{2}\, dW_t. \quad (6)$$

#### 2.1.1. STOCHASTIC GRADIENT METHODS.

As discussed in the introduction, in large-scale settings the requirement to evaluate the gradient of the posterior over the

entire dataset at each iteration is limiting. As such, stochastic gradient MCMC (SG-MCMC) methods replace the full posterior gradient with an unbiased minibatch estimator:

$$\widehat{\nabla U}(\theta; \mathcal{B}) = -\nabla \log p(\theta) - \frac{N}{B} \sum_{i \in \mathcal{B}} \nabla \log p(y_i \mid x_i, \theta),$$
(7)

where $\mathcal{B} \subset \{1, \ldots, N\}$ is a minibatch index set of size $B$, and $\{(x_i, y_i)\}_{i \in \mathcal{B}} \subset \mathcal{D}$ are the corresponding datapoints.

**Stochastic Gradient Langevin Dynamics.** Stochastic Gradient methods for Bayesian posterior sampling were first introduced by Welling & Teh (2011) for Langevin dynamics, and only later generalised to other MCMC samplers (Chen et al., 2014; Ma et al., 2015). Concretely, Welling & Teh (2011) obtain the following Stochastic Gradient Langevin Dynamics (SGLD) update rule:

> **Definition 2.1** (SGLD Update Rule). Given a minibatch $\mathcal{B}$, applying the Euler-Maruyama discretisation to the Langevin SDE Equation (5) and replacing the full gradient with a minibatch estimate yields the SGLD update:
>
> $$\theta_{t+1} = \theta_t - \delta_t \widehat{\nabla U}(\theta_t; \mathcal{B}) + \sqrt{2\delta_t}\, \xi_t, \qquad (8)$$
>
> where $\xi_t \sim \mathcal{N}(0, I)$.

For the original full-gradient Langevin dynamics, Euler-Maruyama has weak order 1 convergence. However, with stochastic gradients, the convergence properties are more subtle (Vollmer et al., 2016). With a fixed step size $\delta_t = \delta$, SGLD converges to a stationary distribution that is biased relative to the true posterior, reflecting both discretisation error and the noise from minibatch gradients. To obtain asymptotically exact expectations, a decreasing step size schedule satisfying

$$\delta_t \to 0, \quad \sum_t \delta_t = \infty, \quad \sum_t \delta_t^2 < \infty, \qquad (9)$$

needs to be used (Teh et al., 2016). However, this does come at the cost of slower mixing as the step size vanishes.

In our analysis below, we decompose the stochastic gradient as $\widehat{\nabla U}(\theta; \mathcal{B}) = \nabla U(\theta) + \zeta(\theta; \mathcal{B})$ and define $G(\theta)$ to be the minibatch-induced gradient covariance, $\text{Cov}_{\mathcal{B}}[\zeta(\theta; \mathcal{B}) \mid \theta] = G(\theta)$. Clearly we have $\mathbb{E}_{\mathcal{B}}[\zeta(\theta; \mathcal{B}) \mid \theta] = 0$.

## 3. Related Work

We now briefly review related work on batch-size sensitivity in SG-MCMC methods and large-scale Bayesian inference.

**Batch-Size Sensitivity.** As stated, in practice, SGLD often requires large minibatches for stability, limiting scalability (Baker et al., 2019). Variance-reduction and control-variate methods reduce gradient noise via auxiliary reference gradients (Dubey et al., 2016; Baker et al., 2019; Li et al., 2023b); adaptive subsampling (Korattikara et al., 2014), preconditioning (Li et al., 2016), importance sampling (Li et al., 2023b), and cyclic schedules (Zhang et al.) likewise address gradient noise or chain mixing rather than the discretisation itself. SGLRW is orthogonal to these design choices and can be composed with them (e.g., cyclic or preconditioned SGLRW). Moreover, recent work characterises stochastic gradient noise as heavy-tailed rather than Gaussian (Simsekli et al., 2019) and proposes specialised fractional dynamics to retarget the sampling distribution when such noise is present (Simsekli et al., 2020). While these methods primarily address the statistical properties of the noise, our solution improves robustness through its lattice-based discretisation, enabling stable sampling with small minibatches.

**Large-Scale Bayesian Inference.** Bayesian uncertainty estimation is important for large neural models, including large language models, where calibration and robustness are critical (Yang et al., 2024). Scalable approximations such as Laplace methods (Daxberger et al., 2021; Yang et al., 2024; Chen & Garner, 2024; Sliwa et al., 2025) and variational inference (Harrison et al.; Wang et al., 2024; Xiang et al., 2025; Samplawski et al., 2025) trade accuracy for efficiency. Recent work shows that sampling-based methods can be applied to large models with appropriate algorithmic structure, as in SGLD-Gibbs (Kim & Hospedales, 2025).

## 4. Stochastic Gradient Lattice Random Walk

With the background established in the previous section, we now introduce the proposed method, Stochastic Gradient Lattice Random Walk (SGLRW). For this purpose, we first review the Lattice Random Walk (LRW) discretisation of Langevin dynamics and then introduce the proposed stochastic gradient extension.

### 4.1. Lattice Random Walk

The Lattice Random Walk (LRW) scheme, recently introduced in (Duffield et al., 2025a), proposes an alternative to standard SDE discretisations, such as Euler-Maruyama, by substituting Gaussian noise with bounded binary increments. Lattice and related skew-symmetric schemes admit weak-convergence guarantees under one-sided or local Lipschitz drift conditions (Duffield et al., 2025a; Iguchi et al., 2026), weaker than the global Lipschitz assumption typically required and often violated in deep learning. Additionally, the structure of LRW also lends itself to low-precision, stochas-

tic hardware (Alaghi & Hayes, 2013) and thermodynamic hardware (Conte et al., 2019) which has recently started to be developed for AI applications (Melanson et al., 2025).

In LRW, at each iteration, the parameters are updated as

$$\Delta\theta_{t+1} = S_t, \qquad (S_t)_i \in \{-\sqrt{2\delta_t}, +\sqrt{2\delta_t}\}, \qquad (10)$$

where each direction $(S_t)_i$ is sampled independently from the following state-dependent probability

$$\mathbb{P}\Big[(S_t)_i = \pm\sqrt{2\delta_t} \,\Big|\, \theta_t\Big] = \tfrac{1}{2} \pm \tfrac{1}{2}\sqrt{\tfrac{\delta_t}{2}}\, f_i(\theta_t). \qquad (11)$$

These probabilities are valid whenever $\sqrt{\delta_t/2}\,|f_i(\theta_t)| \le 1$, where we write $f_i(\theta)$ to denote the $i$th component of the drift vector field $f(\theta)$.

By construction, the first two conditional moments satisfy

$$\mathbb{E}[S_t \mid \theta_t] = \delta_t f(\theta_t), \qquad \mathbb{E}[S_t S_t^\top \mid \theta_t] = 2\delta_t I, \qquad (12)$$

and LRW is shown to be weakly first-order consistent with the continuous-time Langevin dynamics in Equation (4) (Theorem 1 of (Duffield et al., 2025a)). With the specific choice of $f(\theta) = -\nabla U(\theta)$, LRW thus provides a valid discretisation of the Langevin dynamics in Equation (5).

### 4.2. Stochastic Gradient Lattice Random Walk

We now come to the main contribution of our work, the proposal of Stochastic Gradient Lattice Random Walk (SGLRW), which replaces the stochastic gradient update rule of SGLD with a lattice-based update:

> **Definition 4.1** (SGLRW Update Rule). Given a mini-batch $\mathcal{B}$, at the $t$th iteration, the Stochastic Gradient Lattice Random Walk updates the parameter vector as
>
> $$\theta_{t+1} = \theta_t + S_t. \qquad (13)$$
>
> where each coordinate $(S_t)_i \in \{-\sqrt{2\delta_t}, +\sqrt{2\delta_t}\}$ is

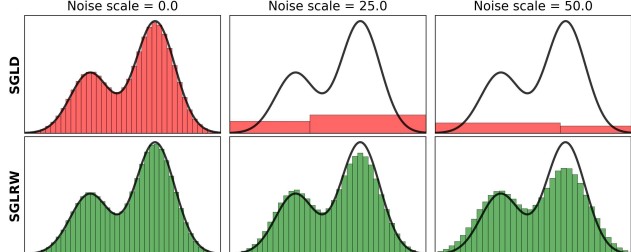

*Figure 2.* Multimodal univariate target with exact gradient corrupted by synthetic $\alpha$-stable noise ($\alpha = 1.5$) of increasing scale. We observe that as we increase the noise scale SGLD quickly fails while SGLRW remains stable.

sampled from the state-dependent probability

$$\mathbb{P}\Big[(S_t)_i = \pm\sqrt{2\delta_t} \,\Big|\, \theta_t\Big] = \tfrac{1}{2} \mp \tfrac{1}{2}\sqrt{\tfrac{\delta_t}{2}}\, \widehat{\partial_i U}(\theta_t; \mathcal{B}), \qquad (14)$$

which is valid whenever $\sqrt{\delta_t/2}\,|\widehat{\partial_i U}(\theta_t; \mathcal{B})| \le 1$.

Per-iteration cost is $\mathcal{O}(d)$, as in SGLD, and no Gaussian sampling is required. We hypothesise, and analytically evaluate next, that due to the bounded structure, large fluctuations in stochastic gradients have a less severe impact on the update in the case of SGLRW than in the case of SGLD. This is illustrated in Figure 2, in a one-dimensional multimodal example.

**Heavy-Tailed Noise.** Using the stochastic gradient as $\widehat{\nabla U}(\theta; \mathcal{B}) = \nabla U(\theta) + \zeta(\theta; \mathcal{B})$, we set $U(\theta)$ to be the negative log-probability of the multimodal Gaussian, and choose $\zeta(\theta; \mathcal{B})$ to follow a heavy-tailed $\alpha$-stable distribution with $\alpha < 2$, for which second moments do not exist. This distribution was shown to closely resemble the minibatch gradient noise in the standard SGD setting by Simsekli et al. (2019).

As Figure 2 shows, in this regime, where stability depends critically on whether large stochastic fluctuations can induce rare but catastrophic updates, SGLD fails while SGLRW remains stable. In Appendix A.4 we extend this comparison to Clipped-SGLD and to Gaussian noise ($\alpha = 2$) alongside the heavy-tailed case (Figure 7); both SGLRW and Clipped-SGLD avoid SGLD's catastrophic failure, with SGLRW maintaining closer agreement with the target distribution.

### 4.3. Mean Squared Error Analysis

Having introduced SGLRW, we now present an analysis of the differences between SGLD and SGLRW that highlights the benefits of using SGLRW with small batch sizes.

We follow a similar approach to the analysis of SGLD in Chen et al. (2015), focussing on the mean squared error (MSE) $\mathbb{E}(\hat{\phi} - \bar{\phi})^2$ between the true posterior expectation

$$\bar{\phi} := \int \phi(\theta)\, p(\theta \mid \mathcal{D})\, d\theta. \qquad (15)$$

and the ergodic average

$$\hat{\phi} := \frac{1}{L}\sum_{n=1}^{L} \phi(\theta_{n\delta_t}), \qquad (16)$$

over the discrete-time Markov chain $\{\theta_{n\delta_t}\}_{n\ge 0}$ generated by an SG-MCMC method, such as SGLD or SGLRW, with step size $\delta_t$. Here, $\phi : \mathbb{R}^d \to \mathbb{R}^m$ represents a smooth test function, such as the posterior predictive distribution of a new data point $x^*$, as defined in Equation (3).

As a comparison of the MSE for SGLRW and SGLD, we present the following theorem:

**Theorem 4.2.** *Under Assumption A.1, we find that the MSE is bounded by the following three contributions*

$$MSE \leq C \left( \mathcal{E}_{drift} + \mathcal{E}_{disc} + \mathcal{E}_{cov} \right)$$

*for some $C$ that depends on the target distribution. The covariance error term for SGLRW is never larger than that of SGLD*

$$\mathcal{E}_{cov}^{SGLRW} \leq \mathcal{E}_{cov}^{SGLD}$$

*while the other contributions are the same for both. Moreover, it is strictly smaller whenever $2\partial_i U \zeta_i + \zeta_i^2$ is non-vanishing for some direction $i$.*

Assumption A.1 adopts the Lyapunov-Poisson regularity setup of Chen et al. (2015) (Lyapunov-controlled derivatives of the Poisson solution, uniformly bounded chain moments, and unbiased minibatching), supplemented by bounded second- and third-moment controls on the increment for the refined covariance term. The first statement regarding the MSE upper bound, and the precise expression for each of the three contributions, is the content of Theorem A.4, which is an extension of Theorem 3 of Chen et al. (2015) to take into account non-vanishing second-order contributions. Under the decreasing-step-size schedule of Equation (9), the right-hand side of this bound vanishes as $L \to \infty$ for SGLD (Chen et al., 2015; Teh et al., 2016); since SGLRW inherits the same drift and discretisation contributions while $\mathcal{E}_{cov}^{SGLRW} \leq \mathcal{E}_{cov}^{SGLD}$, the same conditions yield $\mathbb{E}(\hat{\phi} - \bar{\phi})^2 \to 0$ for SGLRW. Theorem 4.2 therefore provides a rigorous convergence guarantee for ergodic averages of smooth test functions under SGLRW, rather than only a relative comparison against SGLD. In particular, the covariance contribution is given by

$$\mathcal{E}_{\text{cov}} = \frac{\delta_t^2}{L} \sum_{n=1}^{L} \mathbb{E}\left[ \|M_n\|_F^2 \right]. \tag{17}$$

In the above, the scheme-dependent second-order error $M_n$ induced by minibatching is defined as

$$M_n(\theta; \mathcal{B}_n) := \delta_t^{-2} \, \mathbb{E}_{\varepsilon_n} \Big[ \Delta\theta_n^{\text{mb}} (\Delta\theta_n^{\text{mb}})^\top - \Delta\theta_n^{\text{fb}} (\Delta\theta_n^{\text{fb}})^\top \, \Big| $$
$$\theta_{(n-1)\delta_t} = \theta, \mathcal{B}_n \Big], \tag{18}$$

where $\Delta\theta_n^{\text{mb}}$ and $\Delta\theta_n^{\text{fb}}$ denote the one-step increments of the minibatch and full-batch updates, respectively.

To prove the bound in Theorem 4.2, we observe a lemma quantifying the difference between the second-order structure of SGLD and SGLRW

**Lemma 4.3.** *The second moment error of the minibatch update for SGLRW satisfies*

$$M_{n,\text{SGLRW}}(\theta; \mathcal{B}_n) = \text{offdiag}\big( M_{n,\text{SGLD}}(\theta; \mathcal{B}_n) \big), \tag{19}$$

*where $M_{n,\text{SGLD}}(\theta, \mathcal{B}_n)$ is the second-order error for SGLD given by*

$$M_{n,\text{SGLD}} = \zeta \zeta^\top + \nabla U \zeta^\top + \zeta \nabla U^\top.$$

*Proof.* See Appendix A.2. □

The above lemma highlights the fact that, in SGLRW, the lattice constraint enforces fixed-magnitude coordinate updates, so the diagonal of the one-step second moment of the increment is deterministic. In contrast, for SGLD this diagonal depends on the stochastic gradient and is inflated by minibatch noise.

Combining the above, we readily obtain the error bound of Theorem 4.2. This shows that under some mild conditions, the SGLRW discretisation achieves a strictly tighter MSE bound than SGLD, leading to a more robust implementation of minibatch gradient updates.

### 4.3.1. VALIDATION AND PRACTICAL CONSIDERATIONS

We experimentally validate this theoretical finding in Figure 3, where we compare the MSE for SGLRW and SGLD for Bayesian linear and logistic regression. As $\delta$ increases, SGLD becomes unstable and the covariance error explodes, while SGLRW remains stable for all batch sizes. A further discussion of the experimental setup used to generate this insight is provided in Section 5.2.

The MSE bound of Theorem 4.2 guarantees that the ergodic-average error for SGLRW vanishes under the decreasing-step-size schedule of Equation (9), while inheriting a strictly smaller covariance contribution than SGLD. At a fixed step size $\delta_t = \delta$, however, the Markov chain generated by SGLRW (like SGLD and other SG-MCMC schemes) admits a stationary distribution that differs from the target posterior by a non-vanishing $O(\delta)$ discretisation bias (Vollmer et al., 2016; Teh et al., 2016). We report empirical magnitudes of this fixed-step bias for SGLRW, SGLD, and Clipped-SGLD across batch sizes and step sizes in Appendix A.10.

The analysis above concerns the idealised update of Definition 4.1; in the practical implementation we clip $\sqrt{\delta_t/2}\,|\widehat{\partial_i U}(\theta_t; \mathcal{B})|$ to one when the validity restriction is violated, rather than tuning $\delta_t$. The resulting bias is empirically negligible in the regimes of interest, with per-coordinate clipping rates and their dimension-dependence reported in Section A.6.

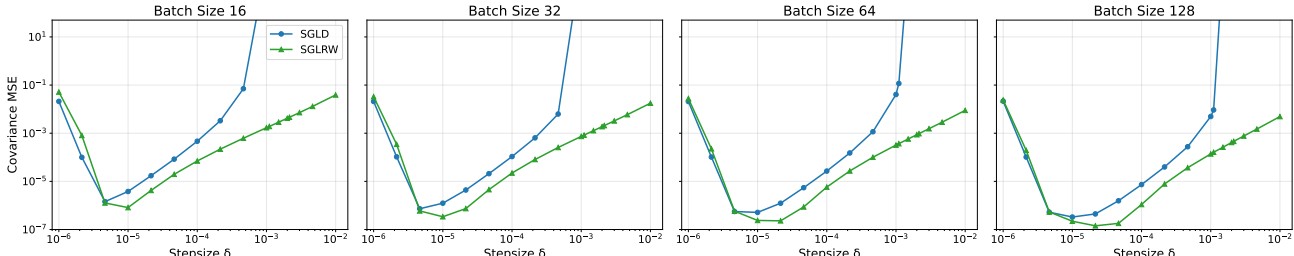

*Figure 3.* Mean-squared error (MSE) of the posterior covariance as a function of the step size $\delta$, shown for different batch sizes for 50-dimensional Bayesian linear regression.

## 5. Experimental Evaluation

We evaluate SGLRW on posterior sampling problems for linear regression, logistic regression, and predictive classification tasks. Across all experiments, we vary the minibatch size $B$ and base step size $\delta_0$ under decaying step-size schedules in the spirit of Welling & Teh (2011); the exact schedule is specified in each experiment's setup paragraph, and reported step-size values refer to $\delta_0$ throughout. Parameters are initialised from a standard normal for the regression tasks and via Xavier initialisation for the LLM classifier head, identically across seeds. All experiments were implemented in `posteriors` (Duffield et al., 2025b).

### 5.1. Strong Baseline: Clipped-SGLD.

Similar to our analysis in Section 4.3, we also compare SGLRW against SGLD in our empirical evaluation here. Additionally, we introduce *Clipped-SGLD* as an additional strong baseline. Gradient clipping is standard in large-scale SGD, where saturating the drift prevents rare large gradients from producing unstable updates. It is therefore natural to ask whether the same stabilisation can be applied to SGLD.

We define the Clipped-SGLD update rule as follows:

$$\theta_{t+1} = \theta_t - \text{clip}(\delta_t \widehat{\nabla U}(\theta_t; \mathcal{B}); R) + \sqrt{2\delta_t}\,\xi_t, \quad (20)$$

where $\text{clip}(x; R)_i = \text{sign}(x_i)\min\{|x_i|, R\}$ and $R = \sqrt{2\delta_t}$. Note that this is a componentwise clipping operation, and deviates slightly from the standard definition of gradient clipping in SGD. In gradient clipping for standard SGD, the clipping is performed over the entire update vector $\Delta\theta_t$, while here we clip only part of the update vector. Clipping the entire update vector would result in the SDE having a different stationary distribution (Appendix A.3), while drift-truncated Euler schemes are known to converge to the exact Langevin dynamics in the small-step limit (Roberts & Tweedie, 1996; Hutzenthaler & Jentzen, 2015).

In Appendix A.4 we provide the same MSE and heavy-tailed noise analysis for Clipped-SGLD as previously discussed in Section 4.3 for SGLRW.

*Table 1.* Kullback–Leibler (KL) divergence between the true posterior and the empirical Gaussian fit of the samples, shown for different samplers (SGLD, SGLRW, Clipped-SGLD), minibatch size $B$, and base learning rate $\delta_0$. Values are means over 12 independent seeds; full results with standard deviations are reported in Section A.5. The Monte Carlo reference KL divergence, which quantifies the intrinsic sampling variability when estimating the analytic posterior, is 0.055201. **Bold** indicates the lowest mean KL for a given $(B, \delta_0)$ pair, together with any method whose $\pm 1\sigma$ interval overlaps the minimum (i.e., statistically indistinguishable).

| Hyperparameters | | KL Divergence | | |
|---|---|---|---|---|
| $B$ | $\delta_0$ | SGLD | SGLRW | Clipped-SGLD |
| 8 | $10^{-3}$ | 19.853 | **6.060** | 18.496 |
| 8 | $10^{-4}$ | 0.504 | **0.196** | 0.822 |
| 16 | $10^{-3}$ | 7.143 | **2.328** | 8.661 |
| 16 | $10^{-4}$ | 0.174 | **0.068** | 0.218 |
| 32 | $10^{-3}$ | 2.406 | **0.722** | 3.470 |
| 32 | $10^{-4}$ | 0.092 | **0.061** | 0.092 |
| 64 | $10^{-3}$ | 0.817 | **0.189** | 1.169 |
| 64 | $10^{-4}$ | **0.062** | **0.055** | **0.066** |
| 128 | $10^{-3}$ | 0.277 | **0.075** | 0.370 |
| 128 | $10^{-4}$ | **0.061** | **0.058** | **0.058** |
| 256 | $10^{-3}$ | 0.135 | **0.060** | 0.139 |
| 256 | $10^{-4}$ | **0.057** | **0.055** | **0.060** |
| 512 | $10^{-3}$ | 0.086 | **0.059** | 0.084 |
| 512 | $10^{-4}$ | **0.057** | **0.056** | **0.058** |
| 1000 | $10^{-3}$ | **0.072** | **0.058** | 0.073 |
| 1000 | $10^{-4}$ | **0.057** | **0.059** | **0.057** |

### 5.2. Bayesian Linear Regression

We first evaluate SGLRW using a linear–Gaussian model where the posterior admits a closed form. Using the closed-form solution, we can analytically compute the KL divergence between the true posterior and the empirical Gaussian fit to the samples. This allows us to provide a more rigorous evaluation of the empirical performance of SGLRW compared to SGLD and Clipped-SGLD.

Concretely, the linear model we consider is given by

$$y = X\theta + \varepsilon, \qquad \varepsilon \sim \mathcal{N}(0, \sigma^2 I), \quad (21)$$

where $X \in \mathbb{R}^{N\times d}$ is the design matrix and $\varepsilon \in \mathbb{R}^N$ is the noise vector. With a Gaussian prior $p(\theta) = \mathcal{N}(0, \tau^{-1}I)$,

the resulting posterior $\mathcal{N}(\mu, \Sigma)$ is therefore given by

$$\Sigma^{-1} = \frac{1}{\sigma^2} X^\top X + \tau I, \qquad \mu = \frac{1}{\sigma^2} \Sigma X^\top y. \qquad (22)$$

Or, equivalently, in the negative log-posterior form,

$$U(\theta) = \frac{1}{2\sigma^2} \|y - X\theta\|^2 + \frac{\tau}{2} \|\theta\|^2. \qquad (23)$$

**Setup.** Synthetic data are generated with $N = 1000$ and $d = 20$, using $\theta^* \sim \mathcal{N}(0, I)$, $\sigma^2 = 1.0$, and $\tau = 10^{-2}$. Each method is run with 2,000 parallel particles for 10,000 iterations, using matched minibatch sizes and the decaying schedule $\delta_t = \delta_0(1 + t)^{-0.33}$.

As stated, the performance is quantified using the analytic Kullback–Leibler divergence between the true posterior $\mathcal{N}(\mu, \Sigma)$ and the empirical Gaussian fit to the samples, computed from their estimated mean and covariance.

### 5.2.1. RESULTS

The KL curves reveal two consistent effects, portrayed in Table 1: (i) *step size sensitivity:* SGLRW remains stable and continues to decrease KL under larger base step sizes $\delta_0$ where SGLD diverges. (ii) *batch efficiency:* for comparable KL at matched $\delta_0$, SGLRW achieves the same accuracy with approximately half the minibatch size, indicating greater robustness to stochastic-gradient noise.

These trends are accompanied by different empirical covariance behaviour across methods: the error in the diagonal terms of the estimated covariance matrices is significantly lower for SGLRW than SGLD and Clipped-SGLD, and in general less impacted by the batch size (see Figure 8 in Section A.5 for the per-method covariance difference matrices).

**Cyclic SGLRW.** Cyclic learning-rate schedules (Zhang et al.) alternate exploration and sampling stages via a cosine annealing of $\delta_t$. To test whether this enhancement composes with SGLRW, we feed the same cosine schedule into the SGLRW update of Definition 4.1, defining cyclic SGLRW. On the linear regression setup of Section 5.2, cyclic SGLRW is stable across the full $(B, \delta_0)$ grid, whereas cyclic SGLD diverges at $\delta_0 = 10^{-2}$ for every batch size and at $\delta_0 = 10^{-3}$ for $B \leq 16$. Where both samplers converge, cyclic SGLRW reaches the Monte Carlo floor and substantially improves on cyclic SGLD under small-batch stress (Table 2; full grid in Appendix A.7).

**Preconditioned SGLRW.** Preconditioned SGLD (Li et al., 2016) uses an RMSprop diagonal preconditioner to handle anisotropic posteriors. Applying the same diagonal preconditioner to the drift and the lattice step size of SGLRW gives pSGLRW. We benchmark all four samplers on a $d = 100$ Gaussian target with condition number $\kappa = 200$. Vanilla SGLD diverges at all but the smallest step

*Table 2.* Cyclic SGLRW vs. cyclic SGLD on Bayesian linear regression. KL divergence as means over 12 seeds. Full grid with standard deviations is reported in Appendix A.7. **Bold** marks the lower mean, together with any method whose $\pm 1\sigma$ interval overlaps the minimum. — indicates divergence on all seeds.

| Hyperparameters | | KL Divergence | |
|---|---|---|---|
| $B$ | $\delta_0$ | cyclic-SGLD | cyclic-SGLRW |
| 8 | $10^{-2}$ | — | **2.48** |
| 16 | $10^{-3}$ | — | **0.138** |
| 32 | $10^{-3}$ | 0.156 | **0.067** |
| 64 | $10^{-3}$ | 0.082 | **0.059** |

*Table 3.* KL divergence on a $d = 100$ anisotropic Gaussian (condition number $\kappa = 200$) at constant step size $\delta_0$, as means over 10 seeds. Full results with standard deviations are reported in Appendix A.8. **Bold** marks the lowest mean per row, together with any method whose $\pm 1\sigma$ interval overlaps the minimum. — indicates divergence on all seeds.

| $\delta_0$ | SGLD | SGLRW | pSGLD | pSGLRW |
|---|---|---|---|---|
| 0.0300 | — | 28.72 | 0.764 | **0.637** |
| 0.0270 | — | 20.51 | 0.742 | **0.622** |
| 0.0243 | — | 13.89 | 0.716 | **0.622** |
| 0.0219 | — | 9.19 | 0.730 | **0.651** |
| 0.0197 | 44.58 | 6.09 | 0.743 | **0.660** |

size while SGLRW remains stable across the sweep but with elevated KL; preconditioning recovers both methods, and pSGLRW achieves lower KL than pSGLD at every step size (Table 3; setup details in Appendix A.8).

### 5.3. UCI Bayesian Logistic Regression

We now compare the sensitivity of SGLRW, SGLD, and Clipped-SGLD on a non-Gaussian posterior sampling task, specifically logistic regression with the breast cancer dataset (Wolberg et al., 1993). The UCI breast cancer dataset consists of 569 samples, 30 features and 2 classes, which results in a 31-dimensional posterior distribution.

**Setup.** Since the true posterior is not available analytically, we compare to a gold-standard sample generated with NUTS (Hoffman et al., 2014) via Pyro (Bingham et al., 2019). Throughout, we use a standard Gaussian prior on all parameters. In all cases, we ran 5,000 parallel chains for 1,000 steps under the decaying schedule $\delta_t = \delta_0(1+t)^{-0.55}$, retaining only the final sample of each chain. All runs are averaged over 12 seeds.

### 5.3.1. RESULTS

Comparing the inferred KL divergence of the three different methods in Table 4, we see that SGLRW consistently outperforms SGLD and Clipped-SGLD across learning-rate settings, similar to what was observed in the linear regression experiment. However, in contrast to the linear regression experiments where Clipped-SGLD performed

*Table 4.* Inferred Kullback–Leibler divergence for the logistic regression problem. The KL divergence is measured between Gaussian distributions fitted to the empirical mean and covariance of a gold-standard NUTS reference sample and those obtained by each algorithm under the specified hyperparameters. Values are means over 12 independent seeds; full results with standard deviations are reported in Section A.9. **Bold** indicates the lowest mean KL for a given $(B, \delta_0)$ pair, together with any method whose $\pm 1\sigma$ interval overlaps the minimum.

| Hyperparameters | | KL Divergence | | |
|---|---|---|---|---|
| $B$ | $\delta_0$ | SGLD | SGLRW | Clipped-SGLD |
| 1 | $10^0$ | 46.351 | **8.230** | 9.276 |
| 1 | $10^{-1}$ | 16.536 | **5.884** | 6.420 |
| 1 | $10^{-2}$ | 10.300 | **3.845** | 4.162 |
| 2 | $10^0$ | 36.271 | **7.582** | 9.004 |
| 2 | $10^{-1}$ | 8.901 | **4.868** | 5.515 |
| 2 | $10^{-2}$ | 5.387 | **2.521** | 2.774 |
| 4 | $10^0$ | 27.221 | **6.870** | 8.811 |
| 4 | $10^{-1}$ | 3.586 | **3.522** | 4.231 |
| 4 | $10^{-2}$ | 2.894 | **1.393** | 1.550 |
| 8 | $10^0$ | 18.889 | **5.904** | 8.333 |
| 8 | $10^{-1}$ | **1.228** | 1.838 | 2.395 |
| 8 | $10^{-2}$ | 1.876 | **0.930** | **0.988** |
| 16 | $10^0$ | 11.756 | **4.385** | 6.973 |
| 16 | $10^{-1}$ | **0.454** | 0.882 | 1.145 |
| 16 | $10^{-2}$ | 1.441 | **0.765** | **0.757** |
| 32 | $10^0$ | 6.308 | **2.733** | 4.762 |
| 32 | $10^{-1}$ | **0.219** | 0.411 | 0.498 |
| 32 | $10^{-2}$ | 1.251 | **0.647** | **0.634** |
| 64 | $10^0$ | 2.914 | **1.570** | 2.783 |
| 64 | $10^{-1}$ | **0.158** | 0.183 | 0.206 |
| 64 | $10^{-2}$ | 1.161 | **0.618** | **0.599** |

roughly similarly to SGLD and significantly worse than SGLRW, for the logistic regression experiment considered here Clipped-SGLD emerges as a strong baseline. In the small-batch limit, where we observe a complete failure of SGLD, Clipped-SGLD performs only slightly worse than SGLRW, and occasionally better as the batch size increases.

## 5.4. Sentiment Classification With LLM Features

Having considered two tasks with well-understood posterior distributions, we now turn to a more realistic problem where, in practice, model size constrains the minibatch size: language modelling using LLMs. Specifically, we evaluate SGLRW on a sentiment classification task using the IMDB dataset (Maas et al., 2011), following a setup similar to Harrison et al.. The dataset consists of 50,000 strongly polarised movie reviews, split evenly into training and test sets. To study the effect of data scale, we additionally consider subsampled training sets of varying sizes.

**Setup.** For each experiment, we extract fixed sequence embeddings from a pretrained OPT language model (Zhang et al., 2022) with 350M parameters by taking the final-layer

representation of the last token. These embeddings are held fixed, and Bayesian posterior sampling is performed over the parameters of a two-layer binary classification head. This isolates the behaviour of the sampling algorithms from learning the data representation.

Each method is run with 15 parallel chains for 10,000 iterations across 3 independent seeds, discarding the first 5,000 iterations of each chain as burn-in, under the decaying schedule $\delta_t = \delta_0(t/T_{\text{burnin}} + 0.1)^{-0.5}$. For each training-set size, we vary the minibatch size to probe batch-size sensitivity while keeping schedule parameters matched across methods.

Performance is evaluated on the held-out test set using classification accuracy, negative log-likelihood (NLL), and expected calibration error (ECE).

### 5.4.1. RESULTS

As highlighted previously, SGLRW has consistently been less sensitive to the choice of learning-rate schedule than standard SGLD; it can handle substantially larger step sizes while still maintaining stability across all training-set sizes and minibatch configurations. As such, we first compare the performance of SGLRW against standard SGLD and Clipped-SGLD with small initial step sizes. Following this, we explore the other end of the spectrum, where we compare the performance of SGLRW against Clipped-SGLD at step sizes for which standard SGLD is unstable.

**Comparison at small step sizes.** Figure 5 shows predictive accuracy and negative log-likelihood with respect to minibatch size for a large training-set size ($N = 25,000$) for a run with $\delta_0 = 7.5 \times 10^{-6}$. At small minibatch sizes, SGLD exhibits a degradation in both accuracy and NLL, whereas SGLRW remains stable across the sweep. As the minibatch size increases, the accuracy of SGLD improves, while differences in NLL persist at moderate batch sizes. Similar to SGLRW, Clipped-SGLD also outperforms SGLD in this regime, again highlighting the strength of the baseline.

**Clipped-SGLD versus SGLRW at larger step sizes.** We next compare Clipped-SGLD and SGLRW in regimes where standard SGLD is unstable, focussing on step sizes beyond the conservative regime. Across all batch sizes considered in Figure 4, SGLRW consistently outperforms Clipped-SGLD in terms of predictive quality, with this behaviour remaining robust across training-set sizes.

The relative advantage of SGLRW becomes most pronounced in small-to-moderate minibatch regimes. This sweet spot arises because at very small batches even SGLRW is dominated by stochastic-gradient noise, whereas at large batches the problem is tractable without the stability benefits of the lattice discretisation. While accuracy dif-

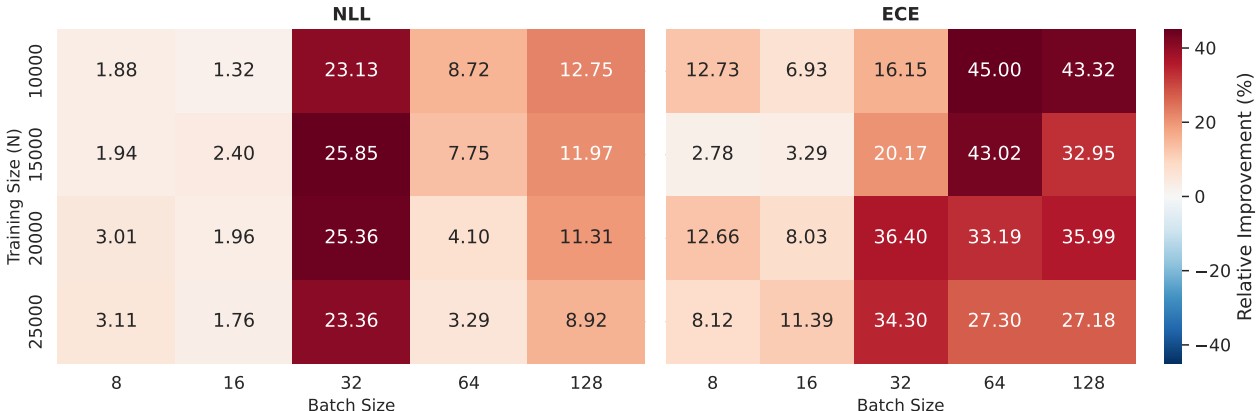

*Figure 4.* Relative improvement of SGLRW over Clipped-SGLD at increased learning-rate scale ($\delta_0 = 1.5 \times 10^{-4}$). Heatmaps show percentage differences in negative log-likelihood (**left**) and expected calibration error (**right**) across training-set sizes and minibatch sizes.

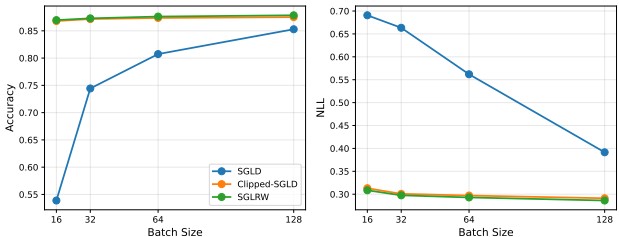

*Figure 5.* Predictive accuracy and negative log-likelihood (NLL) as a function of minibatch size for a large training-set size ($N = 25,000$), using base learning schedule with $\delta_0 = 7.5 \times 10^{-6}$.

ferences remain minor, SGLRW consistently attains lower negative log-likelihood and improved calibration relative to Clipped-SGLD. A representative comparison at an increased learning-rate scale is shown in Figure 4, with complete results across learning-rate schedules reported in Appendix A.12 (Figures 9, 10, and 11).

## 6. Conclusion

This work introduced Stochastic Gradient Lattice Random Walk (SGLRW), a robust discretisation of Langevin dynamics for Bayesian inference. By replacing Gaussian increments with coordinate-wise bounded updates, SGLRW significantly reduces sensitivity to minibatch size and stochastic gradient noise. Our theoretical analysis demonstrated that SGLRW achieves strictly tighter mean squared error (MSE) bounds than SGLD by confining minibatch-induced noise to the off-diagonal elements of the update covariance.

Empirically, SGLRW showed superior stability and predictive performance across diverse tasks, from linear regression to LLM-based sentiment classification, compared to both standard SGLD and a strong Clipped-SGLD baseline. Notably, it remains stable where SGLD diverges and

maintains high calibration even with small minibatches. Although SGLRW and Clipped-SGLD exhibit comparable per-coordinate clipping rates at small batch sizes (Section A.6), SGLRW achieves substantially lower KL throughout this regime, indicating that the robustness gain is attributable to the off-diagonal noise structure of Lemma 4.3 rather than to clipping alone.

Beyond its algorithmic advantages, the structure of SGLRW is uniquely suited for energy-efficient, low-precision, and stochastic hardware (Duffield et al., 2025a), an increasingly relevant constraint for sustainable AI (Aifer et al., 2025). The lattice update composes cleanly with cyclic schedules and diagonal preconditioning, and gives the practical heuristic of gradient clipping a Bayesian-inference analogue with the asymptotic guarantees inherited from SGLD.

Theorem 4.2 yields a vanishing MSE bound on ergodic averages under the decreasing-step-size schedule of Equation (9), inheriting SGLD's asymptotic guarantee (Chen et al., 2015; Teh et al., 2016); it does not characterise the invariant distribution of the discretised chain, so convergence in distribution of SGLRW remains open. The per-coordinate clipping bias of the practical implementation has activation probability vanishing with $\delta_t$ and falls under the standard SG-MCMC discretisation bias; an analytical characterisation would refine this. Following Duffield et al. (2025a) and Iguchi et al. (2026), the global Lipschitz assumption on $\nabla U$ could plausibly be relaxed to one-sided or local Lipschitz conditions. Extending the off-diagonal-only covariance argument of Lemma 4.3 to second-order SG-MCMC schemes such as SGHMC would broaden the scope of the analysis. The MSE bound also assumes finite-variance gradient noise; Figure 2 shows SGLRW remains stable under heavy-tailed ($\alpha$-stable) noise where SGLD diverges, and a theoretical analysis would close this gap.

## Acknowledgements

The authors thank the anonymous reviewers for their constructive feedback. LH is supported by the EPSRC Centre for Doctoral Training in Autonomous Intelligent Machines and Systems (EP/S024050/1). The research of MC and ZM is supported by the Vici grant (number VI.C.232.117) from the Dutch Research Council (NWO) and the Simons Collaboration on the Physics of Learning and Neural Computation (SFI-MPS-POL-00012574-17). Normal Computing thanks the Advanced Research and Invention Agency's (ARIA) Scaling Compute programme for partly funding this work.

## Impact Statement

This work introduces a posterior sampling method that is naturally compatible with stochastic hardware, and as such, SGLRW may reduce energy and memory requirements in practical implementations. Beyond this, this paper presents work whose goal is to advance the field of machine learning, which itself has many potential societal consequences.

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

# A. Appendix

## A.1. Proof of Theorem 4.2

In this appendix we prove Theorem 4.2. The argument follows the Poisson-equation framework of (Chen et al., 2015); we restate the required notation so the appendix is self-contained.

**Generator and Kolmogorov operators.** Consider a continuous-time Itô diffusion on $\mathbb{R}^d$ with infinitesimal generator

$$\mathcal{L}g(\theta) = f(\theta) \cdot \nabla g(\theta) + D(\theta) : \nabla^2 g(\theta), \tag{24}$$

with $f$ the drift of the SDE in (4). Let $(e^{t\mathcal{L}})_{t \geq 0}$ denote the associated Kolmogorov (backward) semigroup, so that for any suitable test function $g$,

$$\mathbb{E}[g(\theta_t) \mid \theta_0 = \theta] = (e^{t\mathcal{L}}g)(\theta). \tag{25}$$

Since $e^{t\mathcal{L}}$ is generally intractable, we consider a time-$\delta_t$ numerical update with one-step Markov operator $P_{\delta_t}^{(\mathcal{L})}$ defined by

$$\mathbb{E}[g(\theta_{n\delta_t}) \mid \theta_{(n-1)\delta_t}] = (P_{\delta_t}^{(\mathcal{L})}g)(\theta_{(n-1)\delta_t}). \tag{26}$$

A one-step scheme $P_{\delta_t}$ is a weak order-$K$ local integrator if, for all sufficiently smooth $g$, we have

$$(P_{\delta_t}g)(\theta) = (e^{\delta_t\mathcal{L}}g)(\theta) + O(\delta_t^{K+1}). \tag{27}$$

**Stochastic gradients and random one-step operators.** The Euler-Maruyama discretisation scheme is known to be weak order-one. We now examine the effect of using minibatches. In SGLD as well as SGLRW, the exact drift $f$ is replaced by a minibatch approximation $\widehat{f}$. Let $\widetilde{P}_{\delta_t}^{(n)}$ denote the resulting (random) one-step Markov operator at iteration $n$, i.e.

$$\mathbb{E}[g(\theta_{n\delta_t}) \mid \theta_{(n-1)\delta_t}] = (\widetilde{P}_{\delta_t}^{(n)}g)(\theta_{(n-1)\delta_t}). \tag{28}$$

For a given minibatch $\mathcal{B}$, define

$$\zeta(\theta; \mathcal{B}) := f(\theta) - \widehat{f}(\theta; \mathcal{B}), \tag{29}$$

and the associated first-order differential operator

$$(\Delta V_n g)(\theta; \mathcal{B}) := \zeta(\theta; \mathcal{B}_n) \cdot \nabla g(\theta), \tag{30}$$

capturing the error due to the minibatch drift error. Note that for overdamped Langevin diffusion, we have $f = -\nabla U$, $\widehat{f}(\theta; \mathcal{B}) = -\widehat{\nabla U}(\theta; \mathcal{B})$, so that $\zeta(\theta; \mathcal{B}) = \widehat{\nabla U}(\theta; \mathcal{B}) - \nabla U(\theta)$.

Beyond the first-order drift perturbation, minibatching can induce a second-order correction through the conditional second moment of the increment. We define the second-order differential operator

$$(\Delta A_n g)(\theta) := \frac{1}{2} M_n(\theta) : \nabla^2 g(\theta), \tag{31}$$

with

$$M_n(\theta, \mathcal{B}_n) := \delta_t^{-2} \mathbb{E}_{\xi_n}\big[\Delta\theta_n^{\mathrm{mb}}(\Delta\theta_n^{\mathrm{mb}})^\top - \Delta\theta_n^{\mathrm{fb}}(\Delta\theta_n^{\mathrm{fb}})^\top \mid \theta_{(n-1)\delta_t} = \theta, \mathcal{B}_n\big]. \tag{32}$$

Here $\Delta\theta_n^{\mathrm{mb}}$ and $\Delta\theta_n^{\mathrm{fb}}$ denote the one-step increments of the minibatch and full-batch updates, respectively, and $\mathbb{E}_{\xi_n}[\cdot]$ denotes expectation with respect to the internal randomness of the update at step $n$ (e.g. injected Gaussian noise or lattice path sampling).

**Poisson equation.** Given a smooth observable $\phi : \mathbb{R}^d \to \mathbb{R}$, define its stationary expectation under the invariant distribution $\pi$ by

$$\bar{\phi} := \int \phi(\theta)\, \pi(\theta)\, d\theta. \tag{33}$$

We analyse ergodic averages via the Poisson equation

$$\mathcal{L}\psi = \phi - \bar{\phi}, \tag{34}$$

and express finite-time errors in terms of the corresponding solution $\psi$. We assume that, for the Poisson solution $\psi$, the remainder satisfies the Lyapunov-weighted bound, in the sense that there exists a constant $p_0 > 0$ such that

$$|\mathcal{R}_n \psi(\theta)| \leq C\, \mathcal{V}(\theta)^{p_0}, \qquad \text{uniformly in } n \text{ and } \delta_t \in (0,1]. \tag{35}$$

**Lyapunov-Poisson regularity.**

**Assumption A.1.** Let $\psi$ solve the Poisson equation $\mathcal{L}\psi = \phi - \bar{\phi}$. Assume there exists a function $\mathcal{V} : \mathbb{R}^d \to [1, \infty)$ such that:

(i) (*Derivative control*) There exist constants $C_k, p_k > 0$ for $k = 0, 1, 2, 3, 4$ such that

$$\|D^k \psi(\theta)\| \leq C_k\, \mathcal{V}(\theta)^{p_k}. \tag{36}$$

(ii) (*Uniform moments along the chain*) There exist constants $p^*$ such that for all $p \leq p^*$,

$$\sup_n \mathbb{E}[\mathcal{V}(\theta_{n\delta_t})^p] < \infty. \tag{37}$$

(iii) (*Growth compatibility*) For the constant $p^*$ and some $C > 0$, for all $p \leq p^*$ and all $s \in (0,1)$,

$$\mathcal{V}^p(s\theta + (1-s)\vartheta) \leq C\big(\mathcal{V}^p(\theta) + \mathcal{V}^p(\vartheta)\big). \tag{38}$$

(iv) (*Uniform second-moment bound*) The second-order coefficient field satisfies

$$\sup_n \mathbb{E}\big[\|M_n(\theta_{(n-1)\delta_t})\|_F^2\big] < \infty. \tag{39}$$

(v) (*Increment moment control*) There exist constants $C > 0$ and exponents $q_2, q_4 > 0$ such that, for all $n$,

$$\mathbb{E}_{\xi_n, \mathcal{B}_n}\big[\|\Delta\theta_n\|^{2k} \,\big|\, \theta_{(n-1)\delta_t}\big] \leq C\, \delta_t^k\, \mathcal{V}(\theta_{(n-1)\delta_t})^{q_{2k}}, \qquad k \in \{1, 2\}, \tag{40}$$

where $\Delta\theta_n := \theta_{n\delta_t} - \theta_{(n-1)\delta_t}$.

(vi) (*Third-moment tensor control*) There exist constants $C > 0$ and an exponent $q_3 > 0$ such that, for all $n$,

$$\left\|\mathbb{E}_{\xi_n, \mathcal{B}_n}\big[(\Delta\theta_n^{\mathrm{mb}})^{\otimes 3} \,\big|\, \theta_{(n-1)\delta_t}\big] - \mathbb{E}_{\xi_n}\big[(\Delta\theta_n^{\mathrm{fb}})^{\otimes 3} \,\big|\, \theta_{(n-1)\delta_t}\big]\right\| \leq C\, \delta_t^2\, \mathcal{V}(\theta_{(n-1)\delta_t})^{q_3}. \tag{41}$$

(vii) (*Unbiased minibatch estimator*) The minibatch drift estimator is conditionally unbiased, in the sense that

$$\mathbb{E}_{\mathcal{B}_n}\big[\zeta(\theta_{(n-1)\delta_t}; \mathcal{B}_n) \,\big|\, \theta_{(n-1)\delta_t}\big] = 0 \qquad \text{a.s. for all } n, \tag{42}$$

where $\zeta(\theta; \mathcal{B}) := f(\theta) - \widehat{f}(\theta; \mathcal{B})$ denotes the minibatch drift error.

Note that the derivative bounds (36) in Assumption A.1 can be verified by constructing a Lyapunov function $\mathcal{V} : \mathbb{R}^d \to [1, \infty)$ which tends to infinity as $\theta \to \infty$, is twice continuously differentiable with bounded second derivatives, and satisfies the following conditions, as shown in the Appendix C of (Chen et al., 2015):

(a) (*Lyapunov drift condition*) There exist constants $\alpha, \beta > 0$ such that the exact drift field $f$ satisfies

$$\langle \nabla \mathcal{V}(\theta), f(\theta) \rangle \leq -\alpha \, \mathcal{V}(\theta) + \beta. \tag{43}$$

(b) (*Minibatch-induced drift fluctuations*) There exists $p_H \geq 2$ such that

$$\mathbb{E}_{\xi_n, \mathcal{B}_n} \big[ \| \zeta(\theta_{(n-1)\delta_t}; \mathcal{B}_n) \|^2 \mid \theta_{(n-1)\delta_t} = \theta \big] \leq C \, \mathcal{V}(\theta)^{p_H}, \tag{44}$$

together with the growth condition

$$\| \nabla \mathcal{V}(\theta) \|^2 + \| f(\theta) \|^2 \leq C \, \mathcal{V}(\theta). \tag{45}$$

*Remark* A.2. Throughout, we write $\theta_{(n-1)\delta_t} = \theta$ and recall that for overdamped Langevin diffusion we have $\zeta(\theta; \mathcal{B}) = \widehat{\nabla U}(\theta; \mathcal{B}) - \nabla U(\theta)$. Assumptions A.1(v)–(vi) hold for SGLD and SGLRW for varying requirements on the minibatch noise $\zeta$.

1. **SGLD.**

   - *Second moment (k = 1):* From $\Delta\theta_n^{\mathrm{mb}} = -\delta_t(\nabla U(\theta) + \zeta) + \sqrt{2\delta_t}\, \xi_n$, and using $\|a + b\|^2 \leq 2\|a\|^2 + 2\|b\|^2$ we have

   $$\mathbb{E}_{\xi_n, \mathcal{B}_n} \big[ \| \Delta\theta_n^{\mathrm{mb}} \|^2 \mid \theta \big] \leq 2\delta_t^2 \, \mathbb{E}_{\mathcal{B}_n} \big[ \| \nabla U(\theta) + \zeta \|^2 \mid \theta \big] + 4d \, \delta_t. \tag{46}$$

   Thus, assuming

   $$\mathbb{E}_{\mathcal{B}_n} \big[ \| \zeta \|^2 \mid \theta \big] \leq C \, \mathcal{V}(\theta)^{q_2}, \qquad \| \nabla U(\theta) \|^2 \leq C \, \mathcal{V}(\theta), \tag{47}$$

   yields

   $$\mathbb{E}_{\mathcal{B}_n, \xi_n} \big[ \| \Delta\theta_n^{\mathrm{mb}} \|^2 \mid \theta \big] = O(\delta_t). \tag{48}$$

   - *Fourth moment control (k = 2):* Similarly,

   $$\mathbb{E}_{\xi_n, \mathcal{B}_n} \big[ \| \Delta\theta_n^{\mathrm{mb}} \|^4 \mid \theta \big] \leq C\delta_t^4 \, \mathbb{E}_{\mathcal{B}_n} \big[ \| \nabla U(\theta) + \zeta \|^4 \mid \theta \big] + C\delta_t^2. \tag{49}$$

   Hence, if

   $$\mathbb{E}_{\mathcal{B}_n} \big[ \| \zeta \|^4 \mid \theta \big] \leq C \, \mathcal{V}(\theta)^{q_4}, \tag{50}$$

   then

   $$\mathbb{E}_{\xi_n, \mathcal{B}_n} \big[ \| \Delta\theta_n^{\mathrm{mb}} \|^4 \mid \theta \big] = O(\delta_t^2). \tag{51}$$

   - *Third-moment control:* From the expansion in (100), the difference in (41) arises from cubic drift noise interaction terms, and scales as $O(\delta_t^3)$. Assuming

   $$\mathbb{E}_{\mathcal{B}_n} \big[ \| \zeta \|^3 \mid \theta \big] \leq C \, \mathcal{V}(\theta)^{q_3}, \tag{52}$$

   this difference is bounded by $C \, \delta_t^2 \, \mathcal{V}(\theta)^{q_3}$ for $\delta_t \in (0, 1]$.

2. **SGLRW.**

   - *Increment control (k = 1, 2):* As $\| \Delta\theta_n^{\mathrm{mb}} \|^2 = 2d \, \delta_t$ and $\| \Delta\theta_n^{\mathrm{mb}} \|^4 = 4d^2 \, \delta_t^2$, the bound (40) holds without any assumption on $\zeta$.
   - *Third-moment control:* As shown in (114), the third-order conditional moments are batch-independent when at least two of the three directions coincide. For distinct indices $(i, j, k)$, the difference is $O(\delta_t^3)$. Consequently, (41) holds provided

   $$\mathbb{E}_{\mathcal{B}_n} \big[ \| \zeta \|^3 \mid \theta \big] \leq C \, \mathcal{V}(\theta)^{q_3}. \tag{53}$$

In SGLD as well as SGLRW, the exact drift $f$ is replaced by a minibatch approximation $\widehat{f}$. Since the underlying Euler-Maruyama scheme is a weak order-one integrator, we can derive a refined weak expansion for the resulting random operator, explicitly capturing the drift and covariance perturbations.

**Lemma A.3** (Refined weak expansion). *Consider a numerical integrator that is weak order-one for the full batch dynamics. Let $\widetilde{P}_{\delta_t}^{(n)}$ denote the random operator obtained by replacing the exact drift with a minibatch estimator, for which Assumption A.1 holds. Then $\widetilde{P}_{\delta_t}^{(n)}$ admits the expansion:*

$$\widetilde{P}_{\delta_t}^{(n)}\psi(\theta) = \psi(\theta) + \delta_t(\mathcal{L} - \Delta V_n)\psi(\theta) + \delta_t^2\,\Delta A_n\psi(\theta) + \delta_t^2\,\mathcal{R}_n\psi(\theta), \tag{54}$$

*where $\Delta V_n$ is the first-order drift perturbation, $\Delta A_n$ is the second-order covariance perturbation, and $|\mathcal{R}_n\psi(\theta)| \leq C\,\mathcal{V}(\theta)^{p_0}$ uniformly in $n$ and $\delta_t \in (0, 1]$.*

*Proof.* By Taylor's theorem with integral remainder applied to the minibatch increment,

$$\psi(\theta + \Delta\theta_n^{\mathrm{mb}}) = \psi(\theta) + \nabla\psi(\theta) \cdot \Delta\theta_n^{\mathrm{mb}} + \frac{1}{2}\Delta\theta_n^{\mathrm{mb}}(\Delta\theta_n^{\mathrm{mb}})^\top : \nabla^2\psi(\theta)$$
$$+ \frac{1}{6}\nabla^3\psi(\theta) : (\Delta\theta_n^{\mathrm{mb}})^{\otimes 3} + R_4(\theta, \Delta\theta_n^{\mathrm{mb}}), \tag{55}$$

where

$$R_4(\theta, \Delta\theta) = \frac{1}{6}\int_0^1 (1 - s)^3 \nabla^4\psi(\theta + s\Delta\theta) : \Delta\theta^{\otimes 4}\,ds. \tag{56}$$

For a given minibatch $\mathcal{B}_n$, taking the conditional expectation yields

$$\widetilde{P}_{\delta_t}^{(n)}\psi(\theta) = \psi(\theta) + \nabla\psi(\theta) \cdot \mathbb{E}_{\xi_n}[\Delta\theta_n^{\mathrm{mb}} \mid \theta] + \frac{1}{2}\mathbb{E}_{\xi_n}[\Delta\theta_n^{\mathrm{mb}}(\Delta\theta_n^{\mathrm{mb}})^\top \mid \theta] : \nabla^2\psi(\theta)$$
$$+ \frac{1}{6}\nabla^3\psi(\theta) : \mathbb{E}_{\xi_n}[(\Delta\theta_n^{\mathrm{mb}})^{\otimes 3} \mid \theta] + \mathbb{E}_{\xi_n}[R_4(\theta, \Delta\theta_n^{\mathrm{mb}}) \mid \theta]. \tag{57}$$

By definition of the increment-based drift difference,

$$\zeta(\theta; \mathcal{B}_n) = f(\theta) - \widehat{f}(\theta; \mathcal{B}), \tag{58}$$

so

$$\nabla\psi(\theta) \cdot \mathbb{E}_{\xi_n}[\Delta\theta_n^{\mathrm{mb}} \mid \theta] = \delta_t\,\nabla\psi(\theta) \cdot \widehat{f}(\theta; \mathcal{B}) = \delta_t\,f(\theta) \cdot \nabla\psi(\theta) - \delta_t\,\Delta V_n\psi(\theta). \tag{59}$$

Similarly, by definition of $\Delta A_n$,

$$\frac{1}{2}\mathbb{E}_{\xi_n}[\Delta\theta_n^{\mathrm{mb}}(\Delta\theta_n^{\mathrm{mb}})^\top \mid \theta] : \nabla^2\psi(\theta) = \frac{1}{2}\mathbb{E}_{\xi_n}[\Delta\theta_n^{\mathrm{fb}}(\Delta\theta_n^{\mathrm{fb}})^\top \mid \theta] : \nabla^2\psi(\theta) + \delta_t^2\,\Delta A_n\psi(\theta). \tag{60}$$

We now treat the cubic term by adding and subtracting the full-batch third moment:

$$\frac{1}{6}\nabla^3\psi(\theta) : \mathbb{E}_{\xi_n}[(\Delta\theta_n^{\mathrm{mb}})^{\otimes 3} \mid \theta] = \frac{1}{6}\nabla^3\psi(\theta) : \mathbb{E}_{\xi_n}[(\Delta\theta_n^{\mathrm{fb}})^{\otimes 3} \mid \theta]$$
$$+ \frac{1}{6}\nabla^3\psi(\theta) : \left(\mathbb{E}_{\xi_n}[(\Delta\theta_n^{\mathrm{mb}})^{\otimes 3} \mid \theta] - \mathbb{E}_{\xi_n}[(\Delta\theta_n^{\mathrm{fb}})^{\otimes 3} \mid \theta]\right). \tag{61}$$

Since the full-batch scheme is weak order one, its contribution is absorbed into the $O(\delta_t^2)$ remainder of the full-batch expansion. By Assumption A.1(vi), the difference of third-order moments is $O(\delta_t^2\,\mathcal{V}(\theta)^{q_3})$, so the net cubic contribution is $O(\delta_t^2\,\mathcal{V}(\theta)^{p_0})$.

For the fourth-order remainder, using $|T : x^{\otimes 4}| \leq \|T\|\,\|x\|^4$ and Assumption A.1(i) with $k = 4$, we obtain

$$|R_4(\theta, \Delta\theta_n^{\mathrm{mb}})| \leq C\,\|\Delta\theta_n^{\mathrm{mb}}\|^4 \int_0^1 \mathcal{V}(\theta + s\Delta\theta_n^{\mathrm{mb}})^{p_4}\,ds. \tag{62}$$

By Assumption A.1(iii), $\mathcal{V}(\theta + s\Delta\theta_n^{\text{mb}})^{p_4} \leq C(\mathcal{V}(\theta)^{p_4} + \mathcal{V}(\theta + \Delta\theta_n^{\text{mb}})^{p_4})$ for $s \in (0,1)$. Taking conditional expectation, applying the increment fourth-moment bound in Assumption A.1(v), and using the uniform moment bound in Assumption A.1(ii), yields

$$\mathbb{E}_{\xi_n}[|R_4(\theta, \Delta\theta_n^{\text{mb}})| \mid \theta, \mathcal{B}_n] \leq C\,\delta_t^2\,\mathcal{V}(\theta)^{p_0}, \tag{63}$$

uniformly in $n$ and $\delta_t \in (0,1]$.

Collecting all $O(\delta_t^2)$ contributions into $\mathcal{R}_n$ yields the stated expansion. $\qquad\square$

**Theorem A.4.** *Under Assumption A.1, there exists $C > 0$, independent of $(L, \delta_t)$, such that*

$$\mathbb{E}(\hat{\phi} - \bar{\phi})^2 \leq C\left(\frac{1}{L^2}\sum_{n=1}^{L}\mathbb{E}\big[\|\zeta_n\|^2\big] + \frac{1}{L\delta_t} + \frac{1}{L^2\delta_t^2} + \delta_t^2 + \frac{\delta_t^2}{L}\sum_{n=1}^{L}\mathbb{E}\big[\|M_n\|^2\big]\right). \tag{64}$$

*Proof.* First write $\hat{\phi} - \bar{\phi} = \frac{1}{L}\sum_{n=1}^{L}\mathcal{L}\psi(\theta_{n\delta_t})$. Next, use Lemma A.3 to rewrite $\mathcal{L}\psi(\theta)$, and using $\mathbb{E}_{\xi_n}[\psi(\theta_{n\delta_t}) \mid \theta_{(n-1)\delta_t}, \mathcal{B}_n] = (\widetilde{P}_{\delta_t}^{(n)}\psi)(\theta_{(n-1)\delta_t})$, we obtain

$$\hat{\phi} - \bar{\phi} = \frac{1}{L\delta_t}\Big(\psi(\theta_{L\delta_t}) - \psi(\theta_0)\Big) - \frac{1}{L\delta_t}\sum_{n=1}^{L}\Big(\mathbb{E}_{\xi_n}[\psi(\theta_{n\delta_t}) \mid \theta_{(n-1)\delta_t}] - \psi(\theta_{n\delta_t})\Big)$$

$$+ \frac{1}{L}\sum_{n=1}^{L}\Delta V_n\psi(\theta_{(n-1)\delta_t}) - \frac{\delta_t}{L}\sum_{n=1}^{L}\Delta A_n\psi(\theta_{(n-1)\delta_t}) - \frac{\delta_t}{L}\sum_{n=1}^{L}\mathcal{R}_n\psi(\theta_{(n-1)\delta_t}). \tag{65}$$

Taking squares and using $(a + b + c + d + e)^2 \leq 5(a^2 + b^2 + c^2 + d^2 + e^2)$ gives

$$\mathbb{E}(\hat{\phi} - \bar{\phi})^2 \leq C\,(\alpha_1 + \alpha_2 + \alpha_3 + \alpha_4 + \alpha_5), \tag{66}$$

where

$$\alpha_1 := \mathbb{E}\left[\left(\frac{\psi(\theta_{L\delta_t}) - \psi(\theta_0)}{L\delta_t}\right)^2\right], \tag{67}$$

$$\alpha_2 := \mathbb{E}\left[\frac{1}{L^2\delta_t^2}\left(\sum_{n=1}^{L}\big(\mathbb{E}_{\xi_n}[\psi(\theta_{n\delta_t}) \mid \theta_{(n-1)\delta_t}] - \psi(\theta_{n\delta_t})\big)\right)^2\right], \tag{68}$$

$$\alpha_3 := \mathbb{E}\left[\frac{1}{L^2}\left(\sum_{n=1}^{L}\Delta V_n\psi(\theta_{(n-1)\delta_t})\right)^2\right], \tag{69}$$

$$\alpha_4 := \mathbb{E}\left[\left(\frac{\delta_t}{L}\sum_{n=1}^{L}\Delta A_n\psi(\theta_{(n-1)\delta_t})\right)^2\right], \tag{70}$$

$$\alpha_5 := \mathbb{E}\left[\left(\frac{\delta_t}{L}\sum_{n=1}^{L}\mathcal{R}_n\psi(\theta_{(n-1)\delta_t})\right)^2\right]. \tag{71}$$

**(i) $\alpha_1$.** By Lyapunov control of $\psi$ (Assumption A.1(i–ii) with $k = 0$), $\sup_n \mathbb{E}[\psi(\theta_{n\delta_t})^2] < \infty$, hence

$$\alpha_1 \leq \frac{C}{L^2\delta_t^2}. \tag{72}$$

**(ii) $\alpha_2$.** Let

$$Z_n := \mathbb{E}_{\xi_n}[\psi(\theta_{n\delta_t}) \mid \theta_{(n-1)\delta_t}, \mathcal{B}_n] - \psi(\theta_{n\delta_t}), \tag{73}$$

so that $\{Z_n\}_{n \geq 1}$ is a martingale difference sequence and hence

$$\alpha_2 = \frac{1}{L^2 \delta_t^2} \sum_{n=1}^{L} \mathbb{E}\big[\mathrm{Var}(\psi(\theta_{n\delta_t}) \mid \theta_{(n-1)\delta_t}, \mathcal{B}_n)\big] \leq \frac{1}{L^2 \delta_t^2} \sum_{n=1}^{L} \mathbb{E}\big[\mathrm{Var}(\psi(\theta_{n\delta_t}) \mid \theta_{(n-1)\delta_t})\big], \tag{74}$$

where the inequality is the law of total variance applied at the minibatch $\mathcal{B}_n$, so the marginal variance gives a valid upper bound that we bound below.

Write $\theta := \theta_{(n-1)\delta_t}$ and $\Delta\theta_n := \theta_{n\delta_t} - \theta$. By the fundamental theorem of calculus,

$$\psi(\theta + \Delta\theta_n) - \psi(\theta) = \Delta\theta_n \cdot \int_0^1 \nabla\psi(\theta + s\Delta\theta_n)\, ds. \tag{75}$$

Therefore,

$$\mathrm{Var}(\psi(\theta_{n\delta_t}) \mid \theta) = \mathrm{Var}(\psi(\theta_{n\delta_t}) - \psi(\theta) \mid \theta) \leq \mathbb{E}\big[(\psi(\theta + \Delta\theta_n) - \psi(\theta))^2 \mid \theta\big]$$
$$\leq \mathbb{E}\left[\|\Delta\theta_n\|^2 \int_0^1 \|\nabla\psi(\theta + s\Delta\theta_n)\|^2\, ds \,\middle|\, \theta\right], \tag{76}$$

where we used Cauchy-Schwarz.

By Assumption A.1(i), $\|\nabla\psi(x)\|^2 \leq C\,\mathcal{V}(x)^{2p_1}$, and by Assumption A.1(iii),

$$\mathcal{V}(\theta + s\Delta\theta_n)^{2p_1} \leq C\big(\mathcal{V}(\theta)^{2p_1} + \mathcal{V}(\theta + \Delta\theta_n)^{2p_1}\big), \qquad s \in (0,1). \tag{77}$$

Hence,

$$\mathrm{Var}(\psi(\theta_{n\delta_t}) \mid \theta) \leq C\,\mathbb{E}\Big[\|\Delta\theta_n\|^2\big(\mathcal{V}(\theta)^{2p_1} + \mathcal{V}(\theta_{n\delta_t})^{2p_1}\big) \mid \theta\Big]. \tag{78}$$

Taking conditional expectation with respect to $\mathcal{B}_n$ and using the tower property yields

$$\mathbb{E}[\mathrm{Var}(\psi(\theta_{n\delta_t}) \mid \theta) \mid \theta] \leq C\,\mathbb{E}\Big[\|\Delta\theta_n\|^2\big(\mathcal{V}(\theta)^{2p_1} + \mathcal{V}(\theta_{n\delta_t})^{2p_1}\big) \mid \theta\Big]. \tag{79}$$

By Assumption A.1(v), the increment satisfies the conditional second-moment bound

$$\mathbb{E}\big[\|\Delta\theta_n\|^2 \mid \theta_{(n-1)\delta_t}\big] \leq C\,\delta_t\,\mathcal{V}(\theta_{(n-1)\delta_t})^q, \tag{80}$$

for some $q > 0$. Taking total expectations and using Assumption A.1(ii) to control moments of $\mathcal{V}(\theta_{n\delta_t})$ yields

$$\mathbb{E}\big[\mathrm{Var}(\psi(\theta_{n\delta_t}) \mid \theta_{(n-1)\delta_t}, \mathcal{B}_n)\big] \leq C\,\delta_t. \tag{81}$$

Substituting this bound into the definition of $\alpha_2$ gives

$$\alpha_2 \leq \frac{1}{L^2 \delta_t^2} \sum_{n=1}^{L} C\,\delta_t = \frac{C}{L\delta_t}. \tag{82}$$

**(iii)** $\alpha_3$. Set $X_n := \Delta V_n \psi(\theta_{(n-1)\delta_t})$. By the assumed unbiasedness we have $\mathbb{E}[X_n \mid \theta_{(n-1)\delta_t}] = 0$, so $\{X_n\}$ is a martingale difference sequence and therefore cross-terms vanish:

$$\mathbb{E}\left[\Big(\sum_{n=1}^{L} X_n\Big)^2\right] = \sum_{n=1}^{L} \mathbb{E}[X_n^2]. \tag{83}$$

Moreover,

$$|X_n| = |\zeta(\theta_{(n-1)\delta_t}; \mathcal{B}_n) \cdot \nabla\psi(\theta_{(n-1)\delta_t})| \leq \|\zeta(\theta_{(n-1)\delta_t}; \mathcal{B}_n)\|\,\|\nabla\psi(\theta_{(n-1)\delta_t})\|, \tag{84}$$

so by Assumption A.1(i-ii),

$$\mathbb{E}[X_n^2] \leq C\,\mathbb{E}\big[\|\zeta(\theta_{(n-1)\delta_t};\mathcal{B}_n)\|^2\big]\,,\tag{85}$$

absorbing $\|\nabla\psi\|^2$ into the constant using the Lyapunov moment bounds. Concretely, $\|\nabla\psi(\theta)\|^2 \leq C\,\mathcal{V}(\theta)^{2p_1}$ by Assumption A.1(i) with $k=1$, and the cross-term $\mathbb{E}[\|\zeta\|^2\mathcal{V}^{2p_1}]$ is controlled by Cauchy–Schwarz together with the moment bounds $\mathbb{E}_{\mathcal{B}_n}[\|\zeta\|^4 \mid \theta] \leq C\mathcal{V}(\theta)^{q_4}$ (Assumption A.1(v) with $k=2$) and $\sup_n \mathbb{E}[\mathcal{V}(\theta_{(n-1)\delta_t})^p] < \infty$ for $p$ large enough (Assumption A.1(ii)), giving a uniform-in-$n$ constant absorbing the $\mathcal{V}$-dependence. Hence

$$\alpha_3 = \frac{1}{L^2}\sum_{n=1}^{L}\mathbb{E}[X_n^2] \leq \frac{C}{L^2}\sum_{n=1}^{L}\mathbb{E}\big[\|\zeta(\theta_{(n-1)\delta_t};\mathcal{B}_n)\|^2\big]\,.\tag{86}$$

**(iv) $\alpha_4$.** By Cauchy-Schwarz,

$$\alpha_4 \leq \frac{\delta_t^2}{L}\sum_{n=1}^{L}\mathbb{E}\big[(\Delta A_n\psi(\theta_{(n-1)\delta_t}))^2\big]\,.\tag{87}$$

Using $\Delta A_n f = \frac{1}{2}M_n : \nabla^2 f$ and the Hilbert-Schmidt inequality,

$$|\Delta A_n\psi(\theta)| \leq \frac{1}{2}\|M_n(\theta)\|_F\,\|\nabla^2\psi(\theta)\|_F \leq C\,\|M_n(\theta)\|\,\mathcal{V}(\theta)^{p_2},\tag{88}$$

where we used Assumption A.1(i) with $k=2$. Taking expectations and using Assumptions A.1(ii, iv), we obtain

$$\mathbb{E}\big[(\Delta A_n\psi(\theta_{(n-1)\delta_t}))^2\big] \leq C\,\mathbb{E}\big[\|M_n(\theta_{(n-1)\delta_t};\mathcal{B}_n)\|^2\big]\,.\tag{89}$$

Here the $\mathcal{V}^{2p_2}$ factor from $\|\nabla^2\psi\|^2$ is absorbed into $C$ by Cauchy–Schwarz with the second-moment bound on $\|M_n\|^2$ (Assumption A.1(iv)) and the uniform-in-$n$ Lyapunov moment bound (Assumption A.1(ii)). Therefore

$$\alpha_4 \leq \frac{C\,\delta_t^2}{L}\sum_{n=1}^{L}\mathbb{E}\big[\|M_n(\theta_{(n-1)\delta_t};\mathcal{B}_n)\|^2\big]\,.\tag{90}$$

**(v) $\alpha_5$.** By Cauchy-Schwarz and the remainder bound from Lemma A.3,

$$\alpha_5 \leq \frac{\delta_t^2}{L}\sum_{n=1}^{L}\mathbb{E}\big[(\mathcal{R}_n\psi(\theta_{(n-1)\delta_t}))^2\big] \leq \frac{C\,\delta_t^2}{L}\sum_{n=1}^{L}\mathbb{E}\big[\mathcal{V}(\theta_{(n-1)\delta_t})^{2p_0}\big] \leq C\,\delta_t^2,\tag{91}$$

using Assumption A.1(ii).

Combining the bounds on $\alpha_1,\ldots,\alpha_5$ yields (64). $\qquad\square$

## A.2. Conditional Covariance and Moments of SGLD and SGLRW Updates

We decompose the stochastic gradient as

$$\widehat{\nabla U}(\theta;\mathcal{B}) = \nabla U(\theta) + \zeta(\theta;\mathcal{B}), \qquad \mathbb{E}_{\mathcal{B}}[\zeta(\theta;\mathcal{B}) \mid \theta] = 0, \qquad \mathrm{Cov}_{\mathcal{B}}[\zeta(\theta;\mathcal{B}) \mid \theta] = G(\theta),\tag{92}$$

where $G(\theta)$ quantifies the minibatch-induced gradient covariance.

Recall the definition of the second-order minibatch contribution $M_n$ used in Theorem A.4:

$$M_n(\theta,\mathcal{B}_n) := \delta_t^{-2}\,\mathbb{E}_{\xi_n}\big[\Delta\theta_n^{\mathrm{mb}}(\Delta\theta_n^{\mathrm{mb}})^{\top} - \Delta\theta_n^{\mathrm{fb}}(\Delta\theta_n^{\mathrm{fb}})^{\top} \mid \theta_{(n-1)\delta_t} = \theta, \mathcal{B}_n\big]\,,\tag{93}$$

where $\xi_n$ denotes the internal randomness of the integrator.

**SGLD.** The standard SGLD update is $\Delta\theta_t = -\delta_t(\nabla U(\theta_t) + \zeta_t) + \sqrt{2\delta_t}\xi_t$ with $\xi_t \sim \mathcal{N}(0, I)$. Using the independence of $\xi_t$ and $\zeta_t$, and $\mathbb{E}[\zeta_t \mid \theta_t] = 0$, the first and second moments are:

$$\mathbb{E}[\Delta\theta_t \mid \theta_t] = -\delta_t \nabla U(\theta_t), \tag{94}$$

$$\mathbb{E}[\Delta\theta_t \Delta\theta_t^\top \mid \theta_t] = 2\delta_t I + \delta_t^2 \left( \nabla U(\theta_t) \nabla U(\theta_t)^\top + G(\theta_t) \right). \tag{95}$$

We next derive the second-order minibatch contribution $M_n$. Fix $\theta$ and a minibatch $\mathcal{B}$, define the full-batch and minibatch increments (sharing the same Gaussian $\xi$)

$$\Delta\theta^{\mathrm{fb}} = -\delta_t \nabla U(\theta) + \sqrt{2\delta_t}\,\xi, \qquad \Delta\theta^{\mathrm{mb}} = -\delta_t(\nabla U(\theta) + \zeta) + \sqrt{2\delta_t}\,\xi = \Delta\theta^{\mathrm{fb}} - \delta_t \zeta. \tag{96}$$

Expanding the outer products gives

$$\Delta\theta^{\mathrm{mb}}\Delta\theta^{\mathrm{mb}\top} - \Delta\theta^{\mathrm{fb}}\Delta\theta^{\mathrm{fb}\top} = -\delta_t\,\Delta\theta^{\mathrm{fb}}\zeta^\top - \delta_t\,\zeta(\Delta\theta^{\mathrm{fb}})^\top + \delta_t^2\,\zeta\zeta^\top. \tag{97}$$

Taking conditional expectation over the internal randomness $\xi$ and using $\mathbb{E}_\xi[\Delta\theta^{\mathrm{fb}} \mid \theta] = -\delta_t \nabla U(\theta)$ yields

$$\mathbb{E}_\xi\left[\Delta\theta^{\mathrm{mb}}\Delta\theta^{\mathrm{mb}\top} - \Delta\theta^{\mathrm{fb}}\Delta\theta^{\mathrm{fb}\top} \mid \theta, \mathcal{B}\right] = \delta_t^2\left(\zeta\zeta^\top + \nabla U(\theta)\zeta^\top + \zeta \nabla U(\theta)^\top\right). \tag{98}$$

Therefore, with $M_n$ defined as in (31),

$$M_{n,\mathrm{SGLD}}(\theta; \mathcal{B}) = \zeta(\theta; \mathcal{B})\zeta(\theta; \mathcal{B})^\top + \nabla U(\theta)\,\zeta(\theta; \mathcal{B})^\top + \zeta(\theta; \mathcal{B})\,\nabla U(\theta)^\top. \tag{99}$$

Averaging additionally over minibatches gives $\mathbb{E}_\mathcal{B}[M_{n,\mathrm{SGLD}}(\theta, \mathcal{B}) \mid \theta] = G(\theta)$.

For the third moment, let $u_t = -\delta_t(\nabla U + \zeta_t)$ and $w_t = \sqrt{2\delta_t}\xi_t$. Expanding $\mathbb{E}[(u_t + w_t)^{\otimes 3} \mid \theta_t]$, terms with odd powers of $\xi_t$ vanish. The remaining terms are $\mathbb{E}[u_t^{\otimes 3}]$ and the cross-terms $\mathbb{E}[u_t \otimes w_t \otimes w_t]$ (and permutations). The leading $O(\delta_t^2)$ error comes from the cross-terms:

$$\mathbb{E}[u_{t,i}w_{t,j}w_{t,k} \mid \theta_t] = \mathbb{E}[-\delta_t(\partial_i U + \zeta_i) \cdot 2\delta_t \delta_{jk}] = -2\delta_t^2 \partial_i U \delta_{jk}.$$

Because $\mathbb{E}[\zeta_i] = 0$, the $O(\delta_t^2)$ part of the tensor is identical for full-batch and minibatch schemes. The third-order moments of the noise appear in the $\mathbb{E}[u_t^{\otimes 3}]$ expansion. By expanding the cubic terms and using $\mathbb{E}[\zeta_t] = 0$, the tensor entries decompose as follows:

$$\mathbb{E}[\Delta\theta_i\,\Delta\theta_j\,\Delta\theta_k \mid \theta_t] = \begin{cases} -6\delta_t^2\,\partial_i U - \delta_t^3\left((\partial_i U)^3 + 3\,\partial_i U\,G_{ii} + \mathbb{E}[\zeta_i^3 \mid \theta_t]\right), & i = j = k, \\[2mm] -2\delta_t^2\,\partial_k U - \delta_t^3\left((\partial_i U)^2\,\partial_k U + \partial_k U\,G_{ii} + 2\,\partial_i U\,G_{ik} + \mathbb{E}[\zeta_i^2\zeta_k \mid \theta_t]\right), & i = j \neq k, \\[2mm] -\delta_t^3\left(\partial_i U\,\partial_j U\,\partial_k U + \sum_{\mathrm{cyc}(i,j,k)}\partial_i U\,G_{jk} + \mathbb{E}[\zeta_i\zeta_j\zeta_k \mid \theta_t]\right), & i, j, k \text{ all distinct,} \end{cases} \tag{100}$$

and permutations.

**SGLRW.** For the lattice random walk (LRW) discretisation, each coordinate $i$ takes a binary step $\Delta\theta_{t,i} \in \{\pm\sqrt{2\delta_t}\}$ with probabilities

$$\mathbb{P}\left[\Delta\theta_{t,i} = \sqrt{2\delta_t} \,\middle|\, \theta_t, \zeta_t\right] = \tfrac{1}{2} - \tfrac{1}{2}\sqrt{\tfrac{\delta_t}{2}}\,[\partial_i U(\theta_t) + \zeta_{t,i}], \tag{101}$$

$$\mathbb{P}\left[\Delta\theta_{t,i} = -\sqrt{2\delta_t} \,\middle|\, \theta_t, \zeta_t\right] = \tfrac{1}{2} + \tfrac{1}{2}\sqrt{\tfrac{\delta_t}{2}}\,[\partial_i U(\theta_t) + \zeta_{t,i}]. \tag{102}$$

Conditionally on $(\theta_t, \zeta_t)$ the coordinates are independent. A short calculation gives

$$\mathbb{E}[\Delta\theta_t \mid \theta_t, \zeta_t] = -\delta_t(\nabla U(\theta_t) + \zeta_t), \tag{103}$$

$$\mathbb{E}[\Delta\theta_t^2 \mid \theta_t, \zeta_t] = 2\delta_t I. \tag{104}$$

Averaging over $\zeta_t$ yields $\mathbb{E}[\Delta\theta_t \mid \theta_t] = -\delta_t \nabla U(\theta_t)$ and $\mathbb{E}[\Delta\theta_{t,i}^2 \mid \theta_t] = 2\delta_t$. For off-diagonal elements ($i \neq j$),

$$\mathbb{E}[\Delta\theta_{t,i}\Delta\theta_{t,j} \mid \theta_t] = \mathbb{E}[\mathbb{E}[\Delta\theta_{t,i} \mid \zeta_t]\mathbb{E}[\Delta\theta_{t,j} \mid \zeta_t] \mid \theta_t] = \delta_t^2[\partial_i U(\theta_t)\partial_j U(\theta_t) + G_{ij}(\theta_t)], \tag{105}$$

hence we have

$$\mathbb{E}[\Delta\theta_t\Delta\theta_t^\top \mid \theta_t] = 2\delta_t I + \delta_t^2 \operatorname{offdiag}\left(\nabla U(\theta_t)\nabla U(\theta_t)^\top + G(\theta_t)\right). \tag{106}$$

We next compute the second-order minibatch contribution $M_n$. Fix $\theta$ and a minibatch $\mathcal{B}$. For SGLRW, since $\Delta\theta_i^2 \equiv 2\delta_t$ deterministically, the diagonal entries of $\mathbb{E}_\xi[\Delta\theta\Delta\theta^\top \mid \theta, \mathcal{B}]$ coincide with their full-batch counterparts, and therefore

$$\left(M_{n,\text{SGLRW}}(\theta, \mathcal{B})\right)_{ii} = 0, \qquad i = 1, \ldots, d. \tag{107}$$

For $i \neq j$, conditional independence (given $(\theta, \mathcal{B})$) implies

$$\begin{aligned}
\mathbb{E}_\xi[\Delta\theta_i\Delta\theta_j \mid \theta, \mathcal{B}] &= \mathbb{E}_\xi[\Delta\theta_i \mid \theta, \mathcal{B}]\,\mathbb{E}_\xi[\Delta\theta_j \mid \theta, \mathcal{B}] \\
&= \delta_t^2\left(\partial_i U(\theta) + \zeta_i\right)\left(\partial_j U(\theta) + \zeta_j\right),
\end{aligned} \tag{108}$$

where we used (103). The corresponding full-batch term is $\mathbb{E}_\xi[\Delta\theta_i^{\text{fb}}\Delta\theta_j^{\text{fb}} \mid \theta] = \delta_t^2\,\partial_i U(\theta)\partial_j U(\theta)$ for $i \neq j$. Subtracting and rescaling therefore gives, for $i \neq j$,

$$\left(M_{n,\text{SGLRW}}(\theta; \mathcal{B})\right)_{ij} = \partial_i U(\theta)\,\zeta_j + \zeta_i\,\partial_j U(\theta) + \zeta_i\zeta_j. \tag{109}$$

Equivalently,

$$M_{n,\text{SGLRW}}(\theta; \mathcal{B}) = \operatorname{offdiag}\left(\zeta\zeta^\top + \nabla U(\theta)\zeta^\top + \zeta\nabla U(\theta)^\top\right) = \operatorname{offdiag}\left(M_{n,\text{SGLD}}(\theta; \mathcal{B})\right). \tag{110}$$

Averaging additionally over minibatches gives $\mathbb{E}_\mathcal{B}[M_{n,\text{SGLRW}}(\theta, \mathcal{B}) \mid \theta] = \operatorname{offdiag}(G(\theta))$.

Since $\Delta\theta_{t,i} \in \{\pm\sqrt{2\delta_t}\}$, we have the identity $\Delta\theta_{t,i}^3 = 2\delta_t\,\Delta\theta_{t,i}$. Higher-order moments are computed via the law of iterated expectations. For the case where at least two indices are equal:

$$\mathbb{E}[\Delta\theta_{t,i}^3 \mid \theta_t] = 2\delta_t\mathbb{E}[\Delta\theta_{t,i} \mid \theta_t] = -2\delta_t^2\,\partial_i U(\theta_t), \tag{111}$$

$$\mathbb{E}[\Delta\theta_i^2\Delta\theta_k \mid \theta_t] = \mathbb{E}[2\delta_t \cdot \mathbb{E}[\Delta\theta_k \mid \theta_t, \zeta_t] \mid \theta_t] = \mathbb{E}[2\delta_t(-\delta_t(\partial_k U + \zeta_k))] = -2\delta_t^2\partial_k U(\theta_t). \tag{112}$$

For fully distinct indices, the coordinates are coupled by the noise $\zeta_t$:

$$\begin{aligned}
\mathbb{E}[\Delta\theta_i\Delta\theta_j\Delta\theta_k \mid \theta_t] &= \mathbb{E}[(-\delta_t(\partial_i U + \zeta_i))(-\delta_t(\partial_j U + \zeta_j))(-\delta_t(\partial_k U + \zeta_k)) \mid \theta_t] \\
&= -\delta_t^3\left(\partial_i U\partial_j U\partial_k U + \sum_{\text{cyc}(i,j,k)} \partial_i U G_{jk} + \mathbb{E}[\zeta_i\zeta_j\zeta_k \mid \theta_t]\right).
\end{aligned} \tag{113}$$

The resulting third-moment tensor entries are:

$$\mathbb{E}[\Delta\theta_{t,i}\Delta\theta_{t,j}\Delta\theta_{t,k} \mid \theta_t] = \begin{cases} -2\delta_t^2\,\partial_i U(\theta_t), & i = j = k, \\ -2\delta_t^2\,\partial_k U(\theta_t), & i = j \neq k, \\ -\delta_t^3\left(\partial_i U\partial_j U\partial_k U + \sum_{\text{cyc}(i,j,k)} \partial_i U G_{jk} + \mathbb{E}[\zeta_i\zeta_j\zeta_k \mid \theta_t]\right), & i, j, k \text{ all distinct,} \end{cases} \tag{114}$$

and permutations.

### A.3. Full-Increment Covariance Analysis.

We analyse *full-increment clipping* for (SG)LD and show that it yields an incorrect diffusion limit.

We consider the SGLD increment

$$\Delta\theta_t = -\delta_t \widehat{\nabla U}(\theta_t) + \sqrt{2\delta_t}\,\xi_t, \qquad \xi_t \sim \mathcal{N}(0, I), \tag{115}$$

and set the clipping radius $R_t = \sqrt{2\delta_t}$. Introduce the rescaled increment

$$Z_t := \frac{\Delta\theta_t}{\sqrt{2\delta_t}} = \xi_t - \sqrt{\frac{\delta_t}{2}} \, \widehat{\nabla U}(\theta_t). \tag{116}$$

Then we define the clipped increment by $\widetilde{\Delta\theta}_t := \sqrt{2\delta_t} \, \mathrm{clip}(Z_t; 1)$, where $\mathrm{clip}(x; R)_i = \mathrm{sign}(x_i) \min\{|x_i|, R\}$. Since clip is bounded and Lipschitz, and since $Z_t \to \xi$ in distribution (and in second moment) conditionally on $\theta_t$ whenever $\widehat{\nabla U}(\theta_t)$ has finite second moment, we obtain

$$\lim_{\delta_t \to 0} \frac{1}{\delta_t} \mathrm{Cov}\big[\widetilde{\Delta\theta}_t \mid \theta_t\big] = \lim_{\delta_t \to 0} 2 \, \mathrm{Cov}\big(\mathrm{clip}(Z_t; 1) \mid \theta_t\big) = 2 \, \mathrm{Cov}\big(\mathrm{clip}(\xi; 1)\big). \tag{117}$$

Thus the diffusion limit is determined by the covariance of $\mathrm{clip}(\xi; 1)$. By independence across coordinates, this yields

$$\mathrm{Cov}\big(\mathrm{clip}(\xi; 1)\big) = sI, \qquad s = \mathbb{E}[\min(\xi_1^2, 1)] = 1 - \sqrt{\frac{2}{\pi}} e^{-1/2} \approx 0.516, \tag{118}$$

which can be shown by explicitly computing $\mathbb{E}[\min(\xi_1^2, 1)]$. Therefore we have

$$\lim_{\delta_t \to 0} \frac{1}{\delta_t} \mathrm{Cov}\big[\widetilde{\Delta\theta}_t \mid \theta_t\big] = 2s \, I. \tag{119}$$

The limiting covariance equals $2sI$ with $s < 1$, whereas the Langevin diffusion requires covariance $2I$. Hence full-increment clipping yields an incorrect diffusion limit and breaks convergence to the target distribution even as $\delta_t \to 0$.

### A.4. Analysis plots including Clipped-SGLD

For completeness, we also include Clipped-SGLD in the plots for the analysis.

For the covariance MSE analysis (Figure 3), Clipped-SGLD already mitigates the sharp error explosion observed for SGLD at larger step sizes (Figure 6). However, across batch sizes and learning rates, SGLRW consistently attains comparable or lower covariance MSE throughout the stable regime.

For the heavy-tailed noise robustness analysis (Figure 2), both Clipped-SGLD and SGLRW prevent the severe instability exhibited by SGLD under heavy-tailed gradient noise. Across noise scales, SGLRW maintains a closer qualitative agreement with the target distribution than Clipped-SGLD.

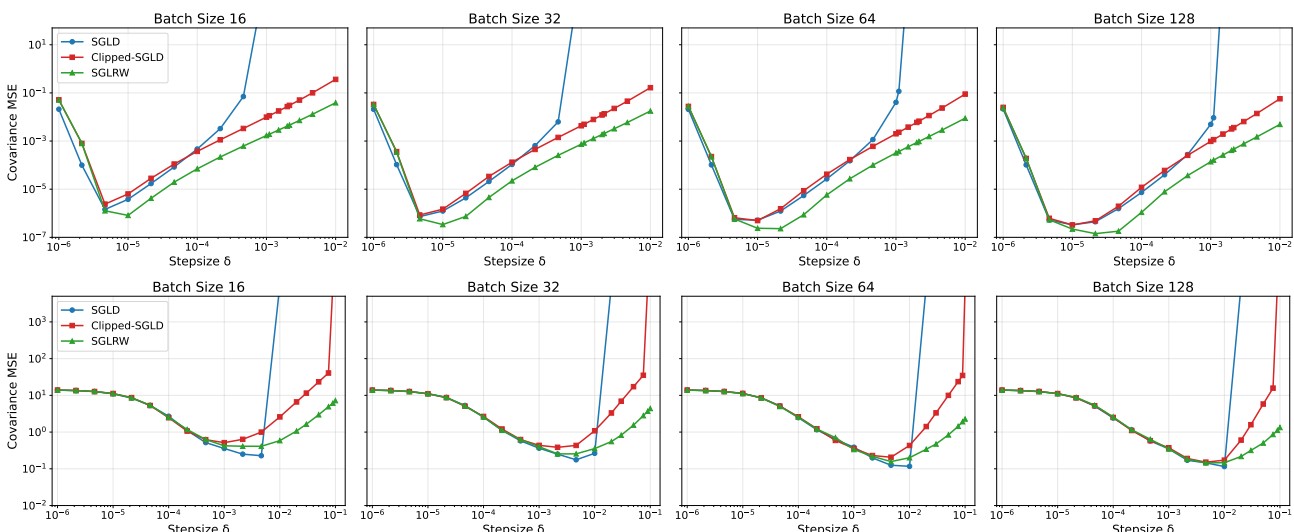

*Figure 6.* Mean-squared error (MSE) of the posterior covariance as a function of the step size $\delta_t$, shown for different batch sizes. **Top:** 50-dimensional Bayesian linear regression. **Bottom:** Bayesian logistic regression on the breast cancer dataset (Wolberg et al., 1993).

*Table 5.* Kullback–Leibler (KL) divergence on Bayesian linear regression, reported as mean $\pm$ standard deviation over 12 independent seeds. **Bold** indicates the lowest mean KL for a given $(B, \delta_0)$ pair, together with any method whose $\pm 1\sigma$ interval overlaps the minimum.

| Hyperparameters | | KL Divergence | | |
|---|---|---|---|---|
| $B$ | $\delta_0$ | SGLD | SGLRW | Clipped-SGLD |
| 8 | $10^{-3}$ | $19.853 \pm 0.337$ | $\mathbf{6.060 \pm 0.137}$ | $18.496 \pm 0.269$ |
| 8 | $10^{-4}$ | $0.504 \pm 0.019$ | $\mathbf{0.196 \pm 0.010}$ | $0.822 \pm 0.037$ |
| 16 | $10^{-3}$ | $7.143 \pm 0.132$ | $\mathbf{2.328 \pm 0.066}$ | $8.661 \pm 0.135$ |
| 16 | $10^{-4}$ | $0.174 \pm 0.008$ | $\mathbf{0.068 \pm 0.007}$ | $0.218 \pm 0.012$ |
| 32 | $10^{-3}$ | $2.406 \pm 0.066$ | $\mathbf{0.722 \pm 0.035}$ | $3.470 \pm 0.082$ |
| 32 | $10^{-4}$ | $0.092 \pm 0.010$ | $\mathbf{0.061 \pm 0.004}$ | $0.092 \pm 0.007$ |
| 64 | $10^{-3}$ | $0.817 \pm 0.022$ | $\mathbf{0.189 \pm 0.016}$ | $1.169 \pm 0.044$ |
| 64 | $10^{-4}$ | $\mathbf{0.062 \pm 0.005}$ | $\mathbf{0.055 \pm 0.003}$ | $\mathbf{0.066 \pm 0.009}$ |
| 128 | $10^{-3}$ | $0.277 \pm 0.016$ | $\mathbf{0.075 \pm 0.004}$ | $0.370 \pm 0.024$ |
| 128 | $10^{-4}$ | $\mathbf{0.061 \pm 0.005}$ | $\mathbf{0.058 \pm 0.005}$ | $\mathbf{0.058 \pm 0.004}$ |
| 256 | $10^{-3}$ | $0.135 \pm 0.013$ | $\mathbf{0.060 \pm 0.005}$ | $0.139 \pm 0.010$ |
| 256 | $10^{-4}$ | $\mathbf{0.057 \pm 0.004}$ | $\mathbf{0.055 \pm 0.004}$ | $\mathbf{0.060 \pm 0.004}$ |
| 512 | $10^{-3}$ | $0.086 \pm 0.007$ | $\mathbf{0.059 \pm 0.005}$ | $0.084 \pm 0.007$ |
| 512 | $10^{-4}$ | $\mathbf{0.057 \pm 0.007}$ | $\mathbf{0.056 \pm 0.005}$ | $\mathbf{0.058 \pm 0.004}$ |
| 1000 | $10^{-3}$ | $\mathbf{0.072 \pm 0.009}$ | $\mathbf{0.058 \pm 0.006}$ | $\mathbf{0.073 \pm 0.006}$ |
| 1000 | $10^{-4}$ | $\mathbf{0.057 \pm 0.006}$ | $\mathbf{0.059 \pm 0.005}$ | $\mathbf{0.057 \pm 0.004}$ |

### A.5. Linear regression: KL divergence with standard deviations

Table 5 reports the same configurations as Table 1 with means and standard deviations over the 12 independent seeds.

Table 6 reports the same configurations as a ratio to the Monte Carlo reference, KL/0.0552, providing a more easily-parsed view of how far each sampler sits above the irreducible sampling-noise floor. Values near 1 indicate convergence to the floor.

### A.6. Dimension-dependence of the clipping bias

The validity restriction $\sqrt{\delta_t/2}\,|\widehat{\partial_i U}| \leq 1$ is enforced coordinate-wise by SGLRW (Definition 4.1). We test whether its activation rate depends on the parameter dimension $d$ using a Bayesian linear regression target with $N = 1000$, $\sigma^2 = 1.0$, $\tau = 10^{-2}$, and $X_{ij} \sim \mathcal{N}(0, 1)$, sweeping $d \in \{10, 20, 50, 100, 200\}$ and comparing SGLRW against SGLD and Clipped-SGLD at three configurations: heavy clipping ($B = 8$, $\delta_0 = 10^{-3}$), moderate clipping ($B = 32$, $\delta_0 = 10^{-3}$), and vanishing clipping ($B = 128$, $\delta_0 = 10^{-4}$). Each configuration uses 2,000 parallel chains for 10,000 steps (scaled linearly with $d$ above $d = 20$ to compensate for higher-dimensional equilibration), averaged over 5 sampler seeds.

The per-coordinate clipping rate (Table 7) does not grow with $d$ in any of the three regimes: weakly decreasing under heavy clipping, more than halving from $d = 10$ to $d = 200$ under moderate clipping, and identically zero under vanishing clipping. The clipping-induced mean bias (Table 8) is also dimension-independent and nearly identical between SGLRW

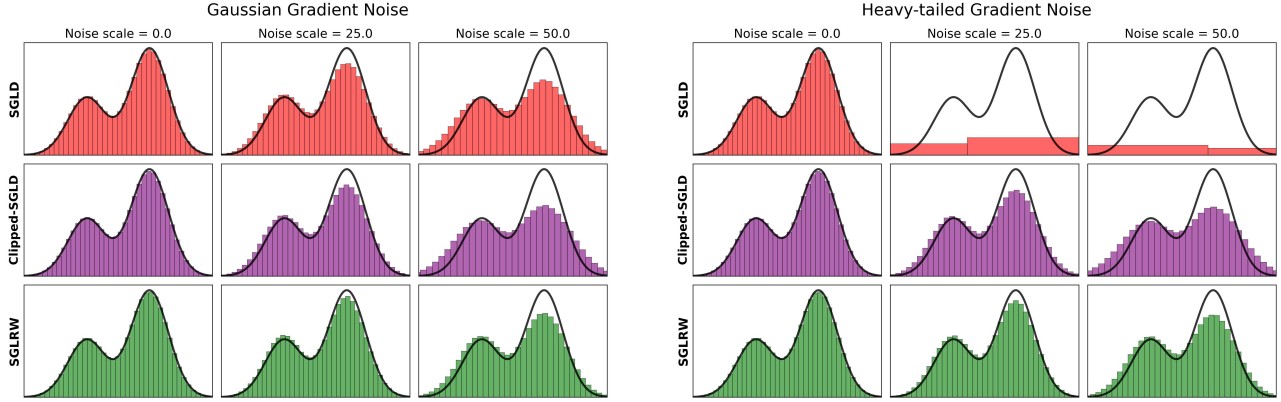

*Figure 7.* Multimodal univariate target with exact gradient corrupted by synthetic noise of increasing scale. **Left:** Gaussian noise ($\alpha = 2$). **Right:** heavy-tailed noise drawn from an $\alpha$-stable distribution with $\alpha = 1.5$.

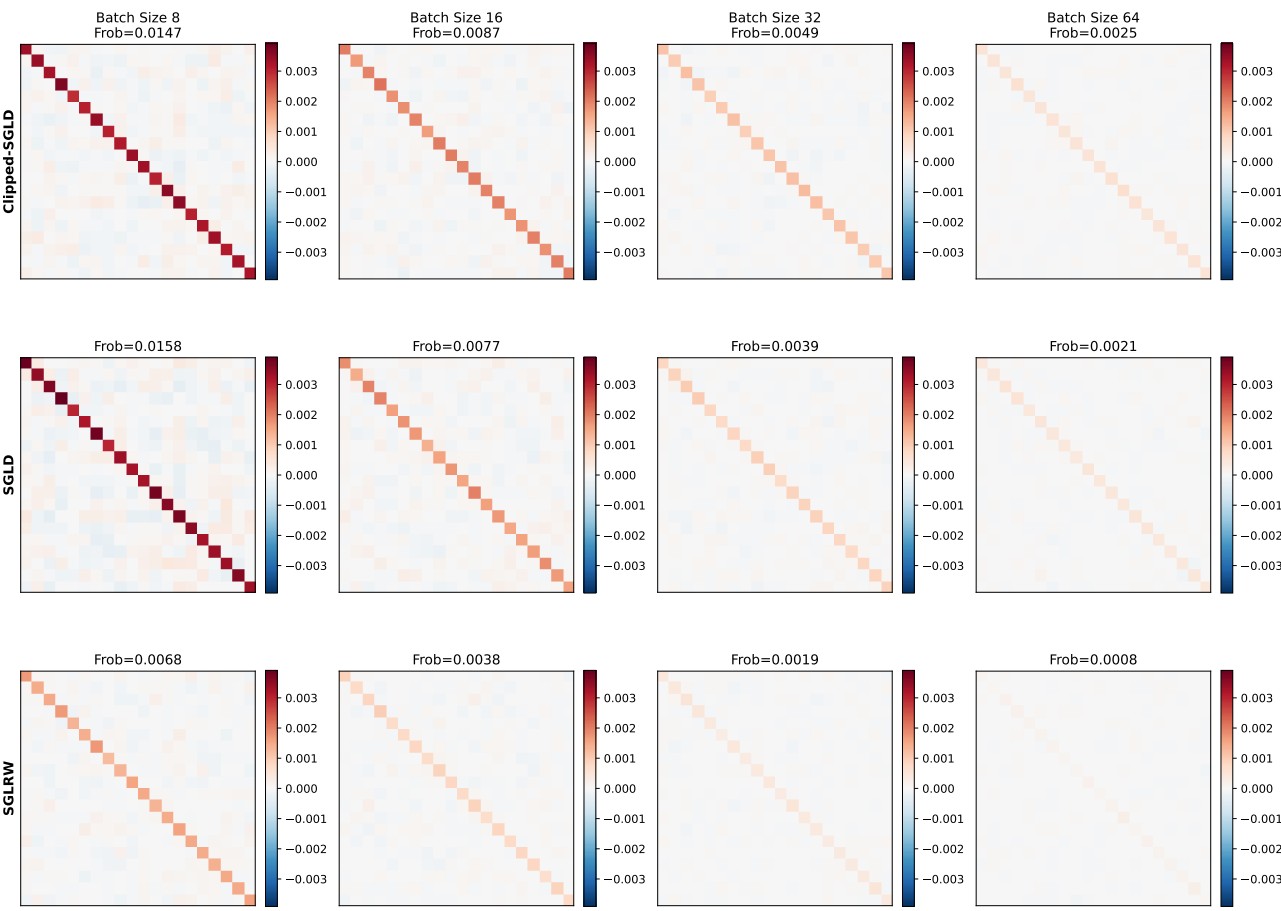

*Figure 8.* Covariance difference matrices $\Sigma_{\text{est}} - \Sigma_{\text{true}}$ for Bayesian linear regression at base step size $\delta_0 = 10^{-3}$, shown across increasing minibatch sizes $B$. **Top:** Clipped-SGLD. **Middle:** SGLD. **Bottom:** SGLRW. Each panel visualises the deviation of the empirical posterior covariance from the analytic posterior covariance; the Frobenius norm (Frob) reports the total error magnitude. The error in the diagonal terms is lower for SGLRW than for SGLD and Clipped-SGLD at every batch size.

and Clipped-SGLD, confirming that the mean-shift cost is a property of the clip threshold rather than the surrounding discretisation.

SGLRW's advantage over both alternatives lies in the covariance contribution, consistent with Theorem 4.2. Under heavy clipping its per-coordinate diagonal covariance error is approximately $0.45\times$ SGLD's at every $d$ (Table 9); at $d = 200$ in this regime SGLD's covariance diverges entirely while SGLRW and Clipped-SGLD remain stable. Under moderate clipping the ratio improves further with $d$; under vanishing clipping all three samplers collapse to the Monte Carlo finite-sample floor. The full Frobenius covariance error (Table 10) tells the same story.

### A.7. Cyclic SGLRW on Bayesian linear regression

The setup matches Section 5.2: the same posterior is sampled by cyclic SGLD (Zhang et al.) and a cyclic SGLRW variant. Cyclic SGLRW substitutes the cosine schedule

$$\delta_t = \frac{\delta_{\max}}{2}\big(1 + \cos(\pi r_t)\big), \qquad r_t = \frac{t \bmod K}{K}, \tag{120}$$

(with $K = \lceil T/n_{\text{cycles}}\rceil$, $n_{\text{cycles}} = 4$, $T = 10{,}000$) into the SGLRW update of Definition 4.1. SGLRW's spatial step size $\Delta x \propto \sqrt{\delta_t}$ vanishes at cycle boundaries, so we do not separately reset the temperature, in contrast to cyclic SGLD's two-phase construction. We use 2,000 parallel chains and 12 seeds. Cyclic SGLRW dominates cyclic SGLD at small $B$ or large $\delta_0$, and matches it at large $B$ and small $\delta_0$ where both reach the Monte Carlo floor. Cyclic SGLD diverges at $\delta_0 = 10^{-2}$ for every batch size.

*Table 6.* Linear regression KL divergence relative to the Monte Carlo reference (KL/0.0552), as means over 12 seeds. **Bold** marks the lowest mean per row, together with any method whose $\pm 1\sigma$ interval overlaps the minimum.

| Hyperparameters | | KL Divergence (relative) | | |
|---|---|---|---|---|
| $B$ | $\delta_0$ | SGLD | SGLRW | Clipped-SGLD |
| 8 | $10^{-3}$ | 343 | **108** | 327 |
| 8 | $10^{-4}$ | 8.39 | **3.42** | 13.77 |
| 16 | $10^{-3}$ | 122 | **40.7** | 150 |
| 16 | $10^{-4}$ | 3.08 | **1.20** | 3.50 |
| 32 | $10^{-3}$ | 41.0 | **12.6** | 60.0 |
| 32 | $10^{-4}$ | 1.56 | **1.10** | 1.63 |
| 64 | $10^{-3}$ | 13.8 | **3.13** | 19.6 |
| 64 | $10^{-4}$ | **1.16** | 1.05 | **1.20** |
| 128 | $10^{-3}$ | 4.93 | **1.34** | 6.01 |
| 128 | $10^{-4}$ | **1.05** | 1.00 | **1.09** |
| 256 | $10^{-3}$ | 2.28 | **1.12** | 2.39 |
| 256 | $10^{-4}$ | **1.01** | 1.03 | **1.07** |
| 512 | $10^{-3}$ | 1.49 | **1.07** | 1.49 |
| 512 | $10^{-4}$ | **1.00** | 1.05 | **1.05** |
| 1000 | $10^{-3}$ | **1.07** | 1.09 | **1.09** |
| 1000 | $10^{-4}$ | **1.00** | 1.03 | **1.05** |

*Table 7.* Per-coordinate clipping rate (steady-state).

| Sampler | $d = 10$ | $d = 20$ | $d = 50$ | $d = 100$ | $d = 200$ |
|---|---|---|---|---|---|
| *Heavy clipping ($B = 8$, $\delta_0 = 10^{-3}$)* | | | | | |
| SGLRW | 0.545 | 0.564 | 0.498 | 0.453 | 0.420 |
| Clipped-SGLD | 0.551 | 0.572 | 0.515 | 0.483 | 0.473 |
| *Moderate clipping ($B = 32$, $\delta_0 = 10^{-3}$)* | | | | | |
| SGLRW | 0.264 | 0.286 | 0.204 | 0.152 | 0.113 |
| Clipped-SGLD | 0.271 | 0.294 | 0.214 | 0.164 | 0.128 |
| *Vanishing clipping ($B = 128$, $\delta_0 = 10^{-4}$)* | | | | | |
| SGLRW | 0.000 | 0.000 | 0.000 | 0.000 | 0.000 |
| Clipped-SGLD | 0.000 | 0.000 | 0.000 | 0.000 | 0.000 |

## A.8. Preconditioned SGLRW on an anisotropic Gaussian

The target is a $d = 100$ zero-mean Gaussian $\mathcal{N}(0, \Sigma)$ with $\Sigma = Q \operatorname{diag}(\lambda_1, \ldots, \lambda_d) Q^\top$, where $Q$ is a random orthogonal matrix and the eigenvalues $\lambda_i$ are geometrically spaced from $\lambda_{\min} = 0.01$ to $\lambda_{\max} = 2.0$, giving condition number $\kappa = 200$. The drift is the exact gradient $\nabla U = \Sigma^{-1} x$ corrupted by additive Gaussian noise of standard deviation 0.05, isolating the effect of anisotropy from minibatch stochasticity. The RMSprop preconditioner uses decay $\alpha = 0.9999$ and regularisation $\epsilon = 10^{-3}$ as in Li et al. (2016); for pSGLRW the same preconditioner scales both the drift and the lattice step size. Each run uses 200,000 steps; the step size $\delta_0$ takes five geometrically-spaced values $0.03 \cdot 0.9^k$ for $k = 0, \ldots, 4$. Table 12 reports the same configurations as Table 3 with standard deviations. Vanilla SGLD diverges at four of five tested $\delta_0$, preconditioning recovers both methods, and pSGLRW achieves lower KL than pSGLD at every step size in the sweep.

## A.9. Logistic regression: KL divergence with standard deviations

Table 13 reports the same configurations as Table 4 with means and standard deviations over the 12 independent seeds.

## A.10. Schedule ablation on Bayesian linear regression

We assess the sensitivity of SGLD, SGLRW, and Clipped-SGLD to the choice of learning-rate schedule on the linear regression posterior of Section 5.2. Each sampler is run under fourteen schedules grouped by family: three fixed step sizes $\delta \in \{10^{-3}, 10^{-4}, 10^{-5}\}$, nine polynomial decays $\delta_t = \delta_0(1 + t)^{-p}$ with $\delta_0 \in \{10^{-2}, 10^{-3}, 10^{-4}\}$ and $p \in \{0.33, 0.55, 0.75\}$, and two cyclic cosine schedules with $\delta_{\max} \in \{10^{-3}, 10^{-4}\}$. Each configuration uses 2,000 parallel chains for 10,000 steps, averaged over 12 seeds with the data seed fixed across runs. Across the $14 \times 8 = 112$ (schedule, batch size) configurations per sampler, SGLRW and Clipped-SGLD produced a finite KL in every case; SGLD diverged in several configurations at large initial step size. Tables 14 to 16 report means $\pm$ standard deviation for four representative

*Table 8.* Normalised mean error $\|\hat{\mu} - \mu\|_2/\sqrt{d}$.

| Sampler | $d = 10$ | $d = 20$ | $d = 50$ | $d = 100$ | $d = 200$ |
|---|---|---|---|---|---|
| *Heavy clipping ($B = 8$, $\delta_0 = 10^{-3}$)* | | | | | |
| SGLD | 0.0014 | 0.0017 | 0.0013 | 0.0013 | — |
| SGLRW | 0.0041 | 0.0044 | 0.0062 | 0.0046 | 0.0048 |
| Clipped-SGLD | 0.0042 | 0.0046 | 0.0060 | 0.0045 | 0.0049 |
| *Moderate clipping ($B = 32$, $\delta_0 = 10^{-3}$)* | | | | | |
| SGLD | 0.0009 | 0.0010 | 0.0009 | 0.0009 | 0.0009 |
| SGLRW | 0.0013 | 0.0015 | 0.0013 | 0.0011 | 0.0011 |
| Clipped-SGLD | 0.0016 | 0.0016 | 0.0015 | 0.0011 | 0.0012 |
| *Vanishing clipping ($B = 128$, $\delta_0 = 10^{-4}$)* | | | | | |
| SGLD | 0.0007 | 0.0007 | 0.0008 | 0.0007 | 0.0008 |
| SGLRW | 0.0007 | 0.0007 | 0.0008 | 0.0007 | 0.0008 |
| Clipped-SGLD | 0.0007 | 0.0007 | 0.0007 | 0.0008 | 0.0008 |
| MC reference | 0.0007 | 0.0007 | 0.0008 | 0.0007 | 0.0008 |

*Table 9.* Diagonal-only covariance error $\|\mathrm{diag}(\hat{\Sigma} - \Sigma)\|_2/\sqrt{d}$.

| Sampler | $d = 10$ | $d = 20$ | $d = 50$ | $d = 100$ | $d = 200$ |
|---|---|---|---|---|---|
| *Heavy clipping ($B = 8$, $\delta_0 = 10^{-3}$)* | | | | | |
| SGLD | $3.02 \times 10^{-3}$ | $3.44 \times 10^{-3}$ | $2.54 \times 10^{-3}$ | $2.10 \times 10^{-3}$ | — |
| SGLRW | $1.36 \times 10^{-3}$ | $1.49 \times 10^{-3}$ | $1.15 \times 10^{-3}$ | $9.63 \times 10^{-4}$ | $8.44 \times 10^{-4}$ |
| Clipped-SGLD | $3.04 \times 10^{-3}$ | $3.31 \times 10^{-3}$ | $2.73 \times 10^{-3}$ | $2.50 \times 10^{-3}$ | $2.57 \times 10^{-3}$ |
| *Moderate clipping ($B = 32$, $\delta_0 = 10^{-3}$)* | | | | | |
| SGLD | $7.67 \times 10^{-4}$ | $8.44 \times 10^{-4}$ | $5.90 \times 10^{-4}$ | $4.66 \times 10^{-4}$ | $3.83 \times 10^{-4}$ |
| SGLRW | $3.46 \times 10^{-4}$ | $3.79 \times 10^{-4}$ | $2.40 \times 10^{-4}$ | $1.48 \times 10^{-4}$ | $8.30 \times 10^{-5}$ |
| Clipped-SGLD | $9.56 \times 10^{-4}$ | $1.08 \times 10^{-3}$ | $7.79 \times 10^{-4}$ | $6.24 \times 10^{-4}$ | $5.32 \times 10^{-4}$ |
| *Vanishing clipping ($B = 128$, $\delta_0 = 10^{-4}$)* | | | | | |
| SGLD | $3.90 \times 10^{-5}$ | $4.20 \times 10^{-5}$ | $3.70 \times 10^{-5}$ | $3.90 \times 10^{-5}$ | $4.20 \times 10^{-5}$ |
| SGLRW | $3.10 \times 10^{-5}$ | $3.20 \times 10^{-5}$ | $3.20 \times 10^{-5}$ | $3.50 \times 10^{-5}$ | $4.10 \times 10^{-5}$ |
| Clipped-SGLD | $4.50 \times 10^{-5}$ | $3.60 \times 10^{-5}$ | $4.00 \times 10^{-5}$ | $3.70 \times 10^{-5}$ | $4.20 \times 10^{-5}$ |
| MC reference | $3.90 \times 10^{-5}$ | $4.20 \times 10^{-5}$ | $3.70 \times 10^{-5}$ | $3.90 \times 10^{-5}$ | $4.20 \times 10^{-5}$ |

batch sizes ($B \in \{8, 32, 128, 1000\}$). At $\delta_0 = 10^{-4}$, $p = 0.75$ the step size decays so aggressively that both bounded-step methods (SGLRW and Clipped-SGLD) cannot traverse from the initialisation to the posterior within the 10,000-step budget, while SGLD's unbounded Gaussian increments still make partial progress; this delimits the lower edge of viable step sizes for bounded-update schemes rather than a failure mode specific to the lattice discretisation.

### A.11. Schedule ablation on UCI logistic regression

We repeat the schedule ablation of Section A.10 on the UCI breast-cancer logistic posterior of Section 5.3, using the same fourteen schedules: three fixed step sizes $\delta \in \{10^{-3}, 10^{-4}, 10^{-5}\}$, nine polynomial decays $\delta_t = \delta_0(1 + t)^{-p}$ with $\delta_0 \in \{10^{-2}, 10^{-3}, 10^{-4}\}$ and $p \in \{0.33, 0.55, 0.75\}$, and two cyclic cosine schedules with $\delta_{\max} \in \{10^{-3}, 10^{-4}\}$. Each configuration uses 5,000 parallel chains for 1,000 steps, averaged over 12 seeds. Unlike the linear regression case, no sampler diverges on this dataset; SGLD's failure mode is to plateau at elevated KL. SGLRW improves substantially on SGLD at aggressive schedules, most strikingly at the cyclic schedule $\delta_{\max} = 10^{-3}$, where SGLRW reaches KL 0.62 at $B = 32$ compared to SGLD's 11.07. At very conservative schedules all three samplers plateau at similar KL within the 1000-step budget; Table 20 confirms this is an under-convergence artefact.

**Supplementary 5000-step convergence test.** A supplementary 5000-step rerun on two representative conservative schedules shows that the elevated KL at 1000 steps reflects under-convergence rather than a structural limit of the samplers. At $\delta_0 = 10^{-3}$, $p = 0.33$ the KL drops by a factor of 3–5 and all three samplers agree within seed variation at $B \geq 32$. At $\delta_0 = 10^{-3}$, $p = 0.55$ a 5× step budget only halves the KL, with all samplers plateauing near 4.3; this row corresponds to the same lower-edge regime as the $\delta_0 = 10^{-4}$, $p = 0.75$ row of Section A.10, where the step size decays faster than the budget can compensate.

*Table 10.* Normalised Frobenius covariance error $\|\hat{\Sigma} - \Sigma\|_F/d$.

| Sampler | $d = 10$ | $d = 20$ | $d = 50$ | $d = 100$ | $d = 200$ |
|---|---|---|---|---|---|
| *Heavy clipping ($B = 8$, $\delta_0 = 10^{-3}$)* | | | | | |
| SGLD | $9.65\times10^{-4}$ | $7.88\times10^{-4}$ | $3.79\times10^{-4}$ | $2.31\times10^{-4}$ | — |
| SGLRW | $4.35\times10^{-4}$ | $3.42\times10^{-4}$ | $1.76\times10^{-4}$ | $1.14\times10^{-4}$ | $8.20\times10^{-5}$ |
| Clipped-SGLD | $9.67\times10^{-4}$ | $7.52\times10^{-4}$ | $4.01\times10^{-4}$ | $2.70\times10^{-4}$ | $2.10\times10^{-4}$ |
| *Moderate clipping ($B = 32$, $\delta_0 = 10^{-3}$)* | | | | | |
| SGLD | $2.46\times10^{-4}$ | $1.96\times10^{-4}$ | $9.40\times10^{-5}$ | $6.10\times10^{-5}$ | $4.70\times10^{-5}$ |
| SGLRW | $1.16\times10^{-4}$ | $9.30\times10^{-5}$ | $4.90\times10^{-5}$ | $3.60\times10^{-5}$ | $3.30\times10^{-5}$ |
| Clipped-SGLD | $3.06\times10^{-4}$ | $2.47\times10^{-4}$ | $1.19\times10^{-4}$ | $7.50\times10^{-5}$ | $5.60\times10^{-5}$ |
| *Vanishing clipping ($B = 128$, $\delta_0 = 10^{-4}$)* | | | | | |
| SGLD | $2.60\times10^{-5}$ | $2.60\times10^{-5}$ | $2.40\times10^{-5}$ | $2.50\times10^{-5}$ | $2.80\times10^{-5}$ |
| SGLRW | $2.40\times10^{-5}$ | $2.40\times10^{-5}$ | $2.40\times10^{-5}$ | $2.50\times10^{-5}$ | $2.80\times10^{-5}$ |
| Clipped-SGLD | $2.60\times10^{-5}$ | $2.50\times10^{-5}$ | $2.40\times10^{-5}$ | $2.50\times10^{-5}$ | $2.80\times10^{-5}$ |
| MC reference | $2.60\times10^{-5}$ | $2.60\times10^{-5}$ | $2.40\times10^{-5}$ | $2.50\times10^{-5}$ | $2.80\times10^{-5}$ |

*Table 11.* Cyclic SGLRW vs. cyclic SGLD on Bayesian linear regression. KL divergence as mean $\pm$ standard deviation over 12 seeds. **Bold** marks the minimum mean per cell, together with any method whose $\pm 1\sigma$ interval overlaps the minimum. — indicates divergence on all seeds.

| | $\delta_0 = 10^{-5}$ | | $\delta_0 = 10^{-4}$ | | $\delta_0 = 10^{-3}$ | | $\delta_0 = 10^{-2}$ | |
|---|---|---|---|---|---|---|---|---|
| $B$ | cSGLD | cSGLRW | cSGLD | cSGLRW | cSGLD | cSGLRW | cSGLD | cSGLRW |
| 8 | $0.108\pm0.013$ | $\mathbf{0.077\pm0.009}$ | $0.346\pm0.028$ | $\mathbf{0.181\pm0.020}$ | — | $\mathbf{0.648\pm0.074}$ | — | $2.48\pm0.21$ |
| 16 | $\mathbf{0.069\pm0.008}$ | $\mathbf{0.060\pm0.006}$ | $0.140\pm0.013$ | $\mathbf{0.070\pm0.005}$ | — | $\mathbf{0.138\pm0.016}$ | — | $0.577\pm0.079$ |
| 32 | $\mathbf{0.061\pm0.006}$ | $\mathbf{0.058\pm0.004}$ | $0.080\pm0.007$ | $\mathbf{0.056\pm0.006}$ | $0.156\pm0.017$ | $\mathbf{0.067\pm0.007}$ | — | $\mathbf{0.133\pm0.016}$ |
| 64 | $\mathbf{0.059\pm0.005}$ | $\mathbf{0.058\pm0.005}$ | $\mathbf{0.066\pm0.006}$ | $\mathbf{0.057\pm0.006}$ | $0.082\pm0.006$ | $\mathbf{0.059\pm0.005}$ | — | $\mathbf{0.066\pm0.003}$ |
| 128 | $\mathbf{0.059\pm0.005}$ | $\mathbf{0.059\pm0.007}$ | $\mathbf{0.061\pm0.006}$ | $\mathbf{0.057\pm0.006}$ | $\mathbf{0.062\pm0.006}$ | $\mathbf{0.057\pm0.005}$ | — | $\mathbf{0.058\pm0.006}$ |
| 256 | $\mathbf{0.060\pm0.005}$ | $\mathbf{0.058\pm0.005}$ | $\mathbf{0.060\pm0.007}$ | $\mathbf{0.058\pm0.007}$ | $\mathbf{0.057\pm0.005}$ | $\mathbf{0.056\pm0.006}$ | — | $\mathbf{0.056\pm0.004}$ |
| 512 | $\mathbf{0.060\pm0.005}$ | $\mathbf{0.058\pm0.004}$ | $\mathbf{0.059\pm0.007}$ | $\mathbf{0.058\pm0.004}$ | $\mathbf{0.055\pm0.005}$ | $\mathbf{0.057\pm0.005}$ | — | $\mathbf{0.057\pm0.005}$ |
| 1000 | $\mathbf{0.060\pm0.005}$ | $\mathbf{0.059\pm0.006}$ | $\mathbf{0.059\pm0.007}$ | $\mathbf{0.057\pm0.005}$ | $\mathbf{0.054\pm0.005}$ | $\mathbf{0.056\pm0.003}$ | — | $\mathbf{0.055\pm0.004}$ |

## A.12. Full Experimental Details: Sentiment Classification

This appendix provides complete experimental details and full results for the sentiment classification experiments presented in Section 5.4. We report results for all evaluated learning-rate schedules, training-set sizes, and minibatch sizes, using fixed embeddings extracted from a pretrained OPT-350M model.

**Experimental grid.** For each method (Clipped-SGLD and SGLRW), we evaluate training set sizes $N \in \{10{,}000, 15{,}000, 20{,}000, 25{,}000\}$ and minibatch sizes $B \in \{8, 16, 32, 64, 128\}$. For each configuration, we perform three independent runs, each consisting of 15 independent chains of length 10,000, discarding the first 5,000 iterations as burn-in.

**Learning-rate schedules.** For the experiments we employ a decaying learning-rate schedule of the form

$$\delta_t = s \cdot \delta_0 \left(t/T_{\text{burnin}} + 0.1\right)^{-0.5}, \tag{121}$$

where $\delta_0 = 5 \times 10^{-5}$, $T_{\text{burnin}} = 5{,}000$, and $s \in \{0.5, 1.0, 3.0\}$ denotes a multiplicative scale factor. We refer to these as the conservative ($0.5\times$), baseline ($1.0\times$), and increased ($3.0\times$) learning-rate scales, respectively.

**Metrics.** Predictive metrics are computed on the held-out test set using posterior predictive probabilities obtained by averaging predicted probabilities across retained MCMC samples. Classification accuracy is computed by thresholding the averaged probability at $0.5$. The negative log-likelihood (NLL) is computed as

$$\text{NLL} = -\frac{1}{N} \sum_{i=1}^{N} \left[ y_i \log(p_i) + (1 - y_i) \log(1 - p_i) \right], \tag{122}$$

where $y_i \in \{0, 1\}$ is the true label for test example $i$, $N$ is the number of test examples, and $p_i$ denotes the averaged posterior predictive probability for sample $i$. Expected calibration error (ECE) is computed using $K = 10$ equal-width bins over

*Table 12.* KL divergence on a $d = 100$ anisotropic Gaussian (condition number $\kappa = 200$), reported as mean $\pm$ standard deviation over 10 seeds. **Bold** marks the lowest mean per row, together with any method whose $\pm 1\sigma$ interval overlaps the minimum. — indicates divergence on all seeds.

| $\delta_0$ | SGLD | SGLRW | pSGLD | pSGLRW |
|---|---|---|---|---|
| 0.0300 | — | $28.72 \pm 1.39$ | $0.764 \pm 0.012$ | $\mathbf{0.637 \pm 0.010}$ |
| 0.0270 | — | $20.51 \pm 0.79$ | $0.742 \pm 0.020$ | $\mathbf{0.622 \pm 0.020}$ |
| 0.0243 | — | $13.89 \pm 0.41$ | $0.716 \pm 0.013$ | $\mathbf{0.622 \pm 0.021}$ |
| 0.0219 | — | $9.19 \pm 0.22$ | $0.730 \pm 0.019$ | $\mathbf{0.651 \pm 0.020}$ |
| 0.0197 | $44.58 \pm 0.62$ | $6.09 \pm 0.11$ | $0.743 \pm 0.019$ | $\mathbf{0.660 \pm 0.034}$ |

*Table 13.* Inferred Kullback–Leibler divergence for the UCI logistic regression problem, reported as mean $\pm$ standard deviation over 12 independent seeds. **Bold** marks the lowest mean per row, together with any method whose $\pm 1\sigma$ interval overlaps the minimum.

| $B$ | $\delta_0$ | SGLD | SGLRW | Clipped-SGLD |
|---|---|---|---|---|
| 1 | $10^0$ | $46.351 \pm 0.208$ | $\mathbf{8.230 \pm 0.064}$ | $9.276 \pm 0.075$ |
| 1 | $10^{-1}$ | $16.536 \pm 0.099$ | $\mathbf{5.884 \pm 0.063}$ | $6.420 \pm 0.064$ |
| 1 | $10^{-2}$ | $10.300 \pm 0.208$ | $\mathbf{3.845 \pm 0.062}$ | $4.162 \pm 0.050$ |
| 2 | $10^0$ | $36.271 \pm 0.154$ | $\mathbf{7.582 \pm 0.059}$ | $9.004 \pm 0.069$ |
| 2 | $10^{-1}$ | $8.901 \pm 0.076$ | $\mathbf{4.868 \pm 0.064}$ | $5.515 \pm 0.070$ |
| 2 | $10^{-2}$ | $5.387 \pm 0.096$ | $\mathbf{2.521 \pm 0.059}$ | $2.774 \pm 0.060$ |
| 4 | $10^0$ | $27.221 \pm 0.106$ | $\mathbf{6.870 \pm 0.048}$ | $8.811 \pm 0.071$ |
| 4 | $10^{-1}$ | $\mathbf{3.586 \pm 0.063}$ | $\mathbf{3.522 \pm 0.070}$ | $4.231 \pm 0.073$ |
| 4 | $10^{-2}$ | $2.894 \pm 0.047$ | $\mathbf{1.393 \pm 0.042}$ | $1.550 \pm 0.039$ |
| 8 | $10^0$ | $18.889 \pm 0.113$ | $\mathbf{5.904 \pm 0.044}$ | $8.333 \pm 0.056$ |
| 8 | $10^{-1}$ | $\mathbf{1.228 \pm 0.049}$ | $1.838 \pm 0.054$ | $2.395 \pm 0.051$ |
| 8 | $10^{-2}$ | $1.876 \pm 0.040$ | $\mathbf{0.930 \pm 0.029}$ | $\mathbf{0.988 \pm 0.042}$ |
| 16 | $10^0$ | $11.756 \pm 0.078$ | $\mathbf{4.385 \pm 0.040}$ | $6.973 \pm 0.076$ |
| 16 | $10^{-1}$ | $\mathbf{0.454 \pm 0.022}$ | $0.882 \pm 0.042$ | $1.145 \pm 0.036$ |
| 16 | $10^{-2}$ | $1.441 \pm 0.042$ | $\mathbf{0.765 \pm 0.033}$ | $\mathbf{0.757 \pm 0.028}$ |
| 32 | $10^0$ | $6.308 \pm 0.045$ | $\mathbf{2.733 \pm 0.043}$ | $4.762 \pm 0.048$ |
| 32 | $10^{-1}$ | $\mathbf{0.219 \pm 0.018}$ | $0.411 \pm 0.018$ | $0.498 \pm 0.023$ |
| 32 | $10^{-2}$ | $1.251 \pm 0.023$ | $\mathbf{0.647 \pm 0.026}$ | $\mathbf{0.634 \pm 0.024}$ |
| 64 | $10^0$ | $2.914 \pm 0.040$ | $\mathbf{1.570 \pm 0.056}$ | $2.783 \pm 0.066$ |
| 64 | $10^{-1}$ | $\mathbf{0.158 \pm 0.010}$ | $0.183 \pm 0.013$ | $0.206 \pm 0.011$ |
| 64 | $10^{-2}$ | $1.161 \pm 0.032$ | $\mathbf{0.618 \pm 0.017}$ | $\mathbf{0.599 \pm 0.019}$ |

$[0, 1]$,

$$\text{ECE} = \sum_{k=1}^{K} \frac{|B_k|}{N} \left| \text{acc}(B_k) - \text{conf}(B_k) \right|, \tag{123}$$

where $\text{acc}(B_k)$ and $\text{conf}(B_k)$ denote the empirical accuracy and mean predicted probability within bin $B_k$. Reported values correspond to means across chains; variability across chains is shown where indicated.

**Reading the heatmaps.** Figures 9, 10, and 11 report absolute predictive accuracy, negative log-likelihood (NLL), and expected calibration error (ECE) for Clipped-SGLD and SGLRW across the full experimental grid. These figures complement the relative-improvement summaries shown in the main text and allow inspection of absolute performance across regimes.

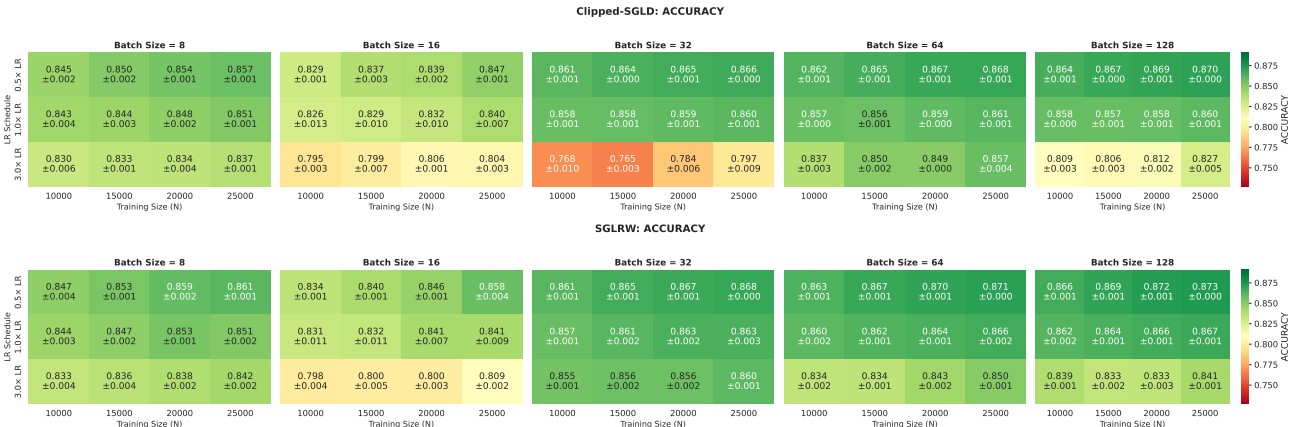

*Figure 9.* Predictive accuracy heatmaps for sentiment classification using fixed OPT-350M embeddings. **Top:** Clipped-SGLD. **Bottom:** SGLRW. Each panel corresponds to a minibatch size $B$, with columns showing training-set size $N$ and rows indicating learning-rate scale. Values report mean test accuracy across chains, with standard deviation shown below each entry.

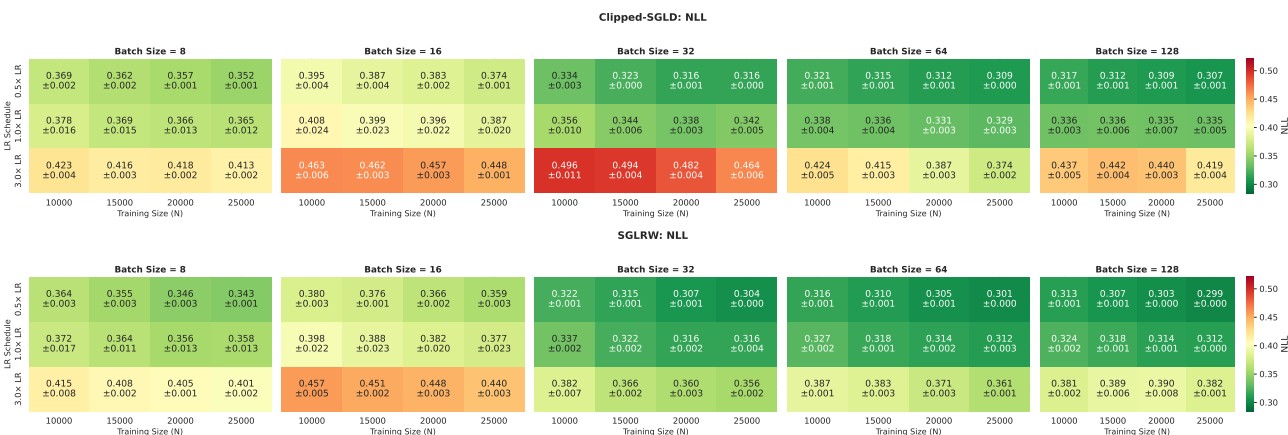

*Figure 10.* NLL heatmaps for sentiment classification using fixed OPT-350M embeddings. **Top:** Clipped-SGLD. **Bottom:** SGLRW. Each panel corresponds to a minibatch size $B$, with columns showing training-set size $N$ and rows indicating learning-rate scale. Values report mean NLL values across chains, with standard deviation shown below each entry.

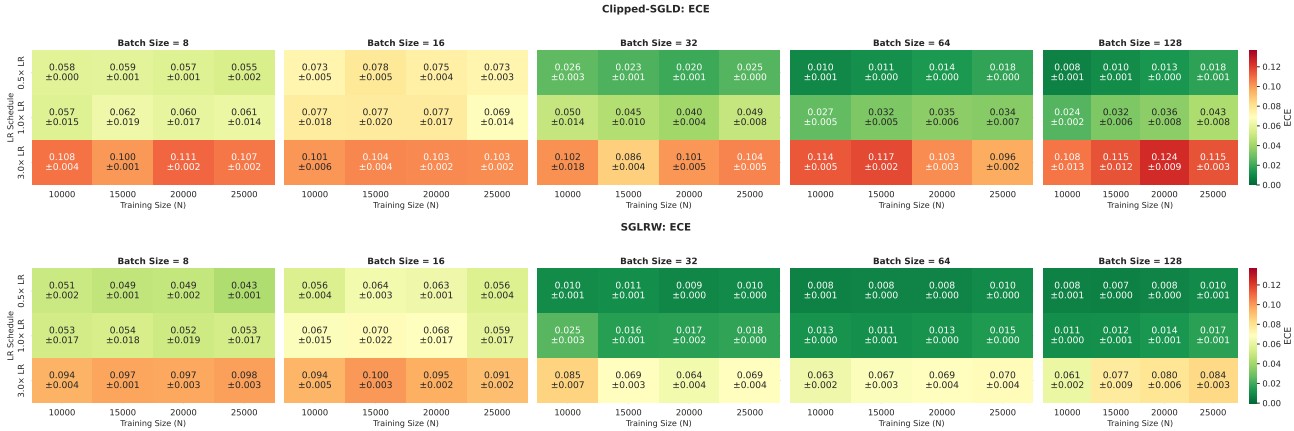

*Figure 11.* ECE heatmaps for sentiment classification using fixed OPT-350M embeddings. **Top:** Clipped-SGLD. **Bottom:** SGLRW. Each panel corresponds to a minibatch size $B$, with columns showing training-set size $N$ and rows indicating learning-rate scale. Values report mean ECE values across chains, with standard deviation shown below each entry.

*Table 14.* SGLD KL divergence under the schedule ablation, as mean $\pm$ standard deviation over 12 seeds. — indicates divergence on all seeds.

| Schedule | $B = 8$ | $B = 32$ | $B = 128$ | $B = 1000$ |
|---|---|---|---|---|
| *Fixed step size* | | | | |
| $\delta = 10^{-3}$ | — | $1094.78 \pm 17.83$ | $91.222 \pm 0.849$ | $10.506 \pm 0.220$ |
| $\delta = 10^{-4}$ | $58.820 \pm 0.404$ | $8.162 \pm 0.140$ | $1.004 \pm 0.043$ | $0.114 \pm 0.008$ |
| $\delta = 10^{-5}$ | $1.690 \pm 0.063$ | $0.189 \pm 0.011$ | $0.069 \pm 0.006$ | $0.059 \pm 0.004$ |
| *Polynomial decay $\delta_t = \delta_0(1 + t)^{-p}$* | | | | |
| $\delta_0 = 10^{-2}, p = 0.33$ | — | — | — | — |
| $\delta_0 = 10^{-2}, p = 0.55$ | — | $3.829 \pm 0.057$ | $0.449 \pm 0.029$ | $0.079 \pm 0.007$ |
| $\delta_0 = 10^{-2}, p = 0.75$ | $1.712 \pm 0.033$ | $0.198 \pm 0.010$ | $0.067 \pm 0.008$ | $0.057 \pm 0.002$ |
| $\delta_0 = 10^{-3}, p = 0.33$ | $19.853 \pm 0.337$ | $2.406 \pm 0.066$ | $0.277 \pm 0.016$ | $0.072 \pm 0.009$ |
| $\delta_0 = 10^{-3}, p = 0.55$ | $0.794 \pm 0.037$ | $0.114 \pm 0.013$ | $0.060 \pm 0.006$ | $0.059 \pm 0.004$ |
| $\delta_0 = 10^{-3}, p = 0.75$ | $0.083 \pm 0.006$ | $0.058 \pm 0.005$ | $0.060 \pm 0.006$ | $0.058 \pm 0.005$ |
| $\delta_0 = 10^{-4}, p = 0.33$ | $0.504 \pm 0.019$ | $0.092 \pm 0.010$ | $0.061 \pm 0.005$ | $0.057 \pm 0.006$ |
| $\delta_0 = 10^{-4}, p = 0.55$ | $0.065 \pm 0.006$ | $0.058 \pm 0.003$ | $0.059 \pm 0.004$ | $0.058 \pm 0.005$ |
| $\delta_0 = 10^{-4}, p = 0.75$ | $12.035 \pm 0.117$ | $11.501 \pm 0.136$ | $11.456 \pm 0.165$ | $11.325 \pm 0.170$ |
| *Cyclic cosine* | | | | |
| $\delta_{\max} = 10^{-3}$ | — | $0.160 \pm 0.009$ | $0.069 \pm 0.007$ | $0.058 \pm 0.005$ |
| $\delta_{\max} = 10^{-4}$ | $0.365 \pm 0.017$ | $0.084 \pm 0.007$ | $0.058 \pm 0.006$ | $0.056 \pm 0.004$ |

*Table 15.* SGLRW KL divergence under the schedule ablation, as mean $\pm$ standard deviation over 12 seeds.

| Schedule | $B = 8$ | $B = 32$ | $B = 128$ | $B = 1000$ |
|---|---|---|---|---|
| *Fixed step size* | | | | |
| $\delta = 10^{-3}$ | $84.978 \pm 0.709$ | $30.536 \pm 0.303$ | $9.927 \pm 0.085$ | $1.489 \pm 0.049$ |
| $\delta = 10^{-4}$ | $12.996 \pm 0.224$ | $2.450 \pm 0.051$ | $0.223 \pm 0.016$ | $0.058 \pm 0.005$ |
| $\delta = 10^{-5}$ | $0.686 \pm 0.025$ | $0.064 \pm 0.004$ | $0.058 \pm 0.004$ | $0.061 \pm 0.007$ |
| *Polynomial decay $\delta_t = \delta_0(1 + t)^{-p}$* | | | | |
| $\delta_0 = 10^{-2}, p = 0.33$ | $49.000 \pm 0.521$ | $15.600 \pm 0.179$ | $3.830 \pm 0.078$ | $0.264 \pm 0.013$ |
| $\delta_0 = 10^{-2}, p = 0.55$ | $8.204 \pm 0.162$ | $1.184 \pm 0.033$ | $0.098 \pm 0.011$ | $0.058 \pm 0.006$ |
| $\delta_0 = 10^{-2}, p = 0.75$ | $0.692 \pm 0.035$ | $0.064 \pm 0.004$ | $0.057 \pm 0.006$ | $0.061 \pm 0.006$ |
| $\delta_0 = 10^{-3}, p = 0.33$ | $6.060 \pm 0.137$ | $0.722 \pm 0.035$ | $0.075 \pm 0.004$ | $0.058 \pm 0.006$ |
| $\delta_0 = 10^{-3}, p = 0.55$ | $0.319 \pm 0.023$ | $0.060 \pm 0.006$ | $0.057 \pm 0.006$ | $0.058 \pm 0.006$ |
| $\delta_0 = 10^{-3}, p = 0.75$ | $0.061 \pm 0.009$ | $0.056 \pm 0.005$ | $0.057 \pm 0.005$ | $0.059 \pm 0.006$ |
| $\delta_0 = 10^{-4}, p = 0.33$ | $0.196 \pm 0.010$ | $0.061 \pm 0.004$ | $0.058 \pm 0.005$ | $0.059 \pm 0.005$ |
| $\delta_0 = 10^{-4}, p = 0.55$ | $0.057 \pm 0.006$ | $0.057 \pm 0.005$ | $0.061 \pm 0.005$ | $0.058 \pm 0.005$ |
| $\delta_0 = 10^{-4}, p = 0.75$ | $197.216 \pm 1.662$ | $100.788 \pm 0.872$ | $83.245 \pm 0.801$ | $79.284 \pm 0.797$ |
| *Cyclic cosine* | | | | |
| $\delta_{\max} = 10^{-3}$ | $0.638 \pm 0.019$ | $0.067 \pm 0.007$ | $0.058 \pm 0.007$ | $0.057 \pm 0.004$ |
| $\delta_{\max} = 10^{-4}$ | $0.168 \pm 0.013$ | $0.062 \pm 0.006$ | $0.058 \pm 0.006$ | $0.058 \pm 0.005$ |

*Table 16.* Clipped-SGLD KL divergence under the schedule ablation, as mean $\pm$ standard deviation over 12 seeds.

| Schedule | $B = 8$ | $B = 32$ | $B = 128$ | $B = 1000$ |
|---|---|---|---|---|
| *Fixed step size* | | | | |
| $\delta = 10^{-3}$ | $220.893 \pm 2.030$ | $87.903 \pm 1.177$ | $36.442 \pm 0.308$ | $11.789 \pm 0.176$ |
| $\delta = 10^{-4}$ | $36.784 \pm 0.304$ | $9.441 \pm 0.125$ | $1.489 \pm 0.054$ | $0.119 \pm 0.012$ |
| $\delta = 10^{-5}$ | $2.739 \pm 0.067$ | $0.231 \pm 0.014$ | $0.069 \pm 0.006$ | $0.060 \pm 0.007$ |
| *Polynomial decay $\delta_t = \delta_0(1 + t)^{-p}$* | | | | |
| $\delta_0 = 10^{-2}, p = 0.33$ | $127.287 \pm 1.041$ | $46.252 \pm 0.528$ | $15.711 \pm 0.175$ | $2.739 \pm 0.079$ |
| $\delta_0 = 10^{-2}, p = 0.55$ | $24.277 \pm 0.318$ | $5.206 \pm 0.115$ | $0.601 \pm 0.022$ | $0.080 \pm 0.006$ |
| $\delta_0 = 10^{-2}, p = 0.75$ | $2.743 \pm 0.055$ | $0.226 \pm 0.013$ | $0.068 \pm 0.004$ | $0.054 \pm 0.006$ |
| $\delta_0 = 10^{-3}, p = 0.33$ | $18.496 \pm 0.269$ | $3.470 \pm 0.082$ | $0.370 \pm 0.024$ | $0.073 \pm 0.006$ |
| $\delta_0 = 10^{-3}, p = 0.55$ | $1.326 \pm 0.040$ | $0.124 \pm 0.013$ | $0.061 \pm 0.006$ | $0.058 \pm 0.005$ |
| $\delta_0 = 10^{-3}, p = 0.75$ | $0.086 \pm 0.008$ | $0.061 \pm 0.006$ | $0.061 \pm 0.006$ | $0.056 \pm 0.005$ |
| $\delta_0 = 10^{-4}, p = 0.33$ | $0.822 \pm 0.037$ | $0.092 \pm 0.007$ | $0.058 \pm 0.004$ | $0.057 \pm 0.004$ |
| $\delta_0 = 10^{-4}, p = 0.55$ | $0.071 \pm 0.005$ | $0.057 \pm 0.008$ | $0.059 \pm 0.003$ | $0.056 \pm 0.005$ |
| $\delta_0 = 10^{-4}, p = 0.75$ | $198.272 \pm 1.740$ | $101.154 \pm 0.857$ | $83.471 \pm 0.898$ | $79.627 \pm 0.895$ |
| *Cyclic cosine* | | | | |
| $\delta_{\max} = 10^{-3}$ | $2.636 \pm 0.061$ | $0.194 \pm 0.021$ | $0.067 \pm 0.007$ | $0.058 \pm 0.006$ |
| $\delta_{\max} = 10^{-4}$ | $0.658 \pm 0.028$ | $0.082 \pm 0.007$ | $0.062 \pm 0.004$ | $0.058 \pm 0.005$ |

*Table 17.* SGLD KL divergence under the schedule ablation on UCI logistic regression, as mean $\pm$ standard deviation over 12 seeds.

| Schedule | $B = 1$ | $B = 8$ | $B = 32$ | $B = 64$ |
|---|---|---|---|---|
| *Fixed step size* | | | | |
| $\delta = 10^{-3}$ | $7.136 \pm 0.028$ | $0.435 \pm 0.012$ | $0.230 \pm 0.006$ | $0.214 \pm 0.010$ |
| $\delta = 10^{-4}$ | $3.915 \pm 0.062$ | $4.329 \pm 0.124$ | $4.398 \pm 0.082$ | $4.444 \pm 0.111$ |
| $\delta = 10^{-5}$ | $19.623 \pm 0.403$ | $19.697 \pm 0.532$ | $19.706 \pm 0.399$ | $19.746 \pm 0.462$ |
| *Polynomial decay $\delta_t = \delta_0(1 + t)^{-p}$* | | | | |
| $\delta_0 = 10^{-2}, p = 0.33$ | $9.240 \pm 0.055$ | $0.462 \pm 0.015$ | $0.187 \pm 0.007$ | $0.165 \pm 0.008$ |
| $\delta_0 = 10^{-2}, p = 0.55$ | $10.319 \pm 0.169$ | $1.885 \pm 0.028$ | $1.278 \pm 0.026$ | $1.186 \pm 0.024$ |
| $\delta_0 = 10^{-2}, p = 0.75$ | $16.069 \pm 0.207$ | $4.619 \pm 0.048$ | $3.137 \pm 0.029$ | $2.962 \pm 0.040$ |
| $\delta_0 = 10^{-3}, p = 0.33$ | $2.012 \pm 0.031$ | $2.527 \pm 0.071$ | $2.620 \pm 0.049$ | $2.668 \pm 0.067$ |
| $\delta_0 = 10^{-3}, p = 0.55$ | $5.920 \pm 0.153$ | $8.376 \pm 0.227$ | $8.604 \pm 0.145$ | $8.663 \pm 0.228$ |
| $\delta_0 = 10^{-3}, p = 0.75$ | $10.484 \pm 0.346$ | $15.063 \pm 0.392$ | $15.386 \pm 0.340$ | $15.453 \pm 0.352$ |
| $\delta_0 = 10^{-4}, p = 0.33$ | $16.827 \pm 0.328$ | $17.059 \pm 0.450$ | $17.090 \pm 0.349$ | $17.116 \pm 0.405$ |
| $\delta_0 = 10^{-4}, p = 0.55$ | $21.667 \pm 0.429$ | $21.761 \pm 0.531$ | $21.762 \pm 0.460$ | $21.792 \pm 0.449$ |
| $\delta_0 = 10^{-4}, p = 0.75$ | $18.467 \pm 0.238$ | $18.528 \pm 0.257$ | $18.536 \pm 0.249$ | $18.544 \pm 0.229$ |
| *Cyclic cosine* | | | | |
| $\delta_{\max} = 10^{-3}$ | $5.674 \pm 0.083$ | $9.729 \pm 0.130$ | $11.072 \pm 0.109$ | $11.373 \pm 0.175$ |
| $\delta_{\max} = 10^{-4}$ | $10.354 \pm 0.276$ | $12.264 \pm 0.343$ | $12.661 \pm 0.332$ | $12.712 \pm 0.373$ |

*Table 18.* SGLRW KL divergence under the schedule ablation on UCI logistic regression, as mean $\pm$ standard deviation over 12 seeds.

| Schedule | $B = 1$ | $B = 8$ | $B = 32$ | $B = 64$ |
|---|---|---|---|---|
| *Fixed step size* | | | | |
| $\delta = 10^{-3}$ | $5.007 \pm 0.065$ | $1.073 \pm 0.018$ | $0.333 \pm 0.014$ | $0.230 \pm 0.006$ |
| $\delta = 10^{-4}$ | $5.541 \pm 0.113$ | $4.474 \pm 0.067$ | $4.406 \pm 0.137$ | $4.439 \pm 0.088$ |
| $\delta = 10^{-5}$ | $19.696 \pm 0.428$ | $19.714 \pm 0.447$ | $19.647 \pm 0.501$ | $19.708 \pm 0.476$ |
| *Polynomial decay $\delta_t = \delta_0(1 + t)^{-p}$* | | | | |
| $\delta_0 = 10^{-2}, p = 0.33$ | $5.173 \pm 0.067$ | $1.130 \pm 0.020$ | $0.291 \pm 0.014$ | $0.177 \pm 0.006$ |
| $\delta_0 = 10^{-2}, p = 0.55$ | $3.897 \pm 0.062$ | $0.977 \pm 0.017$ | $0.654 \pm 0.020$ | $0.612 \pm 0.011$ |
| $\delta_0 = 10^{-2}, p = 0.75$ | $4.423 \pm 0.067$ | $2.182 \pm 0.044$ | $2.080 \pm 0.045$ | $2.088 \pm 0.029$ |
| $\delta_0 = 10^{-3}, p = 0.33$ | $4.400 \pm 0.085$ | $2.733 \pm 0.039$ | $2.682 \pm 0.075$ | $2.712 \pm 0.039$ |
| $\delta_0 = 10^{-3}, p = 0.55$ | $9.618 \pm 0.181$ | $8.605 \pm 0.173$ | $8.673 \pm 0.254$ | $8.722 \pm 0.193$ |
| $\delta_0 = 10^{-3}, p = 0.75$ | $16.151 \pm 0.326$ | $14.925 \pm 0.313$ | $14.908 \pm 0.406$ | $14.964 \pm 0.335$ |
| $\delta_0 = 10^{-4}, p = 0.33$ | $17.077 \pm 0.370$ | $16.947 \pm 0.366$ | $16.957 \pm 0.462$ | $17.005 \pm 0.396$ |
| $\delta_0 = 10^{-4}, p = 0.55$ | $20.823 \pm 0.401$ | $21.452 \pm 0.428$ | $21.538 \pm 0.469$ | $21.586 \pm 0.457$ |
| $\delta_0 = 10^{-4}, p = 0.75$ | $17.814 \pm 0.224$ | $18.213 \pm 0.220$ | $18.280 \pm 0.231$ | $18.300 \pm 0.227$ |
| *Cyclic cosine* | | | | |
| $\delta_{\max} = 10^{-3}$ | $4.339 \pm 0.053$ | $1.019 \pm 0.017$ | $0.622 \pm 0.013$ | $0.560 \pm 0.014$ |
| $\delta_{\max} = 10^{-4}$ | $9.336 \pm 0.177$ | $8.431 \pm 0.191$ | $8.468 \pm 0.249$ | $8.483 \pm 0.165$ |

*Table 19.* Clipped-SGLD KL divergence under the schedule ablation on UCI logistic regression, as mean $\pm$ standard deviation over 12 seeds.

| Schedule | $B = 1$ | $B = 8$ | $B = 32$ | $B = 64$ |
|---|---|---|---|---|
| *Fixed step size* | | | | |
| $\delta = 10^{-3}$ | $5.373 \pm 0.049$ | $1.264 \pm 0.023$ | $0.349 \pm 0.009$ | $0.224 \pm 0.008$ |
| $\delta = 10^{-4}$ | $5.499 \pm 0.070$ | $4.417 \pm 0.125$ | $4.411 \pm 0.099$ | $4.414 \pm 0.116$ |
| $\delta = 10^{-5}$ | $19.727 \pm 0.415$ | $19.719 \pm 0.458$ | $19.695 \pm 0.395$ | $19.613 \pm 0.471$ |
| *Polynomial decay $\delta_t = \delta_0(1 + t)^{-p}$* | | | | |
| $\delta_0 = 10^{-2}, p = 0.33$ | $5.580 \pm 0.048$ | $1.382 \pm 0.023$ | $0.317 \pm 0.009$ | $0.172 \pm 0.007$ |
| $\delta_0 = 10^{-2}, p = 0.55$ | $4.197 \pm 0.042$ | $1.024 \pm 0.030$ | $0.648 \pm 0.011$ | $0.598 \pm 0.020$ |
| $\delta_0 = 10^{-2}, p = 0.75$ | $4.552 \pm 0.051$ | $2.123 \pm 0.052$ | $2.003 \pm 0.036$ | $1.995 \pm 0.054$ |
| $\delta_0 = 10^{-3}, p = 0.33$ | $4.419 \pm 0.054$ | $2.671 \pm 0.072$ | $2.664 \pm 0.053$ | $2.671 \pm 0.069$ |
| $\delta_0 = 10^{-3}, p = 0.55$ | $9.544 \pm 0.146$ | $8.525 \pm 0.205$ | $8.603 \pm 0.209$ | $8.617 \pm 0.268$ |
| $\delta_0 = 10^{-3}, p = 0.75$ | $16.101 \pm 0.311$ | $14.860 \pm 0.282$ | $14.878 \pm 0.303$ | $14.839 \pm 0.361$ |
| $\delta_0 = 10^{-4}, p = 0.33$ | $17.078 \pm 0.334$ | $16.950 \pm 0.376$ | $16.978 \pm 0.349$ | $16.902 \pm 0.425$ |
| $\delta_0 = 10^{-4}, p = 0.55$ | $20.833 \pm 0.398$ | $21.458 \pm 0.432$ | $21.546 \pm 0.403$ | $21.516 \pm 0.448$ |
| $\delta_0 = 10^{-4}, p = 0.75$ | $17.813 \pm 0.217$ | $18.212 \pm 0.224$ | $18.280 \pm 0.220$ | $18.281 \pm 0.236$ |
| *Cyclic cosine* | | | | |
| $\delta_{\max} = 10^{-3}$ | $4.620 \pm 0.046$ | $1.076 \pm 0.026$ | $0.621 \pm 0.010$ | $0.544 \pm 0.015$ |
| $\delta_{\max} = 10^{-4}$ | $9.249 \pm 0.154$ | $8.370 \pm 0.195$ | $8.430 \pm 0.185$ | $8.423 \pm 0.241$ |

*Table 20.* 5000-step convergence test on two representative conservative schedules, as mean $\pm$ standard deviation over 3 seeds.

| $B$ | SGLD | SGLRW | Clipped-SGLD |
|---|---|---|---|
| $\delta_0 = 10^{-3}, p = 0.33$ | | | |
| 1 | $0.620 \pm 0.019$ | $2.315 \pm 0.012$ | $2.374 \pm 0.051$ |
| 8 | $0.616 \pm 0.012$ | $0.790 \pm 0.014$ | $0.772 \pm 0.014$ |
| 32 | $0.631 \pm 0.018$ | $0.634 \pm 0.013$ | $0.624 \pm 0.021$ |
| 64 | $0.621 \pm 0.017$ | $0.661 \pm 0.038$ | $0.644 \pm 0.009$ |
| $\delta_0 = 10^{-3}, p = 0.55$ | | | |
| 1 | $3.147 \pm 0.067$ | $4.432 \pm 0.133$ | $4.411 \pm 0.103$ |
| 8 | $4.109 \pm 0.058$ | $4.300 \pm 0.173$ | $4.188 \pm 0.108$ |
| 32 | $4.236 \pm 0.119$ | $4.332 \pm 0.157$ | $4.230 \pm 0.102$ |
| 64 | $4.260 \pm 0.106$ | $4.280 \pm 0.167$ | $4.399 \pm 0.121$ |

