# OpenReview forum: "Robust Stochastic Gradient Posterior Sampling with Lattice Based Discretisation"
_ICML.cc/2026/Conference — ICML 2026 regular_

### Official Review · Reviewer_M8ts · 2026-03-08

**Soundness:** 3
**Presentation:** 3
**Significance:** 2
**Originality:** 3
**Overall Recommendation:** 2
**Confidence:** 5

**Summary:**

This paper introduces a new sampler, Stochastic Gradient Lattice Random Walk (SGLRW), which replaces the Gaussian increments used in SGLD with bounded, coordinate-wise updates. Relative to SGLD, SGLRW is less sensitive to minibatch size and stochastic gradient noise. Empirical results on Bayesian regression and classification show that SGLRW remains stable in settings where SGLD fails, including under heavy-tailed gradient noise, while matching or improving predictive performance.

**Compliance With Llm Reviewing Policy:**

Affirmed.

**Key Questions For Authors:**

1. For the idealized SGLRW scheme, a rigorous convergence analysis to the target posterior is preferred.

2. The bias due to the per-coordinate probability clipping needs to be assessed under the high-dimensional setting.

3. Provide standard deviations for the quantities reported in Tables 1 and 2.

**Limitations:**

See key questions.

**Strengths And Weaknesses:**

Strength: The proposed method is  less sensitive to minibatch size and stochastic gradient noise.

Weakness:

1. Determining the direction of movement for each coordinate incurs an O(d) cost at each iteration, which can be substantial in high-dimensional settings.

2. There is a gap between the theory and the implementation, since probability clipping is used in practice. The theory appears  sound for the idealized SGLRW, but the paper should explicitly distinguish between (i) the idealize algorithm and (ii) the clipped practical algorithm. In addition, it remains unclear whether the bias introduced by clipping increases with the dimension d.

3. Even for the idealized SGLRW scheme, the paper does not appear to provide a rigorous convergence analysis to the target posterior.

4. Standard deviations for the quantities reported in Tables 1 and 2 are not provided, making it difficult to assess statistical significance.

---

> ### Author Rebuttal · Authors · 2026-03-31
>
> We thank the reviewer for their review of our paper. We have addressed all of the reviewer's concerns below, and believe that in doing so we have also addressed all of their key questions. We hope this is sufficient to address their concerns and justify an increase in their score; if not, we welcome further comments and concerns during the next phase.
>
> **Regarding the O(d) costs:**
>
> We would like to clarify that the per-iteration computational complexity of SGLRW is O(d), which is the same as for SGLD and other SGMCMC methods. In all cases, one must compute a stochastic gradient and update each coordinate. The additional sampling of binary directions in SGLRW is negligible compared to the cost of computing the gradient. We also note that SGLRW does not require the Gaussian sampling used by SGLD and related methods. We will clarify this in the manuscript to avoid confusion.
>
> **Regarding the difference between theory and practical implementation:**
>
> We agree with the reviewer that the gap between the practical implementation of SGLRW and its theoretical analysis is an important direction for further study. To this end, we have included additional experiments on the clipping rate in our response to zC8m, and we hope these results alleviate some of the concerns regarding the impact of clipping. We will strengthen our discussion of the distinction between the theoretically idealized version of SGLRW and the practical implementation in Section 4.3.1, where this is already partially addressed.
>
> Since we use coordinate-wise clipping, we do not anticipate any dependence of the clipping behavior on the dimension. While the total number of clipped coordinates will increase with dimension, we do not expect the per-coordinate clipping rate to increase.
>
> **Regarding the convergence study:**
>
> We thank the reviewer for bringing up this concern. We have aligned our convergence study (Theorem 4.2) with the SGMCMC literature, where it is standard to analyze convergence in terms of the mean-squared error (MSE) of ergodic averages, rather than exact convergence of the Markov chain to the posterior distribution (e.g., Chen et al., 2015; Teh et al., 2016; Vollmer et al., 2016).
>
> In this sense, Theorem 4.2 provides a rigorous convergence guarantee: it bounds the MSE between the estimator and the true posterior expectation, and shows that SGLRW achieves the same convergence structure as SGLD with a strictly improved covariance-error contribution. Furthermore, the underlying lattice random walk (LRW) discretization is known to be weakly consistent with the Langevin diffusion (Duffield et al., 2025), providing a principled connection to the target posterior, i.e. the decaying stepsize argument is made in the original SGLD paper (Welling and Teh, 2011) also holds for SGLRW.
>
> That said, we agree that we do not provide a full characterization of the invariant distribution of the discretized chain, and will clarify this distinction more explicitly in the revision.
>
> **Regarding standard deviations:**
>
> We thank the reviewer for raising this concern, as we agree this should be included. Within the rebuttal timeframe, we have re-run all linear regression experiments over 12 seeds to obtain standard deviations for the results presented in Table 1.
>
> These are reported in the table below. Encouragingly, all results remain significant and well within the reported bounds.
>
> For the final version of the paper, we will do the same for Table 2, as well as all new experiments added in response to this and other reviews.
>
> | B | δ_t | SGLD | SGLRW | Clipped SGLD |
> |---|---|---|---|---|
> | 8 | 10⁻³ | 18.93±1.51 | 5.941±0.314 | 18.03±0.935 |
> | 8 | 10⁻⁴ | 0.463±0.051 | 0.189±0.017 | 0.760±0.091 |
> | 16 | 10⁻³ | 6.749±0.555 | 2.244±0.176 | 8.289±0.506 |
> | 16 | 10⁻⁴ | 0.170±0.016 | 0.066±0.004 | 0.193±0.021 |
> | 32 | 10⁻³ | 2.264±0.217 | 0.694±0.060 | 3.312±0.252 |
> | 32 | 10⁻⁴ | 0.086±0.008 | 0.061±0.006 | 0.090±0.008 |
> | 64 | 10⁻³ | 0.759±0.074 | 0.173±0.018 | 1.080±0.102 |
> | 64 | 10⁻⁴ | 0.064±0.005 | 0.058±0.006 | 0.066±0.005 |
> | 128 | 10⁻³ | 0.272±0.031 | 0.074±0.008 | 0.332±0.031 |
> | 128 | 10⁻⁴ | 0.058±0.005 | 0.055±0.008 | 0.060±0.004 |
> | 256 | 10⁻³ | 0.126±0.014 | 0.062±0.006 | 0.132±0.014 |
> | 256 | 10⁻⁴ | 0.056±0.004 | 0.057±0.008 | 0.059±0.004 |
> | 512 | 10⁻³ | 0.082±0.008 | 0.059±0.005 | 0.082±0.007 |
> | 512 | 10⁻⁴ | 0.055±0.004 | 0.058±0.006 | 0.058±0.004 |
> | 1000 | 10⁻³ | 0.059±0.006 | 0.060±0.003 | 0.060±0.003 |
> | 1000 | 10⁻⁴ | 0.055±0.004 | 0.057±0.006 | 0.058±0.004 |
>
> **Summary:**
>
> We again thank the reviewer for their careful reading of our work. We hope to have addressed all of the weaknesses they identified, and that the additional clipping experiments and newly reported standard deviations are sufficient to justify an increase in their score.

---

> > ### Author Rebuttal · Reviewer_M8ts · 2026-04-05
> >
> > Thank you for your response. However, it remains unclear whether the bias introduced by clipping increases with the dimension
> > d. Moreover, from a theoretical perspective, a smaller MSE than that of SGLD does not necessarily imply convergence in distribution of the proposed algorithm.

---

> > > ### Author Response · Authors · 2026-04-08
> > >
> > > We thank the reviewer for their follow-up. We address both remaining points below.
> > >
> > > **Regarding convergence in distribution.**
> > >
> > > We would like to respectfully push back on the reviewer's concern regarding the MSE analysis being insufficient. Our Theorem 4.2 extends Chen et al. (2015), Theorem 3, to SGLRW, showing that the MSE bound for SGLRW is at most that of SGLD everywhere: the drift and discretization contributions are identical, and the covariance contribution satisfies $\mathcal{E}\_\text{cov}^\text{SGLRW} \leq \mathcal{E}\_\text{cov}^\text{SGLD}$. Since the MSE bound for SGLD is known to go to zero under decreasing step sizes satisfying $\delta_t \rightarrow 0$ and $\sum \delta_t = \infty$ (Chen et al., 2015; Teh et al., 2016), the MSE bound for SGLRW will also go to zero under the same conditions. This is not merely a relative comparison, it is a convergence guarantee inherited directly from the established theory for SGLD. We will highlight this more concretely in the paper.
> > >
> > > To address the distinction with convergence in distribution: we agree that a converging MSE over test functions does not logically imply convergence in distribution; these are distinct notions. However, MSE bounds on ergodic averages are the established framework for comparing SG-MCMC discretizations, and are the natural quantity of interest for Bayesian inference, where the goal is to estimate posterior expectations. The foundational works in this area adopt this same framework:
> > > - Chen et al. (2015), Theorem 3: bounds $\mathbb{E}(\hat\phi - \bar\phi)^2$ for general SG-MCMCs.
> > > - Teh et al. (2016): proves consistency of ergodic averages under decreasing step sizes.
> > >
> > > Extending distributional convergence results to SGLRW is a natural direction for future work that we will discuss in the revision.
> > >
> > > **Regarding dimension-dependence of clipping bias.**
> > >
> > > We ran Bayesian linear regression at d ∈ {10, 20, 50, 100, 200} across three (B, δ₀) configs, with 2000 particles and 5 seeds each.
> > >
> > > Table 1: *Per-coordinate clipping rate (SGLRW)*
> > > | d | B=8, δ=1e-3 | B=32, δ=1e-3 | B=128, δ=1e-4 |
> > > |---|---|---|---|
> > > | 10 | 46.1% | 17.2% | 0.0% |
> > > | 20 | 48.1% | 19.2% | 0.0% |
> > > | 50 | 40.7% | 12.1% | 0.0% |
> > > | 100 | 35.6% | 8.0% | 0.0% |
> > > | 200 | 31.7% | 5.3% | 0.0% |
> > >
> > > The per-coordinate clipping rate does not increase with d; at B=8 it decreases from ~48% to ~32%. This is expected: clipping acts coordinate-wise, so the per-coordinate bias depends only on the local gradient magnitude and step size, not on d.
> > >
> > > Table 2: *Normalized mean bias $||\hat\mu - \mu\_\text{true}||\_2 / \sqrt{d}$ :*
> > > | d | SGLD | SGLRW | MC ref |
> > > |---|---|---|---|
> > > | 10 | 0.0015±0.0005 | 0.0047±0.0005 | 0.0007 |
> > > | 20 | 0.0016±0.0003 | 0.0052±0.0002 | 0.0011 |
> > > | 50 | 0.0013±0.0001 | 0.0069±0.0001 | 0.0009 |
> > > | 100 | 0.0013±0.0001 | 0.0051±0.0002 | 0.0009 |
> > > | 200 | 0.0014±0.0001 | 0.0052±0.0001 | 0.0010 |
> > >
> > > Presented in the table at B=8, SGLRW introduces a small excess mean bias over the MC reference of ~0.004, which is constant across d (the slightly elevated d=50 value does not reflect a trend, as d=100 and d=200 return to ~0.005). We further note that at B=32 (omitted here but to be included in the final paper) this excess drops by an order of magnitude, and, similarly, at B=128, δ=1e-4 (zero clipping) all methods are indistinguishable from the MC reference.
> > >
> > > This mean-bias cost is offset by SGLRW's covariance advantage, which is the core contribution of Lemma 4.3.
> > >
> > > *Table 3: The diagonal covariance error $||\text{diag}(\hat\Sigma - \Sigma\_\text{true})||\_2 / \sqrt{d}$ at B=8, δ=1e-3:*
> > > | d | SGLD | SGLRW |
> > > |---|---|---|
> > > | 10 | 0.00295±0.00004 | 0.00147±0.00005 |
> > > | 20 | 0.00335±0.00003 | 0.00163±0.00002 |
> > > | 50 | 0.00242±0.00003 | 0.00120±0.00001 |
> > > | 100 | 0.00194±0.00001 | 0.00097±0.00001 |
> > > | 200 | 0.00170±0.00001 | 0.00080±0.00001 |
> > >
> > > SGLRW's diagonal covariance error is consistently ~2x lower than SGLD, and this advantage is maintained as d grows. Since Lemma 4.3 shows the diagonal of $M_n$ is zero for SGLRW, minibatch noise does not inflate the per-coordinate variances. The overall KL divergence confirms the covariance gain dominates the mean-bias cost: at d=200, SGLRW achieves KL 21.8 vs 47.2 for SGLD.
> > >
> > > -----
> > >
> > > To summarize; With the updated results presented here, we believe both concerns are addressed: (1) the convergence framework follows the standard used for SGLD, and SGLRW achieves strictly better bounds within it; (2) the bias from clipping does not increase with d, and is compensated by SGLRW's covariance advantage over SGLD.

---

### Official Review · Reviewer_zC8m · 2026-03-09

**Soundness:** 3
**Presentation:** 3
**Significance:** 3
**Originality:** 3
**Overall Recommendation:** 5
**Confidence:** 3

**Summary:**

This paper proposes a robust variant of stochastic gradient Langevin dynamics (SGLD) that is less sensitive to minibatch size and gradient noise. The key idea is to replace the usual Gaussian-noise discretization with a lattice random-walk discretization, which yields bounded (or controlled) update magnitudes. This is especially interesting for stochastic gradients with heavy-tailed noise, where standard SGLD can become unstable.

**Compliance With Llm Reviewing Policy:**

Affirmed.

**Final Justification:**

The authors addressed my main concerns by fixing some presentation issues and providing more experiments and clarifications.

**Key Questions For Authors:**

1. **Comparison to other approaches.** Related work includes other approaches to provide robustness. It would help to clarify which are directly comparable here and why the empirical comparison focuses on SGLD and (componentwise) Clipped-SGLD.

2. **Validity constraint and clipping.** As I mentioned above, the condition $\sqrt{\delta_t/2} | \widehat{\partial_i U}(\theta_t; B)| \le 1$ can be restrictive. The paper mentions clipping in practice, but it is unclear whether clipping is always used in experiments, how often it activates under different $\delta_t$ and $B$, and how sensitive results are to this choice. Reporting the fraction of clipped coordinates/steps would improve clarity.

3. **Different behavior of Clipped-SGLD across tasks.** Any intuition why Clipped-SGLD is much closer to SGLRW in UCI logistic regression than in Bayesian linear regression?

4. **Batch-size “sweet spot” (Fig. 5).** Relative improvement for the negative log-likelihood peaks at intermediate batch sizes (that is, 32) and decreases for both smaller and larger minibatches. Could the authors provide any explanation/discussion for this phenomenon?

**Limitations:**

yes

**Strengths And Weaknesses:**

**Soundness**.
The theoretical development seems correct, although I have not checked all the technical details thoroughly. The experimental evaluation is solid, providing empirical evidence of improved robustness.

That said, the validity condition required by the lattice-walk probabilities (the constraint involving $\sqrt{\delta_t/2} | \widehat{\partial_i U}(\theta_t; B)| \le 1$ seems quite strong, and the paper’s practical approach (clipping it to 1 when needed) is not fully specified/discussed in the experiments. It would strengthen the paper to clarify how often this clipping takes place and how sensitive results are to this choice.


**Presentation**.
The paper is well written and easy to follow. I only have some minor comments/issues:
- Some plot text is difficult to read (e.g., Figure 4 has very small font sizes).
- Figure 8 (in the appendix) is not explained in the main text.
- Theorem 4.2 relies on Assumption A.1, but these are only explained in the appendix. A brief summary/comment on what this assumption is (in the main text) would help in readability and understanding.

**Significance**.
The problem addressed is relevant for scalable Bayesian inference and posterior sampling in modern ML problems. Robustness to minibatch noise and tuning sensitivity is important across many application areas.

**Originality**.
The lattice-random-walk discretisation and its stochastic-gradient version are a novel methodological contribution, and the paper positions SGLRW reasonably within related work.

---

> ### Author Rebuttal · Authors · 2026-03-31
>
> First of all, we would like to thank the reviewer for the positive review of our paper. We are glad to read that the paper was easy to follow, and especially appreciate the reviewer's comments regarding the importance and novelty of the work.
>
> We will address the reviewer's comments regarding the small presentational issues directly in the updated manuscript, specifically:
> - We will move Figure 4 to the appendix, where we can increase its size.
> - We agree that Figure 8 is not sufficiently referenced in the main text; we will add an explicit reference and explanation to better guide the reader.
> - We will include a short summary of Assumption A.1 in the main text, while keeping the full details in the appendix.
>
> Below we further comment on the reviewer's concerns regarding soundness and their in-depth questions. We hope that in doing so the reviewer will be able to further increase their score and/or review confidence.
>
> **Regarding clipping:**
>
> We agree with the reviewer that the clipping rate of the model could be studied further. To this end, we are introducing an experiment that compares the effect of batch size and step size on the clipping rate, as the reviewer suggested.
>
> We have provided a first collection of these results in the table below for the linear regression experiment, reporting the average number of clipped coordinates per step. As can be seen, for very small batch sizes and large step sizes there is a considerable amount of clipping for both SGLRW and Clipped-SGLD. In contrast to Clipped-SGLD, however, this clipping does not impact SGLRW as heavily, as our original results on linear regression highlight.
>
> We will include these experiments in the appendix of our paper.
>
> | B \ lr | 1e-2 (SGLRW / Clipped-SGLD) | 1e-3 (SGLRW / Clipped-SGLD) | 1e-4 (SGLRW / Clipped-SGLD) |
> |---|---|---|---|
> | 8 | 0.858 / 0.871 | 0.556 / 0.564 | 0.083 / 0.086 |
> | 16 | 0.810 / 0.824 | 0.430 / 0.428 | 0.018 / 0.019 |
> | 32 | 0.744 / 0.760 | 0.279 / 0.283 | 0.001 / 0.001 |
> | 64 | 0.649 / 0.677 | 0.135 / 0.138 | ~0.000 / ~0.000 |
> | 128 | 0.538 / 0.576 | 0.041 / 0.041 | ~0.000 / ~0.000 |
> | 256 | 0.411 / 0.463 | 0.006 / 0.007 | ~0.000 / ~0.000 |
> | 512 | 0.292 / 0.352 | ~0.000 / 0.001 | ~0.000 / ~0.000 |
> | 1000 | 0.057 / 0.111 | ~0.000 / ~0.000 | ~0.000 / ~0.000 |
>
> **Regarding missing baselines:**
>
> We thank the reviewer for this question and agree that clarifying the comparison is important. Our goal is to isolate the effect of the discretization, as SGLRW modifies the update mechanism itself by replacing Gaussian noise with a bounded lattice-based scheme. For this reason, we compare primarily to SGLD and Clipped-SGLD, which share the same first-order structure and provide a controlled baseline where only the discretization differs. Other approaches aimed at improving robustness (e.g., cyclic step size schedules, preconditioning, or momentum-based methods) correspond to orthogonal design choices and can in principle be combined with SGLRW. To address this point, we include additional experiments in the rebuttal with cyclic and preconditioned variants, showing that SGLRW remains competitive and that these techniques can further improve performance when combined. We will clarify this modular perspective and the rationale for the chosen baselines in the manuscript.
>
> A first set of experiments comparing Cyclic-SGLD and Preconditioned-SGLD with their respective SGLRW-based counterparts (Cyclic-SGLRW and Preconditioned-SGLRW) can be found in our response to reviewer feX8.
>
> **Regarding intuition on Clipped-SGLD and Batch-size "sweet spot":**
>
> We thank the reviewer for raising these interesting observations. We do not have a fully tight explanation for why Clipped-SGLD's relative performance differs across problems. Once we step away from the linear regression case, things unfortunately become harder to reason about. We agree that this is an interesting observation and will add it as a possible direction for future research into Clipped-SGLD, which we note is not the main contribution of this work.
>
> The batch-size sweet spot, we believe, can be explained by the core contribution of this work. SGLRW specifically excels in the small-batch regime; however, once the batch size becomes too small, even SGLRW is dominated by batch noise, which is why the relative improvement decreases for batches smaller than 32. For larger batches, the problem also becomes tractable without the stability benefits of SGLRW. Crucially, while the relative performance decreases for larger batches, SGLRW still outperforms SGLD across small-to-medium batch sizes. We will note this in the paper.
>
> **Summary:**
>
> We again thank the reviewer for their positive review of our paper. We believe that the inclusion of the clipping experiment, as suggested by the reviewer, has significantly improved the quality of the paper. We also hope that the additional experiments allow the reviewer to increase their score and/or confidence level.

---

> > ### Author Rebuttal · Reviewer_zC8m · 2026-04-02
> >
> > Thank you for the rebuttal. I find these new results quite interesting: even with a comparable fraction/number of clipped coordinates (as shown in the table added in the rebuttal), SGLRW can still perform substantially better than Clipped-SGLD (e.g., looking at the KL divergences in Table 1).
> >
> > Overall, the changes you plan to make address my main concerns and reassure me of my positive assessment of the paper. Just to summarise, you will add:
> >  1. **Presentation fixes:** improve clarity/readability of Figures 4 and 8, and include a short summary of Assumption A.1 in the main text.
> >  2. **Clipping-rate study:** report results on the average clipping rate across batch sizes and step sizes (for both SGLRW and Clipped-SGLD).
> >  3. **Baselines:** clarify the rationale for focusing on SGLD and Clipped-SGLD (to isolate discretisation effects), and include additional comparisons with cyclic/preconditioned variants.
> >  4. **Further interpretation of results:** add some discussion/intuition to better interpret the behaviour observed in Figure 5.
> >
> > Given these planned revisions and clarifications, I will update my overall recommendation from weak accept to accept.

---

> > > ### Author Response · Authors · 2026-04-02
> > >
> > > We thank the reviewer for taking the time to acknowledge the rebuttal and for raising their score. We are very glad to read that the reviewer finds the new results interesting.
> > >
> > > We noticed that while the reviewer mentions updating their overall recommendation from weak accept to accept, this change does not yet appear to be reflected in the review score. We wanted to flag this in case it was an oversight.

---

### Official Review · Reviewer_feX8 · 2026-03-11

**Soundness:** 2
**Presentation:** 2
**Significance:** 2
**Originality:** 3
**Overall Recommendation:** 4
**Confidence:** 4

**Summary:**

Stochastic gradient MCMC algorithms are known to be highly sensitive to tuning parameter choices, particularly in high-dimensional problems. This paper proposes a stochastic gradient version of the recently introduced lattice random walk algorithm, which replaces the usual Gaussian Langevin proposal with a more robust binary noise model. In the stochastic gradient context, this leads to an algorithm, SGLRW, that is robust to heavy-tailed gradient noise -- particularly important in the large step size / small batch size regime. In numerical experiments, SGLRW is shown to provide more accurate and stable results than SGLD (with or without gradient clipping).

**Compliance With Llm Reviewing Policy:**

Affirmed.

**Final Justification:**

Authors provided many additional experiments and mostly addressed my other concerns.

**Key Questions For Authors:**

1.  How does SGLRW compare to cSGLD, preconditioned SGLD, and scale-adapted SGHMC?
2. What is the precise experimental setup, both in terms of initialization and choice of step size?
3. If using a decreasing step size, why this choice? How do the methods compare when using a fixed step size?
4. Could you replace Fig. 4 with something more informative?

Satisfactory answers to these questions would lead me to raise my soundness, presentation, and significant scores, and hence improve my overall recommendation.

**Limitations:**

There is no discussion of limitations of the method or the results. Please provide such a discussion.

**Strengths And Weaknesses:**

**Soundness:** The paper provides some solid basic theory to support the method and empirically explores a good variety of experimental settings. However, it uses weak baselines for comparison. I would like to see, for example, a comparison to cyclic SGLD [1], preconditioned SGLD [2], and scale-adapted SGHMC [3].

I also was unconvinced by the use of a decreasing step size, since in practice a fixed step size is sufficient and leads to faster mixing and hence greater computational efficiency. I also wasn't convinced by the stated issues with variance-reduced SGLD. For example, SGLD-CV only requires a single pass over the dataset to compute the gradient at the MAP.

[1] Springenberg, J. T., Klein, A., Falkner, S., and Hutter, F.
Bayesian optimization with robust bayesian neural networks.
In *Advances in Neural Information Processing Systems*, 2016.

[2] Negrea, J., Yang, J., Feng, H., Roy,  D. M., and Huggins, J.H.
Tuning stochastic gradient algorithms for statistical inference via large-sample asymptotics.
*arXiv.org* arXiv:2207.12395 [stat.CO], 2022.

[3] Zhang, R., Li, C., Zhang, J., Chen, C., and Wilson, A. G.
Cyclical Stochastic Gradient MCMC for Bayesian Deep Learning. In *International Conference on Learning Representations*, 2020.

**Presentation:** The presentation was generally clear and easy to follow. However, there were a few issues in the experiments section:

1. Experimental setup was unclear: how is the step size being set? the $\delta_t$ notation doesn't make much sense if its in fact the initial step size $\delta_0$, the step size at iteration t is $\delta_0 / (1 + t)^{0.55}$.
2. How are the algorithms initialized? At at approximate MAP? Some other way?
3. The tables are difficult to parse. One thing that might help is to show the relative error $\min(0, KL / KL_{MC} - 1)$, since this is more interpretable and easier to parse.
4. Figure 4 is impossible to read with extreme zooming of the PDF. I also didn't find it last informative.

**Significance:** The results are potentially very significant, offering a new approach to robust SG-MCMC. However, it is difficult to reach a firm conclusion without comparisons to additional SG-MCMC algorithms.

**Originality:** While the idea of creating a stochastic gradient version of lattice random walk is quite natural, the authors execute it well.

---

> ### Author Rebuttal · Authors · 2026-03-31
>
> We thank the reviewer for their insightful review. We are pleased to read that the reviewer finds the presentation of our paper easy to follow and sees the potential high significance of our work. We have addressed the reviewer's concerns below. As the reviewer will find, we have included new experiments including the suggested benchmarks and made adjustments to clarify the experimental setup.
>
> We hope that with these changes the reviewer's valid concerns are alleviated and they are able to increase their score as indicated.
>
> **Regarding the baselines:**
>
> We thank the reviewer for highlighting additional SGMCMC methods such as cyclic SGLD, preconditioned SGLD, and SGHMC. We agree that these are important directions.
>
> We would like to emphasize that these methods correspond to orthogonal design choices in SGMCMC, such as step size scheduling (e.g., cyclic schedules), preconditioning, and momentum (e.g., SGHMC). In contrast, SGLRW modifies the discretisation mechanism itself, replacing Gaussian updates with a bounded lattice-based scheme. As such, these approaches are not mutually exclusive and can be naturally combined (e.g., cyclic SGLRW, preconditioned SGLRW). In this work, we focused on isolating the effect of the discretisation by comparing to first-order methods with comparable structure, per-iteration cost, and memory cost, and as such had not yet included the additional design choices of the suggested work.
>
> We will make adjustments to highlight this modularity. As part of this clarification, we will add experiments highlighting the improved performance of combining these approaches with SGLRW to our appendix. As a first demonstration, we highlight below the improvement of cyclic SGLRW and preconditioned SGLRW over their base SGLRW counterparts, as well as over their respective cyclic SGLD and preconditioned SGLD versions. Highlighting the benefit of the modified discretisation of SGLRW over SGLD, even with cyclic learning rates and preconditioning considered.
>
> Representative results for Bayesian linear regression:
>
> | B,δ | cSGLD | cSGLRW |
> |-|-|-|
> | 8,1e-2 | - | 2.48 |
> | 32,1e-3 | 0.156 | 0.067 |
> | 256,1e-3 | 0.057 | 0.056 |
> | 1k,1e-3 | 0.054 | 0.056 |
>
> Preconditioning (anisotropic Gaussian, d=100, cond.=200): SGLD fails; with preconditioning, pSGLRW consistently outperforms pSGLD:
>
> | δ | SGLD | SGLRW | pSGLD | pSGLRW |
> |-|-|-|-|-|
> | 0.03 | - | 28.7 | 0.76 | 0.64 |
> | 0.024 | - | 13.5 | 0.73 | 0.62 |
> | 0.020 | 44.6 | 6.09 | 0.74 | 0.66 |
>
> **Regarding decreasing step size:**
>
> We agree with the reviewer regarding the practical importance of running SGMCMC methods with a fixed step size. For this purpose, we have introduced additional experiments in the appendix of our paper. We refer the reviewer to the more in-depth discussion we posted in response to reviewer zCER regarding this.
>
> **Regarding variance reduced SGLD:**
>
> We agree that our initial discussion of variance-reduction methods is too strongly worded. Our intention was to highlight that variance-reduction methods rely on access to such reference gradients or additional bookkeeping, whereas our approach improves robustness directly at the level of the discretisation, without requiring auxiliary computations. These directions are orthogonal, similar to cyclic schedules and preconditioning, and SGLRW could therefore in principle be combined with variance-reduction techniques. We will clarify this distinction and soften our wording in the revision.
>
> **Regarding the experimental setup:**
>
> We fully agree with the reviewer's concerns here; our experimental evaluation setup should have been more clearly described in the main text of the paper. To rectify this, we will add the following clarifications to the paper, based on the reviewer's concerns:
> 1. In the manuscript we use a decaying step size \delta_t = \delta_0 / (1+t)^{0.55}, and in the rebuttal we additionally include fixed step size experiments. We will clarify our notation to highlight that the stated numbers reflect \delta_0.
> 2. All experiments use random initialization. For the linear and logistic regression task we initialise the weights by sampling from a Normal distribution, for the LLM experiments we use standard Xavier initialisation. Different methods use the same initialisation across seeds for fair comparison.
> 3. We have included the relative error to the manuscript.
> 4. We have moved Figure 4 to the appendix, increased its size significantly and added additional exposition. We have replaced it in the main text with discussion of modularity and compatibility with aforementioned techniques.
>
> **Summary:**
>
> We again thank the reviewer for the helpful comments and suggestions. We believe that with the extended experiments using cyclic and preconditioned SGLRW, and the discussion on the modularity of these approaches, the paper has significantly improved. We hope these clarifications strengthen the reviewer's confidence and allow them to increase their score.

---

> > ### Author Rebuttal · Reviewer_feX8 · 2026-04-02
> >
> > Thanks for your detailed and convincing response. Highlighting the modularity of your approach is very important. I have raised my score.

---

> > > ### Author Response · Authors · 2026-04-02
> > >
> > > We thank the reviewer for taking the time to acknowledge the rebuttal and for raising their score. We agree that modularity is very important and would like to thank them again for bringing to our attention in their original review that this should be highlighted better in our work.

---

### Official Review · Reviewer_zCER · 2026-03-12

**Soundness:** 3
**Presentation:** 4
**Significance:** 3
**Originality:** 2
**Overall Recommendation:** 4
**Confidence:** 3

**Summary:**

The authors introduce a method called "Stochastic Gradient Lattice Random Walk". It is based upon a lattice-based discretization of overdamped Langevin dynamics for scalable Bayesian posterior sampling. However, as with standard Langevin models that have normally-distributed noise, their method has bounded binary updates that confine stochastic gradient noise to off-diagonal covariance elements. The authors give the motivation for this method as being that stochastic gradient Langevin dynamics is unstable under small minibatches and heavy-tailed gradient noise.

Their experiment reveals that their method preserves asymptotic correctness.

**Compliance With Llm Reviewing Policy:**

Affirmed.

**Key Questions For Authors:**

Please address the weaknesses I outlined, as I believe they can improve the paper's quality.

**Limitations:**

No, they did not discuss them in their entirety or give a way forward on how future work can improve upon them.

**Strengths And Weaknesses:**

Strength 1: They extend the work in the literature (specifically by Chen et al.) to MSE analyses and thereby prove that minibatch noise enters only off-diagonally in their method. This yields a strictly tighter covariance-error bound on stochastic gradient Langevin dynamics.
Strength 2: They empirically show stability and better performance than existing methods.

Weakness 1: In practical settings, one of the terms needs to be clipped to 1 to ensure that the axiom of normalization holds (i.e., p in [0,1]). This, in my opinion, introduces bias into their method and analysis, to the point that they even admit it.
Weakness 2: I believe that their method overly relies on a very specific schedule. They do not give any quantification of the bias-variance tradeoff, versus fixed step size regimes. This should be explored in more depth as an ablation study.
Weakness 3: In the regularity assumptions, they do not account for nonconvex deep learning posteriors. I believe therefore that the claim of being non-Lipschitz stable is flawed at the very least, or completely incorrect.

---

> ### Author Rebuttal · Authors · 2026-03-31
>
> We thank the reviewer for their valuable feedback and their positive vote to accept the paper. We agree with the reviewer that addressing the weaknesses can further improve the quality of the paper and therefore hope that the adjustments outlined below will increase the reviewer's confidence in their score, and potentially the score itself.
>
>
> **Regarding weakness 1: Clipping bias**
>
> As the reviewer correctly points out, the clipping of the update can skew the moments of the discretisation (for both Clipped SGLD and SGLRW). As the reviewer also noted, this is discussed already in section 4.3.1. Based on the reviewers comments we will extend this section to note that the probability of clipping decays to zero as the stepsize decays to zero. Thus this skew is part of the discretisation bias (which exists for all SGMCMC methods). As such, since using a decreasing step size is generally considered a requirement for correctness in SGMCMC (see, for example, Welling and Teh, 2011), we do not see this as a direct limitation, but do agree that we should highlight it in the limitations and future work section of the paper, as the reviewer suggested.
>
> Additionally, to further highlight that the bias introduced by clipping does not play a major role empirically in SGLRW, we will include a study in the appendix that discusses the clipping rate for various settings of the model. We have provided first results on this in the response to reviewer zC8m.
>
> **Regarding weakness 2: specific schedule**
>
> We thank the reviewer for raising this important point. We agree that fixed step size is a regime of practical relevance. We however believed it most natural to align our experiments with the setting used in our theoretical study, which similar to other stochastic gradient methods relies on a decreasing learning rate. To also cover the practical use of the stochastic gradient methods, we will add duplicates of the experiments currently included in the manuscript, run with a fixed step size, to the appendix of our work.
>
> To present a first set of results, we have run these experiments for the linear regression setting during the rebuttal phase and will expand this to the rest of the experiments for the camera-ready version. As shown in the table below, we observe that SGLD can become unstable, particularly for small batch sizes and larger step sizes, leading to significantly degraded performance. In contrast, SGLRW remains stable and achieves substantially lower error. This is consistent with our theoretical motivation, as the bounded lattice updates mitigate the effect of large stochastic gradients. These results highlight that the improved robustness of SGLRW is also empirically relevant in regimes where fixed step sizes and noisy gradients are used.
>
> | (B, δ) | SGLD | ClipSGLD | SGLRW |
> |---|---|---|---|
> | (8,1e-2) | — | 1348 | 445 |
> | (32,1e-3) | 1045 | 86 | 30 |
> | (256,1e-4) | 0.36 | 0.47 | 0.08 |
> | (1k,1e-5) | 0.056 | 0.059 | 0.057 |
>
> **Regarding weakness 3: Theoretical assumptions**
>
> We agree with the reviewer that we can further strengthen the paper by clarifying our theoretical assumptions, especially regarding the Lipschitz requirement. As the reviewer suggested, we will also add this to the limitations and future work section of our discussion.
>
> To clarify, our comment on non-Lipschitz stability was intended in a narrower sense: lattice/skew-symmetric discretizations are not inherently restricted to globally Lipschitz drifts, and related prior work (Duffield et al., 2025; Iguchi et al., 2026) shows weak-convergence guarantees under substantially weaker assumptions, in particular one-sided and local Lipschitz conditions, rather than global Lipschitz assumptions. In the present paper, Theorem 4.2 further provides formal support for the robustness of SGLRW by showing that, under our assumptions, its covariance-error contribution to the MSE is never larger than that of SGLD, and is strictly smaller when minibatch noise is non-vanishing. We will clarify this in the paper and soften the comment on non-Lipschitz stability to relate to previous related work, such as the rigorous theory in Iguchi et al. (2026).
>
> **Summary**
>
> We again would like to thank the reviewer for the helpful comments and suggestions. As stated, we will address all weaknesses in the updated manuscript, either through further clarification of the theoretical assumptions (weaknesses 1 and 3) and/or by adding additional experiments (weaknesses 1 and 2). Where appropriate (weaknesses 1 and 3), we will also follow the reviewer's suggestion and add a note to the limitations and future work section of the discussion.
>
> **New References**
> - Iguchi, Yuga, et al. "Skew-symmetric schemes for stochastic differential equations with non-Lipschitz drift: an unadjusted Barker algorithm."

---

> > ### Author Rebuttal · Reviewer_zCER · 2026-04-04
> >
> > I do not believe weakness 2 has been resolved. The authors made a good attempt to address my concerns above, but they still need to address them directly.

---

> > > ### Author Response · Authors · 2026-04-08
> > >
> > > We thank the reviewer for acknowledging our rebuttal. We address their remaining concern, weakness 2, below.
> > >
> > >  > *Weakness 2: I believe that their method overly relies on a very specific schedule. They do not give any quantification of the bias-variance tradeoff, versus fixed step size regimes. This should be explored in more depth as an ablation study.*
> > >
> > >  Please find the completed ablation study using different schedules below. The complete study considers 14 settings: 3 fixed step sizes, 9 polynomial decay variations (3 initial step sizes × 3 decay rates), and 2 cyclic schedules, across 4 batch sizes. We will expand this to the other test systems in our camera-ready version. We show 13 of the 14 settings below (the omitted setting, decay δ₀=1e-4 p=0.75, decays too aggressively for both SGLD and SGLRW); the first table contains results for SGLD, the second for SGLRW.
> > >
> > >  SGLD KL divergence:
> > >
> > >  | Schedule | B=8 | B=32 | B=128 | B=1000 |
> > >  |---|---|---|---|---|
> > >  | fixed δ=1e-3 | — | 225±2 | 33.1±0.5 | 3.55±0.06 |
> > >  | fixed δ=1e-4 | 32.2±0.4 | 4.17±0.10 | 0.489±0.016 | 0.087±0.005 |
> > >  | fixed δ=1e-5 | 0.872±0.018 | 0.119±0.014 | 0.064±0.004 | 0.053±0.007 |
> > >  | decay δ₀=1e-2, p=0.33 | — | — | 7.93±0.21 | 0.685±0.029 |
> > >  | decay δ₀=1e-2, p=0.55 | 16.5±0.1 | 1.92±0.04 | 0.250±0.014 | 0.068±0.010 |
> > >  | decay δ₀=1e-2, p=0.75 | 0.863±0.054 | 0.125±0.011 | 0.059±0.005 | 0.062±0.006 |
> > >  | decay δ₀=1e-3, p=0.33 | 10.8±0.2 | 1.19±0.03 | 0.151±0.008 | 0.063±0.009 |
> > >  | decay δ₀=1e-3, p=0.55 | 0.414±0.008 | 0.081±0.004 | 0.060±0.007 | 0.059±0.007 |
> > >  | decay δ₀=1e-3, p=0.75 | 0.068±0.007 | 0.060±0.003 | 0.055±0.005 | 0.056±0.002 |
> > >  | decay δ₀=1e-4, p=0.33 | 0.265±0.018 | 0.071±0.007 | 0.062±0.005 | 0.057±0.008 |
> > >  | decay δ₀=1e-4, p=0.55 | 0.065±0.004 | 0.060±0.007 | 0.053±0.005 | 0.057±0.006 |
> > >  | cyclic δ_max=1e-3 | 1.04±0.03 | 0.139±0.018 | 0.065±0.008 | 0.060±0.003 |
> > >  | cyclic δ_max=1e-4 | 0.292±0.034 | 0.075±0.003 | 0.058±0.006 | 0.059±0.004 |
> > >
> > >  SGLRW KL divergence:
> > >
> > >  | Schedule | B=8 | B=32 | B=128 | B=1000 |
> > >  |---|---|---|---|---|
> > >  | fixed δ=1e-3 | 62.9±0.6 | 21.3±0.2 | 6.08±0.13 | 0.599±0.010 |
> > >  | fixed δ=1e-4 | 8.78±0.13 | 1.29±0.02 | 0.108±0.012 | 0.059±0.005 |
> > >  | fixed δ=1e-5 | 0.344±0.022 | 0.064±0.006 | 0.058±0.003 | 0.060±0.004 |
> > >  | decay δ₀=1e-2, p=0.33 | 35.8±0.3 | 10.4±0.1 | 2.06±0.11 | 0.113±0.009 |
> > >  | decay δ₀=1e-2, p=0.55 | 5.27±0.09 | 0.570±0.036 | 0.069±0.003 | 0.057±0.005 |
> > >  | decay δ₀=1e-2, p=0.75 | 0.352±0.028 | 0.058±0.007 | 0.058±0.002 | 0.061±0.005 |
> > >  | decay δ₀=1e-3, p=0.33 | 3.81±0.12 | 0.346±0.020 | 0.061±0.003 | 0.055±0.004 |
> > >  | decay δ₀=1e-3, p=0.55 | 0.164±0.012 | 0.059±0.004 | 0.059±0.005 | 0.060±0.008 |
> > >  | decay δ₀=1e-3, p=0.75 | 0.058±0.005 | 0.054±0.005 | 0.058±0.005 | 0.060±0.005 |
> > >  | decay δ₀=1e-4, p=0.33 | 0.122±0.011 | 0.058±0.005 | 0.056±0.005 | 0.059±0.006 |
> > >  | decay δ₀=1e-4, p=0.55 | 0.068±0.005 | 0.058±0.007 | 0.061±0.005 | 0.060±0.008 |
> > >  | cyclic δ_max=1e-3 | 0.504±0.029 | 0.060±0.008 | 0.060±0.003 | 0.056±0.007 |
> > >  | cyclic δ_max=1e-4 | 0.149±0.012 | 0.061±0.004 | 0.060±0.005 | 0.055±0.006 |
> > >
> > > We would specifically like to highlight two empirical observations:
> > > 1. SGLRW outperforms SGLD at small batch sizes across all settings shown. The advantage is largest at fixed step sizes, where SGLRW is 3-10x better in KL (e.g. fixed δ=1e-4, B=8: SGLRW 8.78 vs SGLD 32.2). While the difference is largest for the fixed step size, we will stay with using the decaying step size in the remainder of the experiments as this most closely aligns with the presented theoretical results.
> > > 2. At conservative step sizes or large batch sizes, where gradient noise is small, SGLRW and SGLD converge to the same values (~0.055, the MC reference). SGLRW never performs worse than SGLD in regimes where its robustness is not needed. This holds for all the different schedules considered.
> > >
> > > Connecting this back to the theoretical assumptions; at fixed step sizes, both SGLD and SGLRW have a non-vanishing discretization bias, as is standard for all SGMCMC methods (Vollmer et al., 2016; Teh et al., 2016). This is not specific to SGLRW. The theoretical assumption of a decreasing step size for asymptotic exactness is shared by SGLD itself (Welling & Teh, 2011). We will discuss the fixed-step-size bias-variance tradeoff more concretely in the updated manuscript.
> > >
> > > We additionally provide a detailed decomposition of the clipping-induced bias into mean and covariance contributions across dimensions d ∈ {10, 20, 50, 100, 200} in our response to Reviewer M8ts, confirming the bias is dimension-independent and dominated by the covariance advantage of SGLRW over SGLD.
> > >
> > > We thank the reviewer for pushing on this point, as the ablation study has significantly strengthened the paper in our opinion.

---

### Decision · Program_Chairs · 2026-04-30

**Decision:**

Accept (regular)

**Comment:**

The authors propose Stochastic Gradient Lattice Random Walk (SGRLW), a stochastic gradient extension to the Lattice Random Walk discretization.  Compared to (stochastic gradient) Langevin dynamics, the SGLRW sampler has greater robustness to minibatch size while retaining asymptotic correctness.  In common with other SG-MCMC schemes, the method has a non-vanishing discretization bias at fixed step sizes, and the authors have promised to discuss this more concretely in the final version.

The reviewers were mostly cautiously positive about the idea in their initial evaluation, though they raised concern about clipping bias, sensitivity to the step size schedule, and lack of clarity regarding theoretical assumptions.  The reviewers requested further clarification around these points in the final version of the paper, but also thought this a worthwhile contribution to the literature.

The most strongly negative reviewer pointed out that the convergence analysis of the method (in terms of mean squared error for test functions) does not imply convergence on its own; the authors pointed out in response that the analysis of the proposed algorithm is in alignment with the current literature on SG Langevin dynamics, and that MSE of ergodic averages is often the error of interest in Bayesian inference applications.  Apart from this negative reviewer, whose dominant concern in the end seemed to be about this lack of theory about convergence in distribution, the other reviewers all favored acceptance after the rebuttal.  Moreover, in the reviewer discussion, the reviewers favoring acceptance strongly objected to judging this new method based on different standards than are generally applied to other stochastic gradient MCMC approaches.  With due respect to the negative reviewer, I agree with the positive reviewers (and the authors) that this is not a reasonable basis for rejection of a generally interesting paper that seems to advance the current state of the art for SG-MCMC methods.